# Interactions between atmospheric composition and climate change - Progress in understanding and future opportunities from AerChemMIP, PDRMIP, and RFMIP

Stephanie Fiedler[1], Vaishali Naik[2], Fiona M. O'Connor[3], Christopher J. Smith[4], Paul Griffiths[5], Ryan Kramer[6], Toshihiko Takemura[7], Robert J. Allen[8], Ulas Im[9], Matthew Kasoar[10], Angshuman Modak[11], Steven Turnock[12], Apostolos Voulgarakis[13], Duncan Watson-Parris[14], Daniel M. Westervelt[15], Laura J. Wilcox[16], Alcide Zhao[16], William J. Collins[17], Michael Schulz[18], Gunnar Myhre[19], and Piers M. Forster[20]

[1]GEOMAR Helmholtz Centre for Ocean Research Kiel & Faculty of Mathematics and Natural Sciences, Christian-Albrechts-University of Kiel, Germany
[2]NOAA Geophysical Fluid Dynamics Laboratory, Princeton, NJ, USA
[3]Met Office Hadley Centre, Exeter, UK & Global Systems Institute, Faculty of Environment, Science and Economy, University of Exeter, Exeter, UK
[4]Priestley International Centre for Climate, University of Leeds, Leeds, UK & International Institute for Applied Systems Analysis (IIASA), Laxenburg, Austria
[5]National Centre for Atmospheric Science, Dept of Chemistry, Cambridge University, Cambridge, United Kingdom
[6]Climate and Radiation Laboratory, NASA Goddard Space Flight Center, Greenbelt, MD, USA & Joint Center for Earth Systems Technology, University of Maryland Baltimore County, Baltimore, MD, USA
[7]Research Institute for Applied Mechanics, Kyushu University, Kasuga, Fukuoka, Japan
[8]Department of Earth and Planetary Sciences, University of California Riverside, Riverside, CA, USA
[9]Aarhus University, Department of Environmental Science, Roskilde, Denmark & Interdisciplinary Centre for Climate Change (iClimate), Roskilde, Denmark
[10]Leverhulme Centre for Wildfires, Environment and Society, Department of Physics, Imperial College London, London, UK
[11]Department of Meteorology, Stockholm University, Stockholm, Sweden
[12]Met Office Hadley Centre, Exeter, UK & University of Leeds Met Office Strategic (LUMOS) Research Group, University of Leeds, UK
[13]Department of Chemical and Environmental Engineering, Technical University of Crete, Greece & Leverhulme Centre for Wildfires, Environment and Society, Department of Physics, Imperial College London, London, UK
[14]Atmospheric, Oceanic and Planetary Physics, Department of Physics, University of Oxford, Oxford, UK; now: Scripps Institution of Oceanography and Halıcıoğlu Data Science Institute, University of California San Diego, USA
[15]Lamont-Doherty Earth Observatory of Columbia University, Palisades, NY, USA & NASA Goddard Institute for Space Studies, New York, NY, USA
[16]National Centre for Atmospheric Science, Department of Meteorology, University of Reading, Reading, UK
[17]Department of Meteorology, University of Reading, Reading, UK
[18]Norwegian Meteorological Institute, Oslo, Norway
[19]Center for International Climate and Environmental Research in Oslo (CICERO), Oslo, Norway
[20]Priestley International Centre for Climate, University of Leeds, Leeds, UK

**Correspondence:** Stephanie Fiedler (sfiedler@geomar.de)

**Abstract.** The climate science community aims to improve our understanding of climate change due to anthropogenic influences on atmospheric composition and the Earth's surface. Yet not all climate interactions are fully understood and uncertainty

in climate model results persists as assessed in the latest Intergovernmental Panel on Climate Change (IPCC) assessment report. We synthesize current challenges and emphasize opportunities for advancing our understanding of the interactions between atmospheric composition, air quality, and climate change, as well as for quantifying model diversity. Our perspective is based on expert views from three multi-model intercomparison projects (MIPs) - the Precipitation Driver Response MIP (PDRMIP), the Aerosol and Chemistry MIP (AerChemMIP), and the Radiative Forcing MIP (RFMIP). While there are many shared interests and specialisms across the MIPs, they have their own scientific foci and specific approaches. The partial overlap between the MIPs proved useful for advancing the understanding of the perturbation-response paradigm through multi-model ensembles of Earth System Models of varying complexity. We discuss the challenges of gaining insights from Earth System Models that face computational and process representation limits and provide guidance from our lessons learned. Promising ideas to overcome some long-standing challenges in the near future are kilometer-scale experiments to better simulate circulation-dependent processes where it is possible, and machine learning approaches where they are needed, e.g., for faster and better sub-grid scale parameterizations and pattern recognition in big data. New model constraints can arise from augmented observational products that leverage multiple datasets with machine learning approaches. Future MIPs can develop smart experiment protocols that strive towards an optimal tradeoff between resolution, complexity, and number of simulations and their length, and thereby, help to advance the understanding of climate change and its impacts.

## 1   Introduction

A central aim of climate science is to advance our understanding of how the Earth system responds to human activities. This endeavor involves the assessment of numerous spatiotemporally changing variables in the Earth system, which can be determined by multiple, interacting physical, chemical, and biological processes. For example, changes in irradiance, land use, and atmospheric composition, including for instance aerosols and their precursors, greenhouse gases such as carbon dioxide and methane, perturb the radiation fluxes in and at the top of the atmosphere and hence the Earth's radiation balance. On a timescale of several decades, the Earth's temperature is controlled by a balance between the net amount of absorbed sunlight (solar radiation) and the radiation emitted by the planet and its atmosphere (terrestrial radiation). A perturbation of this balance is called a "radiative forcing" - a concept embedded in the study of the physical basis of climate (Ramaswamy et al., 2019) - and is measured as energy flux in W m$^{-2}$.

Changes to atmospheric composition have distinct effects on the Earth's energy budget and climate, which are classified into radiative forcing, climate response, and feedbacks. Instantaneous radiative forcing (IRF) is the initial change in radiation fluxes that arise from a perturbation in a climate forcer, which could be, for instance, associated with increased greenhouse gas concentrations in the atmosphere due to anthropogenic emissions, in the absence of other changes. IRF excludes any changes in the system other than an imbalance in the Earth's top-of-the-atmosphere (TOA) radiation budget and is a diagnostic output from Earth System Models (ESMs). The system responds to this imbalance by equilibrating a new temperature at which the net TOA fluxes are in balance when they are averaged over several decades. Climate responses can be amplified or weakened via positive or negative feedbacks that are induced by changes in physical and chemical processes. Balancing the system after an initial

perturbation can take several hundred years because of the slow response of ocean temperatures. There are also fast processes influencing the TOA flux that arise from a change in atmospheric composition, even in the absence of surface temperature changes. Examples of such changes, known as rapid adjustments, occurring in the atmosphere include stratospheric cooling due to increasing carbon dioxide concentrations (Manabe and Wetherald, 1967), chemical adjustments due to changes in emissions of reactive trace gases (Thornhill et al., 2021b; O'Connor et al., 2021), and changes in clouds due to circulation changes (e.g. Gregory and Webb, 2008; Bretherton et al., 2013; Merlis, 2015), as well as cloud changes due to shortwave radiation absorption by methane (Allen et al., 2023) and black carbon (Stjern et al., 2017). Moreover, changes in wind-dependent emissions of aerosols that occur due to circulation adjustments can be interpreted as chemical adjustments, although changes in aerosol emissions can occur with surface-temperature responses and would fall into the category of chemical feedback in that case. Relevant examples are adjustments and feedbacks that modify desert-dust and sea-spray aerosol emission changes. Effective Radiative Forcing (ERF), quantified at the TOA, encompasses both the IRF and the contributions from rapid adjustments. Climate responses require an assessment of changes in the fully coupled atmosphere-ocean system determining the surface temperature. These segments in the perturbation-response paradigm of climate science are schematically depicted in Figure 1.

Understanding and quantification of the different segments in the perturbation-response paradigm of climate science are obtained through experiments with Earth System Models, although other methods for some of the segments exist, e.g., radiative transfer models to compute IRF. Current ESMs vary in their design and implementations, e.g., concerning different parameterization schemes, dynamical cores, spatial grids, numerical integration, tuning, and boundary data. These imply diversity in the level of complexity for representing physical, chemical, and biological processes, and how represented processes interact. For example, some ESMs prescribe aerosol properties while models with additional process complexity simulate the complex evolution and interactions of aerosols and their precursors in the atmosphere (Figure 1). The simulated aerosols may interact with the radiative transfer and formation of cloud droplets and ice crystals, but not all ESMs simulate all interactions with the cloud microphysics. The climate modeling community creates multi-model ensembles of a common set of ESM experiments with the same perturbations applied. The simulated climate responses can differ across a multi-model ensemble. This diversity in responses may for instance be due to differences in process complexity and interactions within the respective ESMs. Experimental protocols are used to create multi-model ensemble simulations for specific ESM experimental setups. These aim to better understand the reasons for the diversity in climate responses and feedback and to create future climate projections.

Results from multi-model intercomparison projects (MIPs) are widely used to advance scientific understanding and inform stakeholders on climate change. The most prominent example is the Coupled Model Intercomparison Project (CMIP, Meehl et al., 2000) that has contributed through multiple phases to the assessment reports of the Intergovernmental Panel on Climate Change (IPCC, Meehl, 2023), e.g., the sixth phase of CMIP (CMIP6, Eyring et al., 2016) created experiments that were also used in the sixth IPCC assessment report (IPCC-AR6). The basic idea of a MIP is also used for different foci either outside of or endorsed by the CMIP consortium. For example, the Aerosol Model and Measurement Comparisons (AeroCom) focuses on the role of aerosols in the climate system (e.g., Gliß et al., 2021; Textor et al., 2006), the Chemistry-Climate Model Initiative (CCMI) on the interactions between atmospheric chemistry and climate change (e.g., Morgenstern et al., 2017; Abalos et al., 2020), the Task Force on Hemispheric Transport of Air Pollution (TF HTAP) on global air quality modeling (e.g., Wild et al.,

2012; Turnock et al., 2018) and the Precipitation Driver Response Model Intercomparison Project (PDRMIP, Myhre et al., 2017) on the role of anthropogenic and natural drivers for different precipitation responses. Several MIPs were endorsed during CMIP6, such as the Aerosol and Chemistry MIP (AerChemMIP, Collins et al., 2017) and the Radiative Forcing MIP (RFMIP, Pincus et al., 2016). While the specific foci for AerChemMIP and RFMIP varied, both MIPs were driven by the common goal of better characterizing the preindustrial to present-day radiative forcing and determining climate responses to these forcings.

The aims here are to synthesize and emphasize what has been learned on the experimental design, conceptual thinking, and diagnostic requests through connecting the scientific communities of AerChemMIP, RFMIP, and PDRMIP under one umbrella named TriMIP (Figure 2). In so doing, we discuss the challenges of understanding multi-model climate responses and identify potential opportunities to make further advances in the research areas of these MIPs. Each of the MIPs had their own perspective on how to accomplish their goals, but sufficient similarities inspired a series of joint TriMIP meetings. Similar conceptual understanding helped to build common ground across the community that proved useful to contribute to the same overarching goal – the advancement in understanding of our planet's changing climate.

## 2   Scientific Advancement

### 2.1   MIPs's Key Results

The three MIPs sought to advance the understanding of modern climate change due to anthropogenic influences. MIPs address specific research questions and, in comparison to studies with a single ESM, consider structural differences concerning the design and the level of complexity between ESMs. The multi-model spread in the response allows the quantification of a model-based uncertainty for the answer to the MIP's question. While the MIPs share the conceptual idea of the perturbation-response paradigm (Figure 1), they focus on different segments in the paradigm. RFMIP focused on an improved understanding of the radiative forcing diversity to anthropogenic perturbations in atmospheric composition (e.g., Smith et al., 2020a), and PDRMIP on precipitation responses to atmospheric composition changes (e.g., Richardson et al., 2018). AerChemMIP also focused on quantifying radiative forcing and responses but addressed more segments in the paradigm. Specifically, all participating models in AerChemMIP simulated atmospheric composition based on emissions, transport, chemical transformations, and deposition, making these models more complex in their process representation and interactions than was necessary for participation in the other two MIPs (e.g., Thornhill et al., 2021a). The three MIPs used, to some extent, similar experimental strategies, but developed and adopted their own experimental protocol with a certain class of models in mind, e.g., AerChemMIP required more interactive processes than the other two MIPs. PDRMIP began earlier and to some degree inspired the experimental protocols of AerChemMIP and RFMIP. There are ensembles of ESM experiments of different complexity, spatial resolutions, number of realizations, and length of experiments in the three MIPs. Tables 1–2 summarize key results along with the used experiments, organized by topics that were addressed by the three MIPs.

The primary objective of PDRMIP was to understand global and regional responses of precipitation statistics to the radiative forcing of $CO_2$, $CH_4$, $O_3$, irradiance, and sulfate and black carbon aerosols (Myhre et al., 2017). Based on eleven aerosol-climate models contributing to PDRMIP, energy budgets, and the hydrological cycles were inter-compared for fast (days to

months) and slow (years to decades) response times (e.g., Myhre et al., 2017; Samset et al., 2016; Sillmann et al., 2019).
Rapid adjustments are a key in understanding precipitation responses (e.g., Hodnebrog et al., 2020; Myhre et al., 2018; Smith et al., 2018). Taking advantage of multiple forcing agents in PDRMIP, model spreads in radiative forcing and efficacy for the forcing agents were quantified (Forster et al., 2016; Richardson et al., 2019), and responses to greenhouse gases and aerosols inter-compared across the PDRMIP ensemble (Sillmann et al., 2019; Stjern et al., 2020). Others examined the climate response to forcing for selected regions, e.g., the monsoon regions, the Arctic, and the Mediterranean (Stjern et al., 2019; Tang et al., 2018; Xie et al., 2020). Multiple realizations of such climate change experiments, i.e., a set of simulations with a small initial perturbation but otherwise identical setups, are required to separate the internal variability from the forced response, especially at regional scales and for variables such as precipitation in PDRMIP.

The main goals of AerChemMIP were to quantify the climate and air quality responses of aerosols and chemically reactive gases, specifically near-term climate forcers (NTCFs) including methane, tropospheric ozone, aerosols, and their precursors (Collins et al., 2017). The term NTCF is used in IPCC-AR6 and refers to the same term as short-lived climate forcers (SLCFs) used by Collins et al. (2017). Both NTCFs and SLCFs refer to radiatively active atmospheric constituents whose climate effects occur primarily within two decades of their emission or formation. Amongst TriMIP, AerChemMIP emphasized transient coupled atmosphere-ocean simulations to estimate the real-world evolution and timing of anthropogenic and natural emission changes and associated air quality and climate responses. AerChemMIP experiments were novel in CMIP6 in that they followed the "all-but-one" design, whereby the forcing of interest is held fixed. For example, *hist-piNTCF* simulations are parallel to *historical* simulations, except anthropogenic emissions of NTCFs are held fixed at pre-industrial level (1850) and all other forcing agents evolve as in a historical simulation (*hist*) facilitating attribution of historical climate responses to NTCF emissions. Such an experimental design seeks to minimize the contribution of non-linear climate responses that may occur under the more traditional experimental design for attribution where only the emissions or concentrations of the species of interest are perturbed (Deng et al., 2020). The model output from AerChemMIP was, for instance, used to investigate 21st-century climate and air quality responses to future NTCF changes (Table 1).

Another focus of AerChemMIP was to quantify non-$CO_2$ biogeochemical feedbacks (Thornhill et al., 2021a) with an AerChemMIP-specific experimental design that is unique in CMIP6. It implied a set of idealized simulations with fixed boundary conditions, except for the preindustrial natural emissions or concentrations that are systematically doubled across the ensemble of simulations, e.g., for dust aerosols *piClim-2xdust*. Pairing the radiative fluxes from these experiments with a parallel preindustrial control gives ERF per Tg yr$^{-1}$ change in emissions or concentrations of the climate forcer. The result allowed to obtain the feedback parameter (W m$^{-2}$ per K) for the climate forcer through scaling the simulated changes in emission fluxes per K temperature change from the 4x$CO_2$ experiments of CMIP6. The protocol of AerChemMIP also included transient historical simulations with prescribed sea-surface temperatures (SSTs) to diagnose transient ERFs. Similar to the coupled experiments, these simulations followed the "all-but-one" experimental strategy. Including such analogous prescribed-SST experiments allowed for a better understanding of the drivers of the climate response in the fully coupled experiments (e.g., Allen et al., 2020, 2021). Furthermore, time-slice experiments performed with emissions of one species set to

the present-day value but all other boundary data held fixed at pre-industrial values facilitated quantification of emissions-based ERFs, a policy-relevant metric (Thornhill et al., 2021b).

RFMIP focused on accurately quantifying and identifying errors in the radiative forcing of composition changes in CMIP6 models (Pincus et al., 2016). The largest of the three parts of RFMIP (RFMIP-ERF) was the quantification of ERF across CMIP6 models using a time-slice approach similar to AerChemMIP. It allowed the first quantification of the CMIP inter-model spread in ERF for all major climate forcers as bulk estimates, i.e., for all anthropogenic aerosols taken together, and of the contribution from rapid adjustments to ERF (Smith et al., 2018, 2020a). The second part of RFMIP (RFMIP-IRF) focused on

the IRF excluding contributions from rapid adjustments. Errors in IRF of greenhouse gases were identified using benchmark calculations from line-by-line models (Pincus et al., 2020). The third RFMIP part (RFMIP-SpAer) assessed model differences in ERF for identical anthropogenic aerosol optical properties and effects on clouds. Participating in RFMIP-SpAer required implementing the simple-plumes parameterization (MACv2-SP, Stevens et al., 2017), which was a new approach in CMIP6. The pilot study for RFMIP-SpAer demonstrated the retention of model spread in ERF when moving to identical anthropogenic

aerosols due to differences in the atmospheric host models (Fiedler et al., 2019). Through the combined analysis of output from RFMIP-ERF and RFMIP-SpAer, reasons for model differences in anthropogenic aerosol forcing were inferred (Fiedler et al., 2023).

## 2.2 MIP's Cross-linkages

A major advancement from the synergy between the three MIPs was the widespread adoption of a consistent methodology to

155 quantify radiative forcing within and outside of the three MIPs which facilitated easier comparisons across CMIP6. Estimates of ERF are key in the perturbation-response paradigm by characterizing the influence on the radiation budget due to a perturbation. Yet, a consistent diagnosis of ERF was not possible in CMIP5 (Collins et al., 2017). Specifically, RFMIP helped to establish a consistent practice for diagnosing ERF for CMIP6 and related activities, building on experiences from PDRMIP (Forster et al., 2016). Amongst several approaches to quantifying forcing, graphically summarized in Figure 3, there are two methods

widely used now to estimate ERF from models. Firstly, ERF can be estimated by extrapolating the relationship between the radiation imbalance and temperature change in coupled atmosphere-ocean model experiments subject to abrupt concentration increases of the forcing agent (Regression method, Gregory et al., 2004). Secondly, ERF can be determined by suppressing ocean-temperature changes and calculating the ERF as the radiation imbalance relative to an experiment without the forcing agent (Fixed sea-surface temperature method, Hansen et al., 2005). In this context, the common use of pre-industrial control

experiments in RFMIP and AerChemMIP, i.e., experiments with atmospheric composition set to 1850 levels, proved valuable as a common reference to estimate ERFs from ESMs in CMIP6. RFMIP further requested results from additional diagnostic calls to the radiation schemes, also known as double and triple radiation calls, that enabled calculations of the IRF (Chung and Soden, 2015) and a better understanding of contributions from different processes to ERF. Double calls typically refer to IRF calculations, whereas the term triple calls is used for separating cloud-mediated effects from direct effects of aerosols. Such

model diagnostics for IRF helped to quantify the contribution of adjustments to ERF estimates in the ESMs used in CMIP6

(e.g., Smith et al., 2020a) and to separate direct and cloud-mediated effects following the method by Ghan (2013) in RFMIP experiments (e.g., Fiedler et al., 2023).

The RFMIP protocol included experiments to diagnose radiative forcing for greenhouse gases and aerosols as bulk quantities with setups parallel to the CMIP6 experiments for the "Diagnostic, Evaluation, and Characterization of Klima" (DECK). The RFMIP tier 1 experiments were carried out by many modeling centers. Some of these contributions, e.g., from UKESM1 and CNRM, arose because the experimental setup was identical to the request in AerChemMIP. It meant that the technical workflow for performing and postprocessing the experiments was already in place such that contributing another variant of such experiments required only little effort. Due to the parallel setup of the RFMIP experiments to those requested in DECK and the additional overlap of experiment requests with other MIPs (DAMIP), RFMIP experiments also allowed model analyses of climate responses and climate feedbacks for well-estimated radiative forcing. AerChemMIP further separated contributions to radiative forcing into individual gases and NTCFs including different aerosol species. As such, the AerChemMIP experiment request was tailored to gain insights into why model differences in the forcing-response paradigm arise based on individual perturbations in atmospheric composition.

Experiment requests that were differently designed in RFMIP and AerChemMIP for a similar purpose were the transient historical experiments to attribute the response to individual perturbations. Specifically, RFMIP applied the "only" experimental design where the quantity to be assessed varied over the historical period while all other boundary conditions were kept at the pre-industrial level (*piClim-histX*, where X is the forcing of interest), whereas AerChemMIP applied the "all-but-one" design where the quantity to be assessed was fixed at the pre-industrial level while all other climate forcers varied over the historical period (*histSST-_piX*). These differences in the setup hold the potential to understand where interactions and potential feedbacks arising from chemical composition changes play a role for the climate response, which has not yet been fully explored with the existing MIPs, though individual model studies are being undertaken (e.g., Simpson et al., 2023).

The three MIPs benefited from being embedded in a landscape of other initiatives, with close connections to CMIP on the one hand and specialist MIPs like AeroCom, CCMI, and TF HTAP on the other hand. The community of PDRMIP, AerChemMIP, and RFMIP can therefore be seen as a bridge between the global climate modeling community of CMIP6 and the specialized communities for aerosols and atmospheric chemistry. This setting allows CMIP to benefit from expert subject-specific knowledge that would otherwise be missing. One example is PDRMIP, which began before CMIP6 and had a guiding role for the later MIPs concerning the already mentioned practice of estimating ERF, the parallel use of fully coupled and fixed SST experiments, the choice of perturbation magnitudes and experiment length to quantify forcing and response, as well as the introduction of new model diagnostics. Another example is AerChemMIP, which adopted recommendations for the diagnostic requests and experimental design (e.g., Young et al., 2013; Archibald et al., 2020) from previous non-CMIP6 initiatives.

Coming together of the three MIP communities under the TriMIP umbrella facilitated efficient communication of knowledge gaps and coordination of analysis of multi-model output to address these gaps resulting in publications in peer-reviewed journals. Since several authors of the IPCC-AR6 also participated in TriMIP, the MIP-based publications were tailored to the needs of the IPCC-AR6 working group 1 (WG1) including analysis of ERF (Smith et al., 2020a; Thornhill et al., 2021b), non-$CO_2$ biogeochemical feedbacks (Thornhill et al., 2021a), and climate (Allen et al., 2020, 2021) and air quality responses

(Turnock et al., 2020) to changes in NTCFs. In fact, some key articles based on the experiments were written and submitted very close to the IPCC-AR6 WG1 deadline, which might not have been completed in time if that exchange had not happened. Submission of model output and analyses continued thereafter and are partly still ongoing at the time of writing. We expect this development to continue for several years, although with a decline in new CMIP6 model output until a quorum of CMIP7 model output becomes available. Looking at the history of the use of CMIP data, we would also expect that the output of RFMIP and AerChemMIP will be re-used later for documenting progress across their phases, e.g., for the ERF, which is also often done for tracking progress across CMIP phases.

## 3    Challenges in the MIP's research

A major challenge to further advancing the understanding of climate change with ESMs is that differences in their results for individual segments of the perturbation-response paradigm are not independent of other segments. Specifically, a model-to-model difference in a climate response might be caused by various segments in the paradigm. For instance, the same emissions can lead to different ERFs, the same ERF can induce different climate responses and the same response can trigger different feedbacks across multi-model ensembles. In multi-model studies, one therefore sees inter-model spreads in forcing for the same change in atmospheric composition and model-dependent climate responses to the same forcing involving different types and magnitudes of feedbacks. This challenge is addressed by the three MIPs by suppressing interactions for one segment in the perturbation-response paradigm to advance the understanding of another segment. In this regard, a common approach across the three MIPs is the restriction of model diversity in some parts in order to better characterize and ultimately understand model diversity in others. Methods to separate out some of these model differences include experiments using, for instance, prescribed aerosols (e.g. Fiedler et al., 2019) or reactive trace gases (e.g. Checa-Garcia et al., 2018), which makes the assessment of the contribution of different processes to model diversity more tractable. Such experiments have also been used in the AeroCom community for a better understanding of reasons for model differences in aerosol forcing (Stier et al., 2013) and circulation responses to idealized aerosol forcing (Voigt et al., 2017). Specifically, PDRMIP asked for prescribing the same aerosol information in models to circumvent some aerosol-related sources of model diversity. Such an experimental design allows a deeper exploration of a subset of model components contributing to model diversity - in this case, the translation of aerosol concentration to radiative forcing and the climate response, by removing other sources of model differences. Along similar lines, AerChemMIP allows for the chemical processing of aerosols and reactive gases, and removed feedbacks by performing experiments with prescribed sea-surface conditions. Finally, RFMIP aimed to understand how much of the climate response to a perturbation is due to changes in atmospheric composition rather than due to feedbacks. To that end, RFMIP requested experiments with prescribed sea-surface conditions similar to AerChemMIP to obtain precise model estimates of ERF. The three MIPs, therefore, addressed model differences arising from the segments in the perturbation-response paradigm in a complementary manner for addressing their specific research questions.

### 3.1 Computational Capacity Abyss

#### 3.1.1 Tradeoffs across MIPs

MIPs in CMIP6 as a whole asked for many experiments that jointly placed a big computational demand on climate modeling centers. The requested experiments were designed to address the MIP-specific scientific questions. The three MIPs discussed here contributed to that demand, and the diversity of research interests across the modeling centers meant that some experiments received more attention than others. Setting priorities with tiers was useful to the extent that it highlighted the priority of experiments from the MIP's perspective. In so doing, the tiers guided the participating modelers to set a focus on some experiments to have a larger model ensemble where the MIPs wanted contributions the most. However, in retrospect, some of the Tier 2 experiments may have been more useful than Tier 1. An example here is *piClim-histaer* (Tier 2) from RFMIP, which quantified the spread in magnitude and timing of historical aerosol forcing in CMIP6 models, was informationally rich, and a contributing factor in deriving the aerosol ERF time series for IPCC-AR6 WG1.

Experiments following already known strategies with standard output requests are quicker to set up. These have the advantage that no additional personnel is needed to implement newly requested diagnostic output that is not yet available in the standard variable list of ESMs, e.g., for RFMIP-IRF. On the contrary is an experiment design that needs the implementation of a new parameterization, e.g., for RFMIP-SpAer, which requires dedicated human resources at the modeling center to carry out the work including coding, testing, and performing the experiments. In this case, it takes longer to finish the experiments and to do the associated scientific exploitation, e.g., in the case of RFMIP-SpAer several years after the work began (Fiedler et al., 2023), which is long compared to easy experiments that modelers can quickly set up via a simple change in a setting for performing an experiment, e.g., for RFMIP-ERF, thanks to prior work on the development and testing of models.

A greater number of experiments performed creates more data for statistical analyses and for addressing a variety of research questions, but it is taxing in light of restricted resources. In preparation for the next phase of AerChemMIP and RFMIP, the question of the type and number of experiments in the experimental protocol is therefore revised based on refined research questions. The intention is to keep the computational burden for modeling centers as small as possible. In this process, intended activities are coordinated with other initiatives close to the interests of AerChemMIP and RFMIP, e.g., via a series of workshops organized by us and others. It potentially allows to free some resources and to simplify workflows, e.g., to generate larger ensembles of identical multi-purpose experiments to account for internal variability like done for CMIP6 historical experiments. One such experiment type from our community would be transient coupled experiments to attribute climate responses to different perturbations.

In preparation for the second phase of AerChemMIP and RFMIP, we reviewed the current status of the experiments and their usage in peer-reviewed publications, summarized in Table 3. A total of 67 models performed CMIP6 *historical* experiments (published via ESGF, June 2023) that were used in as many as 15100 publications (listed by Google Scholar, June 2023). Available model output to assess differences in forcing and response was, however, limited, e.g., output for the mid-visible aerosol optical depth is available only for 45 out of the 67 models providing *historical* experiments. Most of the *historical* experiments (40) are performed with NTCF emission-driven models. The ESMs with prescribed aerosols (19) in the *histor-*

*ical* experiments mostly (13) used the MACv2-SP parameterization (Stevens et al., 2017). MACv2-SP was developed in the framework of RFMIP and is, due to the unexpected relatively broad implementation in ESMs, now included in the work of the CMIP climate forcings task team, although the targeted exploitation of MACv2-SP in RFMIP-SpAer was with one publication (Fiedler et al., 2023) small compared to the usage of other experiments of RFMIP and AerChemMIP so far.

RFMIP and AerChemMIP received in total output from 103 experiments leading to 214 publications to date. We separate the RFMIP and AerChemMIP experiments here into three classes, namely experiments with full coupling between the atmosphere and ocean (*hist-X*), with prescribed sea-surface temperatures and sea-ice at pre-industrial level (*piClim-X*), and with prescribed transient changes in sea-surface temperatures and sea-ice from a historical experiment (*histSST-X*). Inter-comparing these classes, *piClim-X* experiments were performed the most with a total of 50 contributing models followed by *hist-X* with 36

models. However, *hist-X* is used three times more often in scientific publications (146) compared to *piClim-X* (52). The higher computational demand of *hist-X*, therefore, seems justified by the much larger scientific output compared to the experiments without a coupled ocean (*histSST-X* and *piClim-X*), measured by the number of published articles.

### 3.1.2    Tradeoffs within MIPs

Available computational capacity affects the experiments for MIPs and the priorities at modeling centers performing many

model experiments for diverse MIPs in a short period of time. Modeling centers provide the resources for the requested experiments with the ESM which they support. They contribute to the decision for which community-driven MIP experiments with the ESM will be conducted, e.g., through granting computational resources and prioritizing experiments to be completed. Additional decisions for the experiments are made by the scientists interested in the MIP. There is some room to make their own choices since not all experimental settings are explicitly defined by the MIP's experiment protocols, e.g., they may use a

coarser spatial resolution and to some degree less model complexity to reduce the computational burden.

There are inevitable tradeoffs in the final experimental designs for individual MIPs. Such choices can be categorized along the three axes of (1) *model complexity* addressing how many process interactions ESMs allow or how much fidelity processes have, (2) *model resolution* referring to the grid spacing of the model, and (3) *simulation length* covering the length and number of simulations in an ensemble of different experimental setups per ESM. These axes, schematically depicted in Figure 4, span a

295 triangle in the complexity - resolution - length space. The volume of the tetrahedron between the origin and the marked triangle indicates the computational need for the experiments. The computational need scales non-linearly. Doubling the simulation length or number of simulations doubles the required computational resources that are needed along these axes, but this is not true for the model resolution and complexity. Increasing the model resolution by a factor of two, for instance, requires computational resources that are an order of magnitude larger. To account for the non-linearity in the computational need, the

300 volume of the tetrahedron would be calculated on scaled values, i.e., an experiment with twice as fine resolution would be marked four times further away from the origin on the resolution axis. The maximum volume of the tetrahedron is limited by the computation capacity abyss, i.e., the available computing capacity at the modeling center.

Although computing power continues to grow, tradeoffs along the three axes of experimental design and prioritizations will continue to be necessary. This is for instance the case in light of the computational cost of interactive chemistry against

the resolution and the number of simulations. All model experiments, irrespective of whether the models have interactive chemistry, compete for priority at modeling centers due to limited computing resources. Experiments with complex ESMs are necessary to understand interactions of chemical species in concert with climate change, for example, the carbon cycle or atmospheric composition-climate interactions. To that end, ESM experiments are performed that have interactive aerosol and chemistry schemes in addition to the fully coupled atmosphere-ocean-land system, making these models complex and resource heavy. For instance, an ESM could simulate changes in vegetation cover due to increased greenhouse gases that in turn have an impact on dust-aerosol emissions in addition to potential changes in soil moisture and winds. In less complex models, the vegetation cover is for instance prescribed such that the number of interactive physical processes is smaller. The computational demand of complex ESMs for simulating many processes limits the attribution of computing resources along the other two axes of experimental design: performing a large number of experiments, which allows the impact of model-internal variability to be reduced; and choosing a fine enough spatial resolution, which explicitly resolves more physical processes on the model grid. For some research questions, the complexity of ESMs can be reduced to a certain degree. For instance, concentrations of well-mixed greenhouse gases can be prescribed instead of being simulated from emissions, if one is interested in computing the forcing and response to a given change in the atmospheric composition (Figure 2). It makes creating large ensembles of ESM experiments possible that are needed to split for instance the imbalance in the radiation budget at the top of the atmosphere into a mean radiative forcing and contributions from internal variability. Similarly, a separation of the response in temperature or air quality into a forced signal and a contribution from internal variability is possible. The required ensemble size and length of experiments for sufficiently reducing the influence of model-internal variability on the global mean radiative forcing (e.g. Forster et al., 2016; Fiedler et al., 2017), climate responses (e.g. Maher et al., 2019; Deser et al., 2020), and impacts on air quality (e.g. Garcia-Menendez et al., 2017; Fiore et al., 2022) depends on the magnitude of the forced signal against the magnitude of the internal variability.

The necessary number of simulated years for separating the signal from internal variability depends on the scientific question. The signal-to-variability ratio is for instance sufficiently good for the response of the global mean of precipitation (Myhre et al., 2018; Allen et al., 2020) and the ERF in the global mean for most climate forcers in the current experiments. Specifically, the suggestion from Forster et al. (2016) for performing 30 years of model experiments with the same boundary data proved useful to diagnose global ERF in most time-slice experiments, except for land-use changes (Smith et al., 2020a). We learned that the exact precision of ERF depends on the model's internal variability inducing year-to-year perturbations in the radiation budget (Fiedler et al., 2019, 2023). Longer simulations of 45 years are needed to diagnose the forcing of some longer-lived trace gases due to the time scale for gas transport through the stratosphere via the Brewer-Dobson circulation (O'Connor et al., 2021). For regional radiative effects, the 30 and 45-year-long simulations are not sufficiently long to obtain a statistical significance for all anthropogenic perturbations in all regions. In UKESM1, the anthropogenic aerosol radiative effects are for instance statistically significant at the 95% level over about 50% of the globe, but the effects are only statistically significant for 10% of the globe for land use and non-methane ozone precursors (O'Connor et al., 2021). Similarly, regional aerosol forcing is not statistically significant over all world regions in models contributing to RFMIP (Fiedler et al., 2019, 2023). For simulated climate responses, the ensemble sizes and simulation lengths of the experiments were not sufficient for addressing all research

questions of interest in the three MIPs, especially for regional responses. Quantifying the regional response of climate to forcing requires larger ensembles of simulations, which the Regional Aerosol MIP (RAMIP, Wilcox et al., 2023) is currently addressing through requesting larger ensembles of experiments with regional perturbations of aerosols than available from AerChemMIP.

Complex models simulating many processes and their interactions are desirable and needed for specific research questions,
and also pose challenges for reducing model-based uncertainty in the assessment of the climate response to various forcings. Model diversity in terms of, for instance, the combination of parameterizations, intricacy and fidelity of represented processes, choice of coupling of model components, choice of the dynamical core, and the resolution is desirable. Model simulations ideally converge to similar solutions for a given question, e.g., how the Earth's temperature responds to anthropogenic perturbations. The diversity in model results should therefore reduce over time to gain confidence in our conclusions drawn from
simulated responses to imposed perturbations.

There are two challenges to reducing model-based uncertainty that can be emphasized in the context of MIPs. One challenge concerns the diversity in the level of complexity included in the ESMs, which is for instance due to choices made for the interacting processes, the representation of chemistry and aerosols, as well as the specification of the spatial resolution by the modeling centers. As an example, this diversity is clearly evident in the complexity of aerosol processes with some CMIP6
models simulating the evolution of different aerosol species and their interactions (e.g. Mulcahy et al., 2018), while other models prescribe spatial distributions of aerosol optical properties (e.g. Mauritsen et al., 2019). Such differences in model capabilities have implications for understanding the reasons for differences in their results (e.g. Wilcox et al., 2013).

The second challenge comes from the consideration of model diversity in the level of complexity inherent in the process of designing a MIP protocol since for instance, a few models can simulate processes that most others cannot. Again, MIPs
already have a specific class of models in mind. For AerChemMIP, emission-driven models were targeted, whereas RFMIP also included contributions from models with less complex representations of aerosols, e.g., those using prescribed aerosol optical properties. Hence, RFMIP received more output from model experiments than for instance AerChemMIP. RFMIP and AerChemMIP were endorsed by CMIP6 and had different structural organizations while PDRMIP started earlier and was in comparison more self-organized and flexible in the MIP life cycle. PDRMIP, therefore, comprises an ensemble of
models of different complexity. Specifically, some of the models in PDRMIP performed experiments with prescribed emissions whereas others used concentrations resulting in an ensemble of experiments partially driven by emissions and partially driven by concentrations of climate forcers. Yet, MIP experimental protocols do not prescribe the level of process complexity in and the resolution of ESMs. This freedom is well justified since ESMs might otherwise not be able to participate in a MIP if they can not fulfill stricter requirements. A wider participation of ESMs in MIPs ensures a sufficiently large multi-model ensemble
needed to robustly quantify forcings and climate responses considering structural model differences. A full exploration of the role of climate-composition feedbacks with focus on biogeochemical processes, however, remains an outstanding challenge due to this difficulty.

## 3.2    Process Understanding Abyss

Although varying model complexity can be a difficulty in understanding differences between model results in a MIP, varying complexity helps in advancing our understanding of climate change. Model simulations with different complexity for instance help in quantifying contributions from feedback mechanisms to climate responses. Additional model components and representations of processes have been incorporated in Earth system models over time in addition to improvements of previously existing physical parameterization schemes and boundary data. Such model developments allowed new insights into the role of processes including feedback mechanisms for climate change, although the overall progress is possibly not as rapid as one would hope. For example, correctly representing clouds and circulation are outstanding challenges that are yet to be resolved.

Multi-model inter-comparisons shed light on where the physical understanding is still limited based on the current representation of processes and where we have accomplished a satisfying advancement in our scientific understanding from such model experiments. An open and unrestricted inclusion of models by key performance indicators allows broad participation of suitable ESMs in MIPs. Scientists can choose which models they include by assessing their fitness-for-purpose.

The results of MIPs alone cannot fully characterize the uncertainty. This is what we call the process understanding abyss (Figure 4), which limits our ability to advance the field with our available models. Other evidence should be considered in parallel or in synergy with MIPs to gain new knowledge - may it be observational data from different sources or completely different models that are not suitable for participation in MIPs - as has been done for assessing the equilibrium climate sensitivity (Forster et al., 2021).

Constraining ESMs with observations is key to advancing our understanding. Although many observations and reanalysis data are already well used, more could be done in the future. Specifically, instead of comparing to single observational or reanalysis datasets, using multiple observational data sources would allow us to first quantify the observational uncertainty against which model results can be better evaluated, e.g., a good performance might mean that model results fall within the observational uncertainty. Moreover, new combined observational products could help to evaluate model output, which may include translating observables into modeled variables. In the past, approaches have been used to translate simulated data into satellite-observable space (e.g., COSP, Bodas-Salcedo et al., 2011).

Machine learning seems promising to develop new and easy ways for exploiting and combining observational data suitable for comparison to model output, e.g., artificial intelligence has been used for filling observational gaps (Kadow et al., 2020). Such ideas could be explored more to unfold the new potential to evaluate and constrain model results in the future in ways we have not done in the past. Future work could also expand on the use of emergent constraints for responses including feedback mechanisms (Hall et al., 2019; Williamson et al., 2021). For example, an emergent constraint approach was used to address the present-day forcing of halocarbons leading to a reduced spread in the forcing estimate (Morgenstern et al., 2020). Another example is using hemispheric differences in albedo to constrain anthropogenic aerosol forcing (McCoy et al., 2020).

There are some limits to advancing climate science with today's complex ESMs since we miss or do not represent some processes that are thought to be relevant to reproducing observed and projected future climate change. This process understanding abyss additionally restricts what can be simulated with even the most comprehensive ESMs (Figure 4). Known gaps from our

community are listed in Table 4. Some chemical reactions and species, as well as their interactions, are currently not represented or differently represented across ESMs, such that their relevance for the climate is difficult to assess. Indeed, including previously missing interactive sources of chemical species in an ESM has the potential for surprising results in estimates of
forcing (e.g., Morgenstern et al., 2020). There was diversity in the representation of nitrate aerosols in the ESMs in CMIP6 (Turnock et al., 2020). Five CMIP6 models included climate-dependent emissions of Biogenic volatile organic compounds (VOCs) from vegetation. Namely, CESM2-WACCM, GFDL-ESM4, GISS-E2-1-G, NorESM2-LM and UKESM1-0-LL yield relatively large increases in BVOC emissions with warming and in turn, large increases in secondary organic aerosols and associated particulate matter (PM) which other models do not simulate (Gomez et al., 2023). Marine primary organic aerosols
are represented by some ESMs (e.g., Burrows et al., 2022a), but marine VOCs other than dimethyl sulfide (DMS) are not. Also, natural primary biological aerosol particles (PBAPs), such as bacteria, pollen, fungi, and viruses (Szopa et al., 2021), are not simulated by ESMs, although PBAP emissions might increase with future warming (Zhang and Steiner, 2022) with potential health impacts. Moreover, both DMS and PBAPs are thought to aid in cloud formation; the effects of such ice-nucleating aerosols on clouds is an area where more progress is needed (Burrows et al., 2022b).

AerChemMIP played a role in the quantification of non-$CO_2$ biogeochemical feedbacks (Thornhill et al., 2021a), illustrated in Figure 5. Almost all non-$CO_2$ biogeochemical feedbacks are negative and therefore counteract warming. The only exception is the positive feedback from methane wetland emissions that amplifies warming and is the largest in magnitude compared to the other non-$CO_2$ biogeochemical feedbacks. The positive feedback from wetland emissions may be partly offset by the negative feedback of the methane lifetime. Together with the large model-dependent feedback for biogenic VOCs, the multi-
model mean feedback is negative, but the uncertain methane feedback gives rise to the large spread in the total non-$CO_2$ biogeochemical feedbacks ranging from positive to negative. Climate change induced feedbacks associated with methane can be better characterized with ESMs that include an interactive representation of the global methane cycle allowing for simulations to be driven by methane emissions (e.g., Folberth et al., 2022). ESMs do currently not simulate effects on methane concentrations. Hence, there is a need to develop methane emissions-driven ESMs.

Not all potentially relevant chemistry-climate feedbacks involving natural climate forcers are yet incorporated or well simulated, e.g., climate-induced changes in fire activity and dust-aerosol emissions. Although some CMIP6 models represented fire dynamics, they did not fully include the interaction with atmospheric chemistry (e.g., Teixeira et al., 2021). And of those feedbacks that are simulated, erroneous model consensus or small magnitudes for feedbacks might lead to a misleading perception that these feedbacks are not important. The dust trend over the historical period is one such example. The CMIP6 models show
trends of different signs and magnitudes for desert-dust aerosols over the historical time period (Bauer et al., 2020; Thornhill et al., 2021a), and there is no ESM in CMIP5 or CMIP6 that reproduces the magnitude of the reconstructed dust increase from the pre-industrial to the present-day (Kok et al., 2023). This points towards an insufficient process-based understanding of dust-aerosol changes with warming, which has implications for the understanding and quantification of the radiation imbalance. Modeling surface conditions is a challenge and a potential source of the diversity in simulated dust trends. Not all
models participating in CMIP6 have the capability to simulate interactive vegetation dynamics but some do, e.g., UKESM1 and GFDL-ESM4. A lack of coupled vegetation dynamics is not the only potential reason for differences in dust and other

aerosols. Winds control the emission and transport of desert-dust aerosols and the soil erodibility is influenced by the available moisture from rain events. There is a large diversity in model-simulated regional changes in winds and precipitation in response to warming which in turn introduces uncertainty in simulated dust trends.

Of those processes that are simulated, a large driver in model diversity for atmospheric composition is thought to stem from the representation of natural processes (e.g., Séférian et al., 2020; Zhao et al., 2022). Circulation is a grand challenge for ESMs (Bony et al., 2015), affecting the spatiotemporal distribution of aerosols. Again desert-dust aerosols are, for instance, emitted and transported by winds, with a persistently large diversity across ESMs (e.g. Evan et al., 2014; Checa-Garcia et al., 2021; Zhao et al., 2022; Kok et al., 2023). The ability to accurately simulate atmospheric circulation is also relevant to the challenge

of realistically simulating clouds and rainfall, including their regional trends due to atmospheric composition changes, (e.g. Sperber et al., 2013; Stevens and Bony, 2013; Fiedler et al., 2020; Wilcox et al., 2020). The simulated clouds influence how aerosols can affect them and rainfall determines when and where aerosols are removed from the atmosphere. Another example of the crucial role of representing natural processes is the ability of ESMs to simulate aerosols in the Arctic. In particular, a better understanding of natural aerosols in the rapidly warming Arctic may be a key factor in resolving the puzzle of Arctic

amplification (Schmale et al., 2021), where diversity across ESMs for NTCFs is large (Whaley et al., 2022).

There are a number of challenges in better understanding historical trends in aerosol species and their precursors from different natural and anthropogenic sources. A further improved knowledge would help to unravel model diversity in the evolution of aerosol forcing over time, and how it is related to time-dependent temperature biases in CMIP6 models (Flynn and Mauritsen, 2020; Smith and Forster, 2021b; Smith et al., 2021a; Zhang et al., 2021). ESMs simulate, for instance, different

historical trends for $O_3$ and aerosols (Mortier et al., 2020; Griffiths et al., 2021). Even for present-day conditions, outstanding challenges for simulating aerosols persist, e.g., for the concentrations of secondary organic aerosols (Turnock et al., 2020), which have natural and anthropogenic origins (Fan et al., 2022). Moreover, aerosol optical properties are partially biased (e.g., Brown et al., 2021), the size distributions of different aerosol species are not sufficiently understood (Mahowald et al., 2014; Croft et al., 2021), and inter-model differences in aerosol optical depth persist across different phases of CMIP and AeroCom

(Wilcox et al., 2013; Vogel et al., 2022).

## 4    Opportunities

There are several opportunities to advance the understanding of climate responses to perturbations in emissions, atmospheric composition, and/or the land surface. These are opportunities to augment traditional ESM experiments through (1) the use of emulators where they are informative, i.e., where a climate response to a perturbation is expected to fall within the solution

space of existing ensembles of ESM experiments, (2) the use of novel global kilometer-scale experiments where they are possible in light of the tradeoffs along the complexity - resolution - length axes, (3) the development and application of machine learning across the paradigm to speed up and improve processes in complex ESMs where it is needed, and finally (4) new process-based evaluation and analysis methodologies that leverage multiple observational datasets to constrain models. Moreover, there is an opportunity to further improve radiative forcing calculations, and diagnostic requests for ESM experiments

to allow more in-depth scientific analyses with potential synergies with impact assessments. Opportunities arising from novel capabilities and diagnostics are listed in Table 5–6 and elaborated on in the following sections.

## 4.1 Augmented ESMs

### 4.1.1 Machine learning where useful

New opportunities arise from machine learning approaches. These can for instance contribute to improving or speeding up
process representations in ESMs, as well as designing smart tools for post-processing and evaluating ESM output. We see primarily four areas where machine learning could help in advancing the research in our community. These are (1) to include faster and more precise representations of processes in models, e.g., for replacing or modifying physical parameterizations that are thought to not work sufficiently well in all conditions in which they are needed, (2) to develop novel ways to gain a better understanding of physical and chemical interactions, e.g., through data mining employing machine learning techniques, (3) to
fill observational gaps, e.g., in satellite products to allow the creation of spatially complete data to more easily validate model results against observational information, and (4) to mimic climate responses to radiative forcing, e.g., to prioritize experiments for the design of new MIP protocols.

Proofs of the concept of applying machine learning in our research field exist. One example is using deep learning for the design of new parameterizations (e.g., Rasp et al., 2018; Eyring et al., 2021; Veerman et al., 2021). Atmospheric chemistry
parameterizations can, for instance, be replaced by fast representations based on machine learning (Keller and Evans, 2019; Shen et al., 2022). The causes of multi-model diversity highlighted in previous studies (Young et al., 2018; Mortier et al., 2020; Griffiths et al., 2021) can also be elucidated using machine learning. There is an increase in the availability of globally gridded fused model-observation data products (e.g., Randles et al., 2017; Buchard et al., 2017; Inness et al., 2019; Betancourt et al., 2021; van Donkelaar et al., 2021; Betancourt et al., 2022) that can be used as benchmarks in model evaluation of atmospheric
composition. Novel aspects of such benchmarks include providing data relevant to health impacts (e.g., DeLang et al., 2021) and using machine learning techniques for global mapping of atmospheric composition (e.g., Betancourt et al., 2022). Liu et al. (2022) used deep learning to quantify the sensitivity of surface $O_3$ biases to different input variables in a CMIP6 model (UKESM1), thereby providing a new understanding of biases and enabling future projections of bias-corrected surface $O_3$. Similarly, such approaches have been used to improve our understanding of model diversity in other aspects of atmospheric
composition, e.g., surface PM (Anderson et al., 2022). Including necessary variables for such algorithms in the model output of future MIPs can enable a multi-model intercomparison of different contributions to model biases and provide bias-corrected data for future projections of changes that can be tailored toward impact studies, e.g., concerning future air quality and human health.

Emulators (e.g., Meinshausen et al., 2011; Leach et al., 2021), a class of models that mimics the behavior of an ESM, can help
to prioritize new ESM experiments. Emulators are trained on output from existing experiments with ESMs, of which there are now many, e.g., from the CMIP6-endorsed MIPs and several CMIP phases. Unlike ESMs, emulators perform fast calculations instead of numerical integration of non-linear physical and chemical equations over time on a three-dimensional grid. Both

techniques allow spatially resolved predictions of temperature and other variables, but emulators can do it at massively reduced computational costs compared to ESMs (Beusch et al., 2020; Watson-Parris et al., 2021, 2022). Once established, emulators can be used to explore the climate response to radiative forcing, e.g., to inform experimental designs of future emission scenarios in CMIP6 (O'Neill et al., 2016). In terms of physically-based emulators of the climate system (i.e. simple climate models), RFMIP and AerChemMIP experiments were invaluable to determine aerosol ERF, ozone ERF and the factors influencing methane chemical lifetime. Some of these relationships were developed in the lead-up to IPCC-AR6 WG1 and used directly in the report (e.g., Smith et al., 2021a; Thornhill et al., 2021a, b).

Training emulators requires a broad range of ESM experiments such that they interpolate rather than extrapolate into unseen climate conditions. This training data could be made up of CMIP experiments, an ensemble of idealized experiments (Westervelt et al., 2020), or perturbed parameter ensembles where several ESM experiments with systematically different settings in parameterizations are performed to study sources of model-internal uncertainties (e.g., Johnson et al., 2018; Regayre et al., 2018; Wild et al., 2020). Emulators have been used for some time (Murphy et al., 2004; Lee et al., 2013, 2016; Yoshioka et al., 2019; Johnson et al., 2020; Watson-Parris et al., 2020; Wild et al., 2020) and modern techniques also utilize machine learning to allow validation against observations (Watson-Parris et al., 2021). Emulators can incorporate model spreads similar to the output from classical MIPs with ESMs. A review of emulation techniques that are routed in statistical mechanics highlights the potential to further improve emulators for use in climate sciences by using machine learning (Sudakow et al., 2022). The difficulty of accounting for non-parametric biases of CMIP models in emulators however remains (Jackson et al., 2022). Nevertheless, emulators have already been proven useful to sample parametric differences and to study climate change (e.g., Tebaldi and Knutti, 2018).

### 4.1.2 Kilometer-scale experiments where possible

Much finer spatial resolutions with horizontal grid spacings of a few kilometers hold the potential to overcome some of the long-standing challenges concerning the representation of clouds, precipitation, and circulation in global climate simulations, which would require a step change in collaboration between climate science and high-performance computing (Slingo et al., 2022). Representing clouds and circulation correctly in coarse resolution models of several tens to hundreds of kilometers of grid spacings is an outstanding challenge (e.g., Bony et al., 2015). High spatial resolution naturally improves the representation of clouds and precipitation, at least in part, due to better resolved orographic effects on atmospheric dynamics and the explicit simulation of convective cloud systems along with the mesoscale circulation (Oouchi et al., 2009; Berckmans et al., 2013; Heinold et al., 2013; Klocke et al., 2017; Satoh et al., 2019; Hohenegger et al., 2020), although not all model biases are eliminated (Caldwell et al., 2021). These processes are tightly connected to atmospheric composition changes and associated effects on the atmosphere including feedback mechanisms. Furthermore, the coupling of atmospheric processes with the land improves in kilometer-scale experiments. It can reduce biases in the simulated temperature and precipitation (Barlage et al., 2021), which can help to better understand regional climate change that involves land-mediated feedbacks. Moreover, better-resolved ocean dynamics hold the potential for surprises in understanding climate responses, e.g., with respect to future projections of temperatures and rare high-impact events (Hewitt et al., 2022).

Kilometer-scale experiments, therefore, allow new insights into processes in the Earth system following the perturbation-response paradigm and can leverage the experiences made with regional kilometer-scale climate experiments for different world regions (e.g., Prein et al., 2015; Liu et al., 2017; Kendon et al., 2019). Kilometer-scale experiments are presently only possible for climate studies on limited area domains or globally for restricted time periods of a few weeks to years (Hohenegger et al., 2023). Given the role of resolution in maintaining concentrated emissions, non-linearities in chemistry, and fronts in the atmospheric transport of pollutants, more kilometer-scale climate change experiments might prove valuable to advance the understanding of climate and air quality interactions. Such experiments within limited area domains would also help to alleviate the computational cost of both high process complexity and high spatial resolution.

Global coupled atmosphere-ocean simulations with a few kilometers resolution can be done and progress has been made in incorporating some representation of atmospheric composition for example the carbon cycle (Hohenegger et al., 2023). For some questions on atmospheric composition and the associated air quality and climate response, kilometer-scale experiments are already used, e.g., for a better understanding of aerosol-cloud interactions (Simpkins, 2018; McCoy et al., 2018), which is one of the key uncertainties in ERF from ESMs (e.g., Smith et al., 2020a). Another question that can be better addressed with kilometer-scale experiments is the resolution dependence of radiative forcing and feedbacks, especially for those that involve clouds that are a key uncertainty in our understanding of climate change with ESMs (Stevens and Bony, 2013). Another question is to what extent more resolved meteorological processes aid in improving the representation of atmospheric composition and air quality, e.g., concerning health impacts in urban areas.

The community of the three MIPs will not be able to mainly rely on global kilometer-scale model experiments in CMIP7, especially in the context of a MIP since fully coupled ESMs with interactive aerosols and chemistry at a resolution of 1 km fast enough to perform multi-decadal simulations are unlikely to be ready in the time of CMIP7. In light of this restriction, we see two main routes forward for immediately using spatially refined information in our next MIPs. The first possible way is to use the output from global kilometer-scale experiments that are run for other purposes to drive offline models for aerosols and chemistry or atmospheric radiative transfer calculations. This approach is suitable to answer some but not all research questions in our community. For instance, the response of dust emission fluxes to changes in winds and moisture can be addressed with offline modeling and allows to identify underlying reasons for changes and model differences in the dust response (Fiedler et al., 2016), but the implication of such dust emission changes for air quality and climate responses can not be quantified with such an approach. For the latter, regional one- or two-way dynamical downscaling experiments could be used. We perceive dynamical downscaling as the second main avenue for our near-future work to obtain regionally refined spatial information. Regional climate modeling is already well developed and organized via CORDEX with a focus on providing regional climate change information. Regional climate models with the capability to perform experiments with coupled aerosols and chemistry exist for instance in Europe and the US (e.g., Pietikäinen et al., 2012; Schwantes et al., 2022), but have not been used in our past MIPs. For CMIP7, UKESM2 and CESM aim also to have regional model configurations. At least two different regional composition-climate models therefore could exist and be used in future MIPs. The regional models will nevertheless need output from global ESM experiments with coupled aerosols and chemistry as boundary data for performing the regional experiments. As such a need for experiments with classical global ESMs is retained, at least for CMIP7, although moving

towards global kilometer-scale modeling with a sufficient coupling of physical processes to aerosols and chemistry to address the community's research interests will be a goal to aspire to.

## 4.2 Improved diagnostics and analyses

### 580 4.2.1 Radiative Forcing Calculation

The concept of radiative forcing is central to the perturbation-response paradigm for understanding climate change (e.g., Sherwood et al., 2015), in which radiative forcing is eventually balanced by a temperature response mediated by feedback processes. Definitions of radiative forcing have evolved over time to allow an increasingly wide range of different climate forcers or contexts for perturbations to be considered interchangeably (Ramaswamy et al., 2019), schematically depicted in 585 Figure 3. Early definitions of radiative forcing focused on changes in the net radiation at the tropopause, ideally after the stratosphere had adjusted to a new radiative equilibrium in the presence of the forcing agent (stratospherically adjusted radiative forcing, Hansen et al., 1997). These definitions have been generalized in the concept of ERF (ERF, see Sherwood et al., 2015).

Quantifying IRF for ESMs is desirable, even if the IRF of the forcing agent is constrained by other methods. There is little fundamental uncertainty for IRF of $CO_2$ changes, as indicated by errors on the order of a fraction of a percent from the 590 most accurate line-by-line radiative transfer models (Pincus et al., 2020). However, due to the high computational demand, ESMs do not compute the radiative transfer with a line-by-line model. Instead, they rely on parameterizations, speeding up the computation at the expense of accuracy. Consequently, a model spread in IRF occurs despite so little fundamental uncertainty. For instance, a spread in $CO_2$ IRF has persisted across CMIP phases and accounts for a majority of the model spread in the $CO_2$ ERF (Chung and Soden, 2015; Soden et al., 2018; Kramer et al., 2019; Smith et al., 2020a). Quantifying the model's 595 IRF, e.g., with double calls of the radiative transfer calculations (Section 2.2), is particularly relevant in light of the model-state dependence of IRF (Stier et al., 2013; Huang et al., 2016), referring to ESM differences in atmospheric conditions that affect the radiative transfer. Both CMIP6 models (He et al., 2022) and theoretical arguments (Jeevanjee et al., 2021) suggest that $CO_2$ IRF is correlated with temperature, i.e., $CO_2$ IRF increases as the surface warms and the stratosphere cools. This feedback-like effect on radiative forcing is thought to account for a $\sim$10% increase in $CO_2$ IRF for present-day against pre-industrial (He 600 et al., 2022) and $\sim$30% for quadrupled $CO_2$ (Smith et al., 2020b). It requires clarity in the experimental design and reporting of resulting ERF estimates to disentangle the contributions of forcing from feedbacks in future experiments.

There are several methods to quantify ERF across ESMs that can be further improved and standardized (Figure 3). As mentioned earlier (Section 2.2), ERF is often computed from model experiments using prescribed sea-surface temperatures and sea ice (fixed-SST method; Forster et al., 2016) and has been adopted in RFMIP (Figure 6). RFMIP requested 30-year- 605 long experiments for ERF calculations (Pincus et al., 2016) following recommendations based on CMIP5 output (Forster et al., 2016). That experiment length proved to be sufficient for ERF estimates of most climate forcers in RFMIP, e.g., for ERF of anthropogenic aerosols although more simulated decades further improve the precision of the ERF calculation (Fiedler et al., 2017, 2019). Differently from RFMIP, AerChemMIP found that a spin-up time associated with long-lived trace gases, e.g. halocarbons, is necessary before calculating the ERF. This meant that the approach of 30-year-long time slice experiments was

not entirely appropriate for the AerChemMIP experiments for all individual climate forcers (Section 3.1). The longer spin-up period should be accounted for in future requests for new experiments for ERF calculations of such climate forcers.

The fixed-SST method has two advantages compared to the traditional approach based on coupled atmosphere-ocean experiments (Gregory et al., 2004). Firstly, the impact of internal variability on ERF estimates is reduced through sufficiently long experiments with prescribed sea-surface conditions (Forster et al., 2016; Fiedler et al., 2017) that are computationally less expensive. Secondly, the use of fully coupled experiments to estimate ERF relies on a linear relationship between ERF and temperature, now known not to be true in general (Armour, 2017; Rugenstein et al., 2020; Smith and Forster, 2021b), and can lead to ERF estimates that differ from the results of fixed-SST experiments (Forster et al., 2016).

One weakness of fixed-SST experiments to estimate ERF is the adjustment of land-surface temperatures. A change in land-surface temperatures affects the energy budget, leading to biased estimates of ERF at the order of 10% (Smith et al., 2020a). Such unwanted influences on the ERF estimate can be post-corrected (Tang et al., 2019; Smith et al., 2020a). If the capability of fixed land-surface temperatures (Andrews et al., 2021) was facilitated in more ESMs, biases in ERF arising from surface temperature adjustments would be virtually eliminated in the future. If adopting the fixed sea- and land-surface temperature method (Figure 3) in a MIP becomes feasible, the change in the radiation budget would then be equal to the change in the energy budget of the system, which overcomes the limitations of other methods for estimating ERF. Prescribing sea- and land-surface temperature is different from the experiments carried out for CMIP6 and RFMIP. The requested experiments used prescribed sea-surface temperatures and sea ice following the experimental design of the Atmosphere Model Intercomparison Project (AMIP, Gates, 1992), but the land-surface temperatures were freely evolving. Prescribing the sea ice, sea- and land-surface temperatures has not been done in a MIP to date.

The radiative forcing of anthropogenic aerosols depends on the optical properties and the effects on clouds. Improved diagnostics and observational constraints in the output analysis for aerosol burden and optical properties would be useful for better understanding the model diversity in the associated radiative forcing and the climate response. As discussed in Section 3, the significant diversity across ESMs in the simulated distributions of aerosol burden, optical properties, radiative effects, and the resulting climate responses, including temperature and precipitation, limits building confidence in model projections of climate change. Analysis of relevant and correlated model diagnostics together with observational constraints can shed light on the source of diversity in the full cause-and-effect chain and inform improvements in the treatment of aerosols in models. For example, Samset (2022) underscores the diversity in aerosol absorption as the dominant cause of model diversity in historical precipitation changes in CMIP6. RFMIP experiments point to overestimated aerosol absorption from anthropogenic black carbon and a relatively small share of natural aerosol absorption which leads to direct radiative effects of anthropogenic aerosols in some CMIP6 models which are implausible in light of other lines of evidence (Fiedler et al., 2023). That multi-model assessment was not as broad as it could have been due to the limited availability of requested output for aerosol properties and diagnostic calls to the radiative transfer scheme for aerosol effects in the CMIP6 models. Wider availability of relevant output from the next phases of RFMIP and AerChemMIP would allow deeper exploration of the inter-model differences in the radiative forcing of anthropogenic aerosols.

### 4.2.2 Synergies with Impact Assessments

There is the opportunity to increase synergies with impact assessments of climate change through improved model diagnostics. A common tropopause diagnostic, included for the first time in CMIP6 models due to AerChemMIP, was available for the calculation of tropospheric $O_3$ burden in a consistent manner. However, the tropopause height was found to vary across the models, and as $O_3$ is found in large concentrations in the stratosphere and upper troposphere, the tropopause definition contributed to the model spread in the calculated tropospheric $O_3$ burden. Through TriMIP, it was identified that a tropopause defined by the

$O_3$ mixing ratio results in a smaller model spread in tropospheric $O_3$, which is relevant for air quality assessments.

Moreover, not all ESMs currently output diagnostics for PM, and those models that calculate PM use different formulas and combinations of species. Such differences make any intercomparison of PM between models and observations difficult. To circumvent this issue, AerChemMIP tested (Allen et al., 2020; Turnock et al., 2020; Allen et al., 2021) estimating PM from model output following Fiore et al. (2012), but associated uncertainties are hard to quantify. Future MIPs could standardize

calculations for $PM_{2.5}$ and $PM_{10}$ across experiments, e.g., consistent with air quality assessments following the standards of the World Health Organization. It would allow the use of PM measurements for air quality monitoring as an independent validation data set and could create a bridge to health impact studies.

Another opportunity to connect more with impact-oriented research can arise from ESM experiments for additional future socio-economic and mitigation-based pathways such that uncertainty in emission developments, including mitigation and asso-

660 ciated impacts of atmospheric composition changes, can be systematically explored. In addition to new phases of AerChemMIP and RFMIP, examples are a MIP on future methane removal (Jackson et al., 2021) in support of potential climate solutions or on fire emission developments possibly accounting for the new capability to represent fire feedbacks (Teixeira et al., 2021) and leveraging on experiences made in the Fire Model Intercomparison Project (FireMIP, Rabin et al., 2017). If stronger interactions with communities concerned with climate-change impacts would be pursued, e.g., with the Vulnerability, Impacts,

Adaptation and Climate Services (VIACS, Ruane et al., 2016) and the Inter-Sectoral Impact Model Intercomparison Project (ISIMIP, Frieler et al., 2017) community, the usage of output from MIPs for societally relevant problems could be enhanced. Such engagement could lead to a better integrated understanding of links between climate change, extremes, air quality, and the impacts in different sectors, e.g., health, energy, and economics, for climate change preparedness.

### 5 Conclusions

The existence of TriMIP was coincidental, yet the joint community of three MIPs has proven valuable for advancing the research field on atmospheric composition and associated air quality and climate responses. RFMIP helped to establish a consistent practice for diagnosing radiative forcing from CMIP6 models, and having preindustrial experiments across AerChemMIP and RFMIP facilitated the comparability of results. Challenges in advancing the understanding of climate change with Earth System Models (ESMs) following the perturbation-response paradigm remain. For instance, the approach works well for un-

derstanding temperature responses to radiative forcing, but it seems less satisfying for precipitation responses, e.g., due to reduced model consensus on regional changes in precipitation compared to temperature for a given forcing.

In part, the difficulty of simulating precipitation responses is related to the grand challenge of representing clouds and circulation, which can be addressed with newly evolving capabilities. Moreover, model-state dependencies affecting radiative forcing and climate responses can potentially be reduced or even resolved in the near future. Promising ideas are the use of:

- ESM experiments with prescribed land temperatures in addition to prescribed sea ice and sea-surface temperatures in more models to quantify the effective radiative forcing free of artifacts arising from temperature adjustments over land,

- kilometer-scale model experiments with resolutions of a few kilometers to improve the understanding of interactions of atmospheric composition with circulation, clouds, and precipitation which are long-standing challenges in climate modeling with coarse-resolution ESMs affecting for instance the representation of atmospheric composition and associated air quality assessments and aerosol-cloud interactions,

- novel machine learning approaches to speed up and improve parameterizations of sub-gridscale processes for experiments with ESMs and kilometer-scale models, to do data mining for pattern recognition in big model output and to develop augmented observational products for new constraints on model output,

- emulation techniques to mimic climate responses to different forcings within the solution space of existing experiments to reduce the computation burden on modeling centers, and

- sufficiently long experiments or sufficiently many ensemble members for an experiment to better distinguish climate and air quality responses to atmospheric composition changes from internal variability and therefore substantially reduce the risk of ambiguity in attributing responses to anthropogenic perturbations.

The optimal choice for new model experiments along the three axes of resolution, complexity, and length is specific to the research question and can be a challenge for modeling centers, especially when several MIPs are simultaneously requesting new experiments. Computationally efficient model code for new computer architectures and smart experimental designs are therefore important when we move outwards along any of the three axes in experimental design. The former can be addressed by even closer collaborations with computer scientists, which is needed to translate existing model codes for use on exascale machines (e.g., Fuhrer et al., 2018). The latter can benefit from information from computationally fast models that emulate results from existing ESM experiments, of which there are now many, e.g., from several CMIP phases. Exploiting results from emulators can help to prioritize new ESM experiments, e.g., to perform those that are needed for questions for which the answer is not expected in the solution space of existing experiments.

The three MIPs have built an international vibrant community that goes on to tackle new research endeavors. First new MIP proposals emerging from the TriMIP community have already been made. The new Regional Aerosol MIP (RAMIP, Wilcox et al., 2023) will explore the role of regional aerosol changes in near-future climate change, drawing on earlier experimental designs to be directly comparable with CMIP6. The planning of the second phases of RFMIP and AerChemMIP is currently underway as community MIPs for CMIP7 (https://wcrp-cmip.org/model-intercomparison-projects-mips/). Keeping the two MIPs separate has advantages over connecting both initiatives in a single new MIP. The MIP names and the general ideas of

RFMIP and AerChemMIP are already known through CMIP6. Moreover, the science questions and the associated experiment protocols are clearer, and the workload for coordination and management is smaller for the separate MIPs. Nevertheless, multi-purpose experiments can be useful and less burdensome in terms of human and computational resources, as long as they facilitate answering the science questions laid out by the MIPs. We therefore propose enhanced coordination across the MIPs during the experimental protocol design phase potentially aiding in reducing the number of experiments.

The community of these MIPs continues to maintain the exchange across the community and interdisciplinary boundaries under the new Composition Air quality Climate inTeractions Initiative (CACTI, cacti-committee@geomar.de). CACTI has the aim to quantify and better understand the global and regional forcing, the climate and air quality responses, and the Earth system feedbacks due to atmospheric composition and emission changes. Through joint strengths and diverse expertise, CACTI strives to contribute to the advancement of the understanding of anthropogenic climate change by adopting the established school of thought of the perturbation-response paradigm with novel methodological tools.

*Data availability.* Data of RFMIP and AerChemMIP are publicly available via ESGF, and data of PDRMIP via WDCC.

*Author contributions.* All authors contributed to the discussion and writing of the manuscript. V.N., F.M.O'C., C.J.S., and S.F. designed the figures. V.N., F.M.O'C., C.J.S., R.P., T.T., W.J.C., M.S., G.M., and S.F. participated in the organization of TriMIP meetings that contributed to the manuscript content. S.F. led the synthesis and writing.

*Competing interests.* F.M.O'C. declares a competing interest as topical editor of GMD.

*Acknowledgements.* We thank the TriMIP community for the discussions at the TriMIP-athlon-3 workshop held online everywhere in December 2021 and at the hybrid CACTI workshop in June 2023. S.F. was supported by the German Science Foundation (DFG) with projects in the Collaborative Research Centers SFB1502 (grant no. DFG 450058266) and SFB1211 (grant no. DFG 268236062), and DOMOS funded by the European Space Agency. F.M.O'C. was supported by the Met Office Hadley Centre Climate Programme funded by BEIS (GA01101) and the EU Horizon 2020 Research Programme CRESCENDO (grant no. 641816) and ESM2025 (grant agreement number 101003536) projects. C.J.S. was supported by a NERC/IIASA Collaborative Research Fellowship (NE/T009381/1). T.T. was supported by the Environment Research and Technology Development Fund (grant no. JPMEERF21S12010) of the Environmental Restoration and Conservation Agency, Japan, and the Japan Society for the Promotion of Science (JSPS) KAKENHI (grant no. JP19H05669). M.K. and A.V. are supported by the Leverhulme Centre for Wildfires, Environment and Society through the Leverhulme Trust (grant number RC-2018-023). A.V. is also funded by the by the AXA Research Fund (project 'AXA Chair in Wildfires and Climate') and by the Hellenic Foundation for Research and Innovation (Grant ID 3453). A.M. was supported through the funding from the European Research Council Grant agreement 770765. R.J.K. was supported by NOAA Award NA18OAR4310269 and NASA grant no. 8NSSC21K1968. G.M. was supported by the Horizon 2020

project CONSTRAIN (grant no. 820829). D.W.P. acknowledges funding from NERC project NE/S005390/1 (ACRUISE). S.T. would like to acknowledge that support for this work came from the UK-China Research and Innovation Partnership Fund through the Met Office Climate Science for Service Partnership (CSSP) China as part of the Newton Fund. L.J.W. is supported by the National Centre for Atmospheric Science, the Natural Environment Research Council (NERC; grant NE/W004895/1, TerraFIRMA), and the Research Council of Norway (grant no. 324182, CATHY).

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

1345

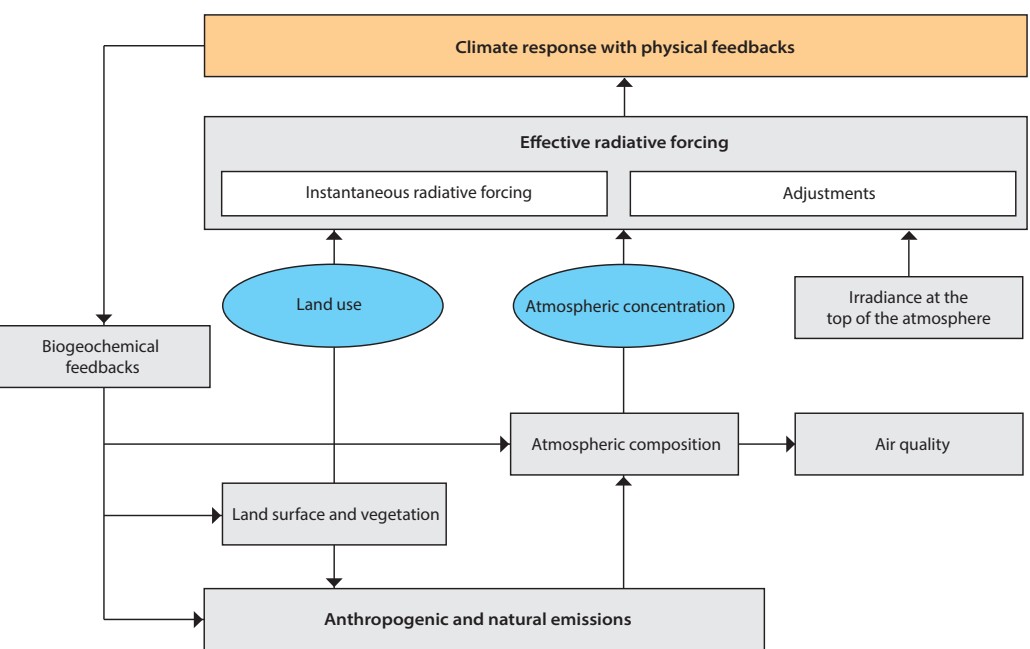

**Figure 1.** Schematic depiction of the perturbation-response paradigm in understanding and quantifying climate changes to perturbations using an Earth System model. Blue circles indicate options for simpler ESMs that prescribe perturbations in concentrations and land use. Climate responses including physical feedbacks marked in orange are simulated with a model configuration coupled to an ocean model.

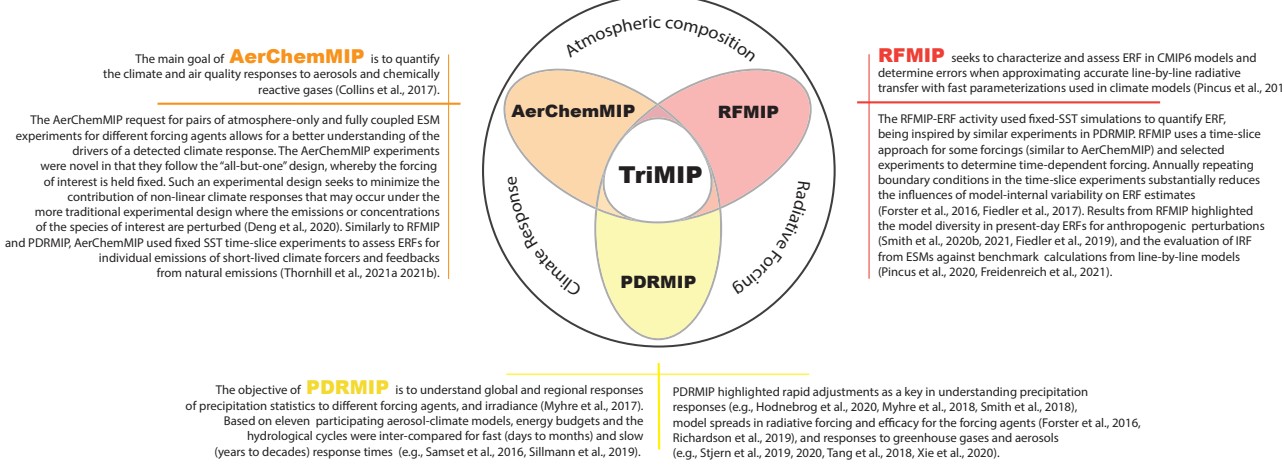

The main goal of **AerChemMIP** is to quantify the climate and air quality responses to aerosols and chemically reactive gases (Collins et al., 2017).

The AerChemMIP request for pairs of atmosphere-only and fully coupled ESM experiments for different forcing agents allows for a better understanding of the drivers of a detected climate response. The AerChemMIP experiments were novel in that they follow the "all-but-one" design, whereby the forcing of interest is held fixed. Such an experimental design seeks to minimize the contribution of non-linear climate responses that may occur under the more traditional experimental design where the emissions or concentrations of the species of interest are perturbed (Deng et al., 2020). Similarly to RFMIP and PDRMIP, AerChemMIP used fixed SST time-slice experiments to assess ERFs for individual emissions of short-lived climate forcers and feedbacks from natural emissions (Thornhill et al., 2021a 2021b).

**RFMIP** seeks to characterize and assess ERF in CMIP6 models and determine errors when approximating accurate line-by-line radiative transfer with fast parameterizations used in climate models (Pincus et al., 2016).

The RFMIP-ERF activity used fixed-SST simulations to quantify ERF, being inspired by similar experiments in PDRMIP. RFMIP uses a time-slice approach for some forcings (similar to AerChemMIP) and selected experiments to determine time-dependent forcing. Annually repeating boundary conditions in the time-slice experiments substantially reduces the influences of model-internal variability on ERF estimates (Forster et al., 2016, Fiedler et al., 2017). Results from RFMIP highlighted the model diversity in present-day ERFs for anthropogenic perturbations (Smith et al., 2020b, 2021, Fiedler et al., 2019), and the evaluation of IRF from ESMs against benchmark calculations from line-by-line models (Pincus et al., 2020, Freidenreich et al., 2021).

The objective of **PDRMIP** is to understand global and regional responses of precipitation statistics to different forcing agents, and irradiance (Myhre et al., 2017). Based on eleven participating aerosol-climate models, energy budgets and the hydrological cycles were inter-compared for fast (days to months) and slow (years to decades) response times (e.g., Samset et al., 2016, Sillmann et al., 2019).

PDRMIP highlighted rapid adjustments as a key in understanding precipitation responses (e.g., Hodnebrog et al., 2020, Myhre et al., 2018, Smith et al., 2018), model spreads in radiative forcing and efficacy for the forcing agents (Forster et al., 2016, Richardson et al., 2019), and responses to greenhouse gases and aerosols (e.g., Stjern et al., 2019, 2020, Tang et al., 2018, Xie et al., 2020).

**Figure 2.** Overview of the MIPs and their topics in TriMIP.

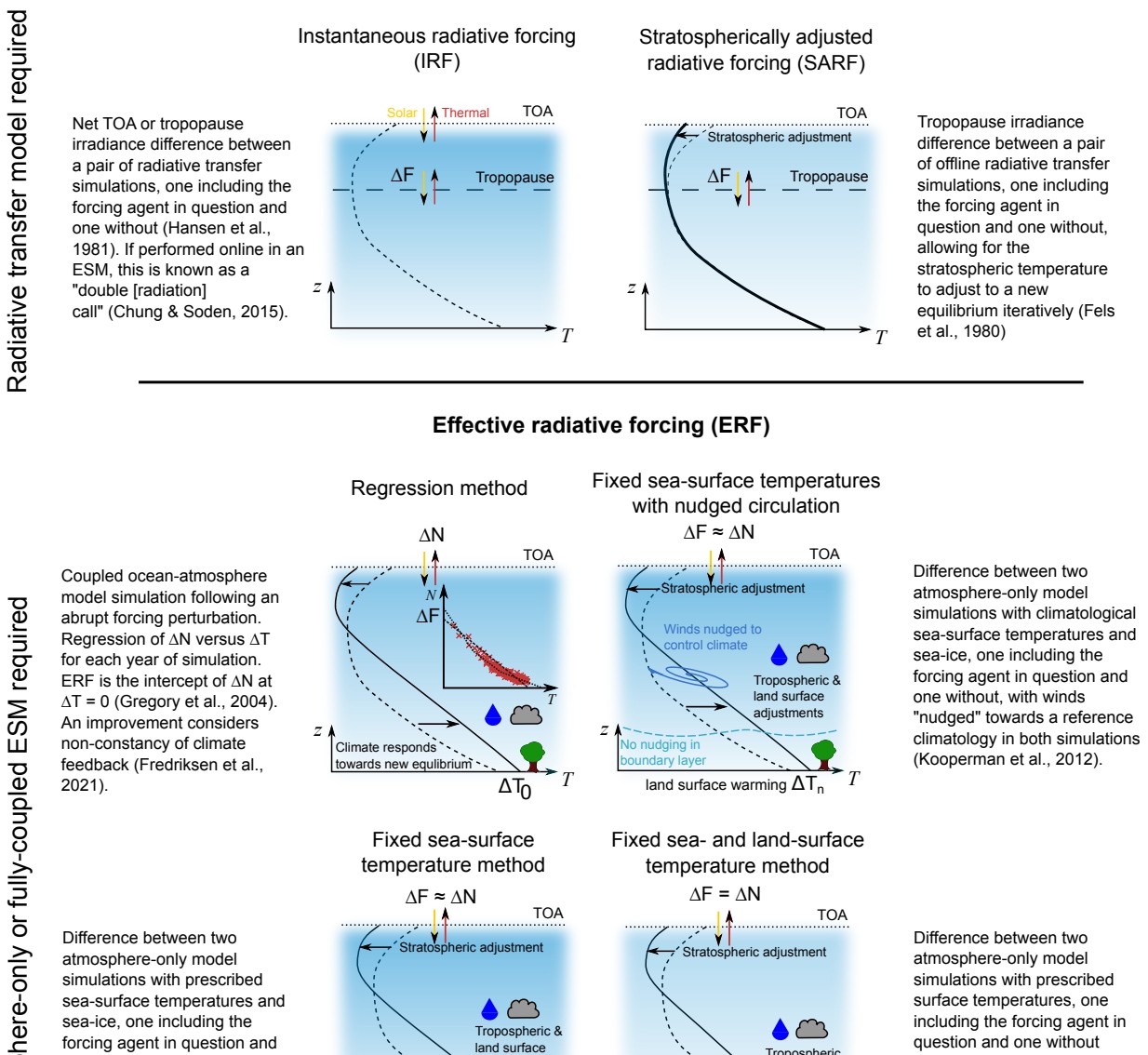

**Figure 3.** Methods for calculating radiative forcing. Shown are graphical depictions of the different methods for calculating radiative forcing based on (top) radiative transfer models and (bottom) general circulation models (GCMs). The latter have substantially developed over time by including more biological, physical and chemical processes, resulting in today's most comprehensive Earth System Models (ESMs). The methods differ in accuracy indicated by the differences between changes in the radiation ($\Delta F$) and energy budgets ($\Delta N$) at the top of the atmosphere (TOA) that arise from method-dependent temperature changes ($\Delta T$). ERF of ESMs in CMIP6 was for instance quantified with the fixed sea-surface temperature method. More accurate results might be obtained with the fixed sea- and land-surface temperature method in future experiments.

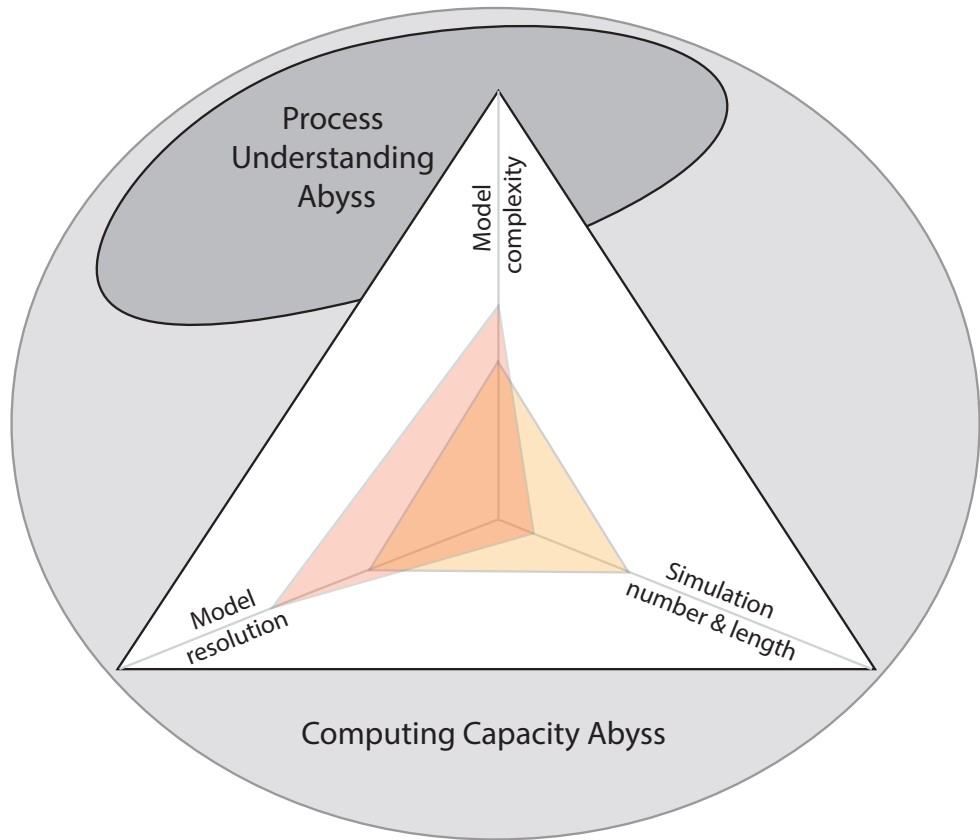

**Figure 4.** Tradeoffs in model configuration. Shown are the aspects of model experiment choices along the three axes: complexity, resolution, as well as number and length. The latter axis includes the ensemble size, referring to the number of members in an experiment ensemble. The triangles mark potential choices along these axes with the volume of the tetrahedron filling the space between the origin and the colored triangle indicating the computational need. The circles mark limits concerning the availability of computing resources (Computing Capacity Abyss) and the understanding of physical, chemical, and biological processes (Process Understanding Abyss).

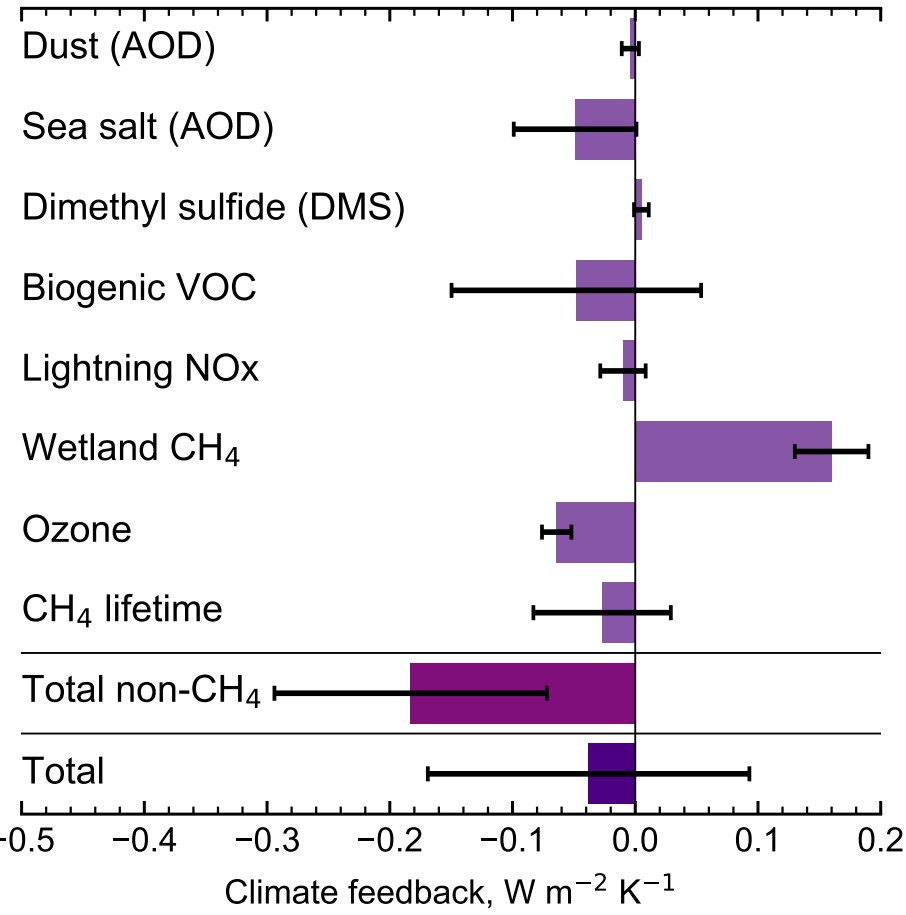

**Figure 5.** Feedback parameter, a measure of change in net energy flux at the top of the atmosphere for a given change in surface temperature, for chemistry and aerosol processes from AerChemMIP experiments (adjusted Figure 5 from Thornhill et al., 2021a, under the Creative Commons Attribution 4.0 International License - CC BY 4.0). Shown are the multi-model means (bars) and the model-to-model standard deviations (lines). Totals are sums of the individual feedbacks. Dust and sea-salt feedbacks are measured by their aerosol optical depth. Details on the calculation are given in Thornhill et al. (2021a).

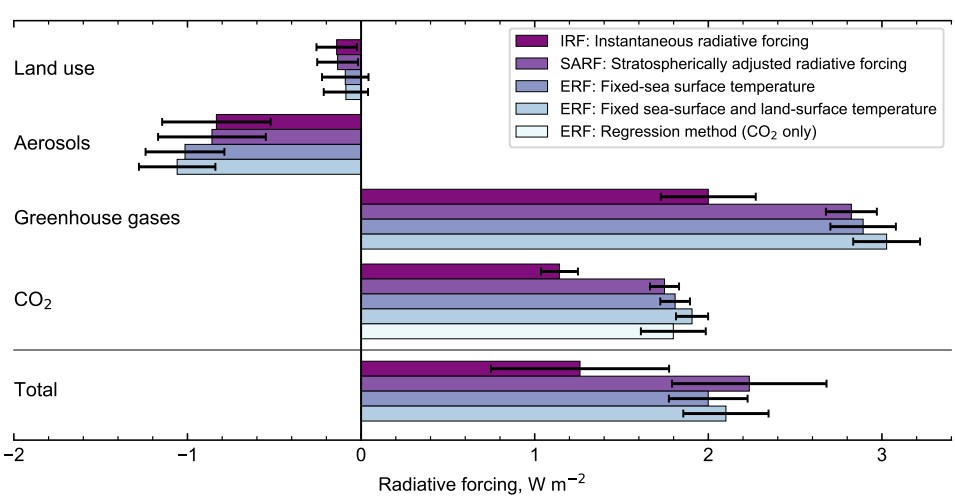

**Figure 6.** Radiative forcing for 2014 relative to 1850 from RFMIP experiments (adjusted Figure 1 from Smith et al., 2020a, under the Creative Commons Attribution 4.0 International License - CC BY 4.0). Shown are the multi-model means (bars) and the model-to-model standard deviations (lines) following different methods, graphically depicted in Figure 3. Details on the calculations are given in Smith et al. (2020a).

**Table 1.** Key results from the three MIPs for their research topics. Listed experiments are fully coupled atmosphere-ocean experiments (*hist-X*), experiments with prescribed transient changes in sea-surface temperatures and sea-ice from a historical experiment (*histSST-X*) and experiments with prescribed sea-surface temperatures and sea-ice at pre-industrial level (*piClim-X*), where *X* refers to single or several climate forcers and *piClim-2xX* refers to experiments with prescribed doubled emission fluxes.

| Topic | Experiments | Key results | References |
|---|---|---|---|
| Atmospheric Composition | *hist, histSST-X* | Model consistency in historical OH trends driven by NTCF emissions; Evolution of tropospheric and stratospheric ozone, its attribution to different drivers, and evaluation against observations; Evaluation of aerosol lifecycle, optical properties and trends in CMIP6 generation models | Stevenson et al. (2020), Griffiths et al. (2021), Keeble et al. (2021), Zeng et al. (2022), Gliß et al. (2021), Mortier et al. (2020) |
| Air Quality (AQ) & Human Health | *hist, sspX.Y, ssp370-lowNTCF, ssp370-lowNTCFCH4, ssp370SST, ssp370pdSST, ssp370SST-X* | Historical and future evolution of air pollution; Climate penalty and benefit for surface ozone; Impact of climate mitigation on AQ and human health | Turnock et al. (2020), Zanis et al. (2022), Brown et al. (2022), Allen et al. (2020, 2021), Turnock et al. (2022, 2023) |
| Climate Response | *piControl, hist, hist-piAer, ssp370, ssp370-lowNTCF, ssp370-lowNTCFCH4, piClim-X* | Climate and AQ impacts of mitigating NTCFs and non-methane NTCFs; Fast responses from aerosols in PI climate; Role of aerosols in historical climate; Impact of NTCFs on Atlantic Meridional Overturning Circulation; Regional climate extremes; Fast and Slow Precipitation responses | Allen et al. (2020, 2021), Zanis et al. (2020), Zhang et al. (2021), Hassan et al. (2022), Li et al. (2023), Samset et al. (2016) |
| Non-CO2 Biogeochemical Feedbacks | *piControl, piClim-control, Abrupt-4xCO2, piClim-2xX* | First multi-model estimates for biogeochemical feedbacks | Thornhill et al. (2021a) |

**Table 2.** Table 1 continued.

| Topic | Experiments | Key results | References |
|---|---|---|---|
| Radiative Forcing | *hist, histSST-X, piClim-X, ssp370SST-X* | Recommendations for diagnosing forcing from CMIP models; Estimates of rapid adjustments for different forcing agents; First estimates of present-day effective radiative forcing in CMIP; Historical evolution of ozone forcing; Observationally-constrained estimate of present-day halocarbon forcing; Observationally constrained time series of historical aerosol forcing; Role of chemistry-aerosol-cloud coupling in estimates of forcing; Impact of climate mitigation measures on climate forcing; new anthropogenic aerosol parameterization for use in CMIP6; Little change in aerosol forcing between 1970s and 2000s | Forster et al. (2016), Smith et al. (2018, 2020a), Skeie et al. (2020), Morgenstern et al. (2020), Smith et al. (2021a), Thornhill et al. (2021b), O'Connor et al. (2021), O'Connor et al. (2022) Turnock et al. (2022), Stevens et al. (2017), Fiedler et al. (2017, 2019, 2023) |

**Table 3.** Overview of existing experiments from RFMIP and AerChemMIP and their use in scientific publications. Numbers for existing experiments are based on data from the Earth System Grid Federation (ESGF) and publications listed by Google Scholar as of June 2023. Totals are calculated by adding the individual numbers listed aloft and are generous estimates since some publications used more than one experiment type.

| Experiment name | MIP | Number of models | Number of publications |
| --- | --- | --- | --- |
| *historical* | CMIP6 | 67 | 15100 |
| *hist-aer* | DAMIP | 15 | 111 |
| *hist-piAer* | AerChemMIP | 10 (8 also did hist-piNTCF) | 21 |
| *hist-piNTCF* | AerChemMIP | 11 (3 did only hist-piNTCF) | 14 |
| **Total number coupled experiments** | | **103** | **15246** |
| **(Total excl. *historical*)** | | **(36)** | **(146)** |
| *histSST* | CMIP6 | 12 | 32 |
| *histSST-piAer* | AerChemMIP | 7 | 9 |
| *histSST-piNTCF* | AerChemMIP | 10 | 7 |
| **Total number *histSST* experiments** | | **29** | **48** |
| **(Total excl. *histSST*)** | | **(17)** | **(16)** |
| *piClim-control* | CMIP6 | 23 | 69 |
| *piClim-histaer* | RFMIP | 10 (+4 not on ESGF) | 16 |
| *piClim-spAer-histaer* | RFMIP | 1 (+3 not on ESGF) | 1 |
| *piClim-aer* | AerChemMIP, RFMIP | 19 | 27 |
| *piClim-NTCF* | AerChemMIP | 10 | 7 |
| *piClim-spAer-aer* | RFMIP | 3 | 1 |
| **Total *piClim* experiments** | | **73** | **121** |
| **(Total excl. *piClim-control*)** | | **(50)** | **(52)** |
| **Total** | | **205** | **15415** |
| **(Total RFMIP and AerChemMIP)** | | **(103)** | **(214)** |

**Table 4.** List of known gaps in our knowledge from ESMs for different topics of the three MIPs.

| Topic | Gap in knowledge |
|---|---|
| Aerosol absorption | Aerosol absorption substantially differs across ESMs |
| Aerosol optical depth | Unknown reasons for large model spread in aerosol optical depth |
| Aerosol-cloud interactions | Unclear resolution dependency of the magnitude on global scales |
| Biogenic VOCs | Not included in all ESMs and a wide variety of emissions and responses in those that do |
| Emissions inventories of NTCFs | Emissions not well characterized |
| Fires | Most ESM do not simulate interactive fires, i.e., the role of fire for climate change can currently not b|
| Methane Feedbacks | ESMs do not simulate effects on methane concentrations |
| Mineral dust | Unclear future trend of dust aerosol concentrations in a warming world, the role of anthropogenic versus natural dust emissions for radiative forcing and feedbacks |
| Natural primary biological aerosol particles | Not included in ESMs |
| Nitrate aerosol | Not simulated by all ESMs |
| Non-DMS marine volatile organic compounds | Forcers are not well represented in ESMs, and uncertainties in process understanding |
| Pre-industrial aerosol | Pre-industrial state of aerosol burden and properties |

**Table 5.** List of opportunities for our community that can arise from novel capabilities.

| Theme | Opportunities |
|---|---|
| Machine learning | Development of new parameterization schemes that are faster and better than existing schemes |
| | Data mining to better characterize processes in big data |
| | New observational products to constrain model simulations |
| | Improvements of emulators to better inform decisions for future experiments with ESMs |
| Kilometer-scale modeling | More resolved physical processes that potentially better link changes in atmospheric composition to clouds and circulation |
| | Possibly better regional information on climate change and air quality impacts |
| | Global quantification of scale-dependence of forcing and response from synoptic to submesoscale |
| | Better understanding of scale-dependent processes relevant for atmospheric composition, such as natural emissions including mineral dust, marine organics, and others |

**Table 6.** Proposed new and improved diagnostics and experiments.

| Method | Usage |
| --- | --- |
| Improved diagnostic for PM | Air quality assessments and impact studies for health sector |
| Improved diagnostic for $O_3$ | Air quality assessments and impact studies for health sector |
| Diagnostics for hourly winds at 100 m above ground level | Wind power studies with associated impact studies for the energy sector |
| Diagnostics for hourly direct and diffuse irradiance | Solar power studies with associated impact studies for the energy sector |
| Hourly output of surface shear stress and near-surface soil moisture | Dust emission studies with impact studies for health and energy sector |
| Diagnostic from multiple calls to the radiative transfer scheme | Calculation of IRF and better understanding of process contributions to model diversity in ERF |
| Experiments with fixed sea ice, land, and sea surface temperatures | Calculation of ERF free of artifacts from land-temperature adjustments for more precise model intercomparisons on radiative forcing |
| Experiments for short-term climate change mitigation | Information for stakeholders on climate penalties and benefits from air pollution emission changes |