# Peer review of "Interactions between atmospheric composition and climate change -Progress in understanding and future opportunities from AerChemMIP, PDRMIP, and RFMIP"

_Geoscientific Model Development, 2023_

## Author Comment (AC1)

**Point-by-point reply for the manuscript „Interactions between atmospheric composition and climate change - Progress in understanding and future opportunities from AerChemMIP, PDRMIP, and RFMIP" by Fiedler et al.**

We thank the reviewers for commenting on our perspective. The comments helped us to develop a substantially revised manuscript. Newly designed tables, revised figures, and added text sharpen the presentation and give more details. Below are our replies in blue to the statements of the reviewers in black along with details on our intended changes in the manuscript.

**Response to reviewer 1**

As much as I applaud the motivation behind this paper, to show the connections between the three MIP's and thinking about ways forward, I can't recommend publication of this article as it is missing any meaningful conclusions. I don't want to discourage the authors from writing this paper, however I would like to encourage them to further develop ideas and summarize more developed ideas about ways forward.

Overall, the paper summarizes what the three MIP's are about (which has been already done in other papers), followed by presenting some new aspects to be included in future MIP's. However, all those ideas stay at the surface by just naming them, and giving short paragraphs on what those buzz words are, without really discussing how they can be connected to future MIPs. As such the paper does not give any new information and unfortunately is not useful in its current form.

Thank you for the constructive comments. We aimed for a balanced presentation of different aspects that the three MIPs have in common while keeping their specific purposes in mind. One published manuscript, even if it was much longer than this one, could not fully consider all aspects in detail and reading further literature will therefore always be necessary. The three MIPs cover different broad areas of research in Earth system sciences and there are multiple future directions that one could pursue. Our intention here is to synthesize and emphasize different developments from the three MIPs based on available literature and the experience that we made during performing the works. We anticipate this manuscript to serve as a starting point for planning new MIPs on the interactions of atmospheric composition and climate change. We think this is useful for others, especially for those who might want to design a successful MIP of their own, e.g., given the recent call for community MIPs for CMIP7 (https://wcrp-cmip.org/model-intercomparison-projects-mips/).

Please note that we would not want to recommend single specific directions for good reason. It is our belief that diversity in scientific methods and questions are essential and integral values of science. Any restriction, whether through recommendation or maybe imposed somewhere, for certain research directions would be limiting the freedom of choice of scientists and, hence, we do not suggest, or worse even prescribe, what future research should primarily focus on. The research questions and methods should be freely picked by the future MIP leaders who may or may not use the information provided in this manuscript. We intend to revise our manuscript to even better fulfill this purpose as support of the development of ideas for future MIPs on the interactions of atmospheric composition, air quality, and climate change. The reviewer comments and questions are useful and are greatly acknowledged for this purpose.

For example:

Emulators: they are just mentioned, and it should be pointed out that some very successful work including emulators, including perturbed physics experiments have already been carried out under CMIP6. What is the future vision here? How should this be more integrated into CMIP? Challenges, possibilities etc?

In our community we try to meet requirements from such simple climate models by considering them in the experimental design, e.g., for AerChemMIP2, in addition to their use in our research. The question on how to best integrate emulators into CMIP as a whole is being addressed by the CMIP Strategic ensemble design task team (https://wcrp-cmip.org/cmip7-task-teams/ensemble-design/). A future vision for the broader utility of emulators in CMIP7 might be developed by the task team.

We refer to several studies employing emulation techniques for CMIP6 including perturbed parameter ensembles. In the revised manuscript, we add the references to Sudakow et al. (2022) and Tebaldi and Knuti (2018) to the previously existing paragraphs on emulators:

"In terms of physically-based emulators of the climate system (i.e. simple climate models), RFMIP and AerChemMIP experiments were invaluable to determine aerosol ERF, ozone ERF and the factors influencing methane chemical lifetime. Some of these relationships were developed in the lead-up to AR6 WG1 and used directly in the report (e.g., Smith et al., 2021, Thornhill et al., 2021a, 2021b). (...) A review of emulation techniques that are routed in statistical mechanics highlights the potential to further improve emulators for the use in climate sciences by using machine learning (Sudakow et al., 2022). Also the difficulty of accounting for non-parametric biases of CMIP models in emulators remains (Jackson et al., 2022). Nevertheless, emulators have already been proven useful to sample parametric differences and to study climate change (e.g., Tebaldi and Knutti, 2018)." Due to the link with machine learning, we combine the former Sections 4.1.1 and 4.1.3 into one Section 4.1.1 in the revised manuscript.

Jackson, L. S., Maycock, A. C., Andrews, T., Fredriksen, H.-B., Smith, C. J., & Forster, P. M. (2022). Errors in simple climate model emulations of past and future global temperature change. *Geophysical Research Letters*, 49, e2022GL098808. https://doi.org/10.1029/2022GL098808

Sudakow, I., Pokojovy, M., & Lyakhov, D. (2022). Statistical mechanics in climate emulation: Challenges and perspectives. *Environmental Data Science, 1*, E16. doi:10.1017/eds.2022.15

Smith, C. J., Harris, G. R., Palmer, M. D., Bellouin, N., Collins, W., Myhre, G., et al. (2021). Energy budget constraints on the time history of aerosol forcing and climate sensitivity. *Journal of Geophysical Research: Atmospheres*, 126, e2020JD033622. https://doi.org/10.1029/2020JD033622

Tebaldi, C., & Knutti, R. (2018). Evaluating the accuracy of climate change pattern emulation for low warming targets. *Environmental Research Letters*, *13*(5), 055006, doi:10.1088/1748-9326/aabef2

Thornhill, G., Collins, W., Olivié, D., Skeie, R. B., Archibald, A., Bauer, S., Checa-Garcia, R., Fiedler, S., Folberth, G., Gjermundsen, A., Horowitz, L., Lamarque, J.-F., Michou, M., Mulcahy, J., Nabat, P., Naik, V., O'Connor, F. M., Paulot, F., Schulz, M., Scott, C. E., Séférian, R., Smith, C., Takemura, T., Tilmes, S., Tsigaridis, K., and Weber, J. (2021a). Climate-driven chemistry and aerosol feedbacks in CMIP6 Earth system models, Atmos. Chem. Phys., 21, 1105–1126, https://doi.org/10.5194/acp-21-1105-2021

Thornhill, G. D., Collins, W. J., Kramer, R. J., Olivié, D., Skeie, R. B., O'Connor, F. M., Abraham, N. L., Checa-Garcia, R., Bauer, S. E., Deushi, M., Emmons, L. K., Forster, P. M., Horowitz, L. W., Johnson, B., Keeble, J., Lamarque, J.-F., Michou, M., Mills, M. J., Mulcahy, J. P., Myhre, G., Nabat, P., Naik, V., Oshima, N., Schulz, M., Smith, C. J., Takemura, T., Tilmes, S., Wu, T., Zeng, G., and Zhang, J. (2021b) Effective radiative forcing from emissions of reactive gases and aerosols – a multi-model comparison, Atmos. Chem. Phys., 21, 853–874, https://doi.org/10.5194/acp-21-853-2021

Km- scale modeling: This is a very big topic. Again, here it is just mentioned without giving any perspective? How would km-scale modeling be integrated for CMIP? What are the challenges? Aka not resolving chemistry while this paper discusses composition related MIPs. A much more critical assessment is needed. Can these experiments even be done using coupled oceans? How would that be integrated? How does that relate to TriMIP climate change experiments? What is the real use as high resolution weather models already exist since a long time. How would pushing to even higher resolution solve any climate issues? There is a lot to be discussed, like connecting climate change to impacts etc. Connecting weather modeling to CMIP modelling, etc. However this paper discusses no real issues, just mentions general topics.

Agreed, it is a big topic. The best way to integrate the new generation of climate models into CMIP in general needs to be decided through broader discussion involving more than the three MIP communities here. From our perspective, there will always be a tradeoff between model resolution, experiment length, and number of experiments, and this choice needs to be adequate for the scientific question being addressed, as stated in the

manuscript. In the revised manuscript, we intend to expand on the role of spatial resolution and possible ways forward for our community, primarily by adding the following paragraphs in Section 4.1.2:.

"There is evidence that global coupled atmosphere-ocean simulations with a few kilometers resolution with first interactions with atmospheric composition, namely the carbon cycle (Hohenegger et al., 2023), can be done. Such model experiments have been carried out with a computational performance that sparks hope that kilometer-scale modeling will be possible to answer open questions concerning climate change to provide information for societal needs. Areas that kilometer-scale experiments could advance in the field of interest of the three MIPs encompass some pressing questions for the understanding of interactions of atmospheric composition and climate change. For some research questions on atmospheric composition and the associated climate response in our field, kilometer-scale experiments are already used, e.g., for a better understanding of aerosol-cloud interactions (Simpkins, 2018), which is one of the key uncertainties in ERF from ESMs (e.g., Smith et al., 2020). One question that can be better addressed with kilometer-scale experiments is the resolution dependence of radiative forcing and feedbacks, especially for those that involve clouds that are a key uncertainty in our understanding of climate change with ESMs (Stevens and Bony, 2013). Another question is to what extent more resolved meteorological processes aid in improving the representation of atmospheric composition and air quality, e.g., concerning health impacts in urban areas. "

"The community of the three MIPs will not be able to entirely rely on global kilometer-scale model experiments in CMIP7, especially in the context of a MIP since fully coupled Earth System Models (with aerosols and chemistry) at a resolution 1 km fast enough to perform multi-decadal simulations are unlikely to be ready in the timeline of CMIP7. In light of this restriction, we see two main routes forward for immediately using spatially refined information in our next MIPs. The first possible and computationally smart way is to use the output from global kilometer-scale experiments that are run for other purposes to drive offline models for aerosols and chemistry or atmospheric radiation transfer calculations. This approach is suitable to answer some but not all research questions in our community. For instance, the response of dust emission fluxes to changes in winds and moisture can be addressed with offline modeling and allows to identify underlying reasons for changes and model differences in the dust response (Fiedler et al., 2016), but the implication of such dust emission changes for air quality and climate responses can not be quantified with offline modeling. For the latter, regional one- or two-way dynamical downscaling approaches could be used. We perceive dynamical downscaling as the second main avenue for our near-future works to obtain regionally refined spatial information. Regional climate modeling is already well developed and organized via CORDEX with a focus on providing regional climate change information. Regional climate models with the capability to perform experiments with coupled aerosols and chemistry exist for instance in Europe and the US (e.g., Pietikäinen et al., 2012, Schwantes et al., 2022), but have not been used in our past MIPs. For CMIP7, UKESM2 and CESM aim also to have regional model configurations. An ensemble of regional composition-climate models therefore exists and could be used in future MIPs. The regional models will nevertheless need output from global climate model experiments with coupled aerosols and chemistry as boundary data for performing the regional experiments. As such our need for experiments with classical global ESMs is retained, at least for CMIP7, although we are not averse to the idea of moving towards global kilometer-scale modeling with a sufficient coupling of physical processes to aerosols and chemistry to address the community's research interests."

Bony, S., Stevens, B., Frierson, D. *et al*. (2015) Clouds, circulation and climate sensitivity. *Nature Geosci* 8, 261–268. https://doi.org/10.1038/ngeo2398

Hohenegger, C., Korn, P., Linardakis, L., Redler, R., Schnur, R., Adamidis, P., Bao, J., Bastin, S., Behravesh, M., Bergemann, M., Biercamp, J., Bockelmann, H., Brokopf, R., Brüggemann, N., Casaroli, L., Chegini, F., Datseris, G., Esch, M., George, G., Giorgetta, M., Gutjahr, O., Haak, H., Hanke, M., Ilyina, T., Jahns, T., Jungclaus, J., Kern, M., Klocke, D., Kluft, L., Kölling, T., Kornblueh, L., Kosukhin, S., Kroll, C., Lee, J., Mauritsen, T., Mehlmann, C., Mieslinger, T., Naumann, A. K., Paccini, L., Peinado, A., Praturi, D. S., Putrasahan, D., Rast, S., Riddick, T., Roeber, N., Schmidt, H., Schulzweida, U., Schütte, F., Segura, H., Shevchenko, R., Singh, V., Specht, M., Stephan, C. C., von Storch, J.-S., Vogel, R., Wengel, C., Winkler, M., Ziemen, F., Marotzke, J., and Stevens, B. (2023) ICON-Sapphire: simulating the components of the Earth system and their interactions at kilometer and subkilometer scales, Geosci. Model Dev., 16, 779–811, https://doi.org/10.5194/gmd-16-779-2023.

Fiedler, S., Knippertz, P., Woodward, S. *et al.* (2016) A process-based evaluation of dust-emitting winds in the CMIP5 simulation of HadGEM2-ES. *Clim Dyn* 46, 1107–1130. https://doi.org/10.1007/s00382-015-2635-9

Schwantes, R. H., Lacey, F. G., Tilmes, S., Emmons, L. K., Lauritzen, P. H., Walters, S., et al. (2022) Evaluating the impact of chemical complexity and horizontal resolution on tropospheric ozone over the conterminous US with a global variable resolution chemistry model. *Journal of Advances in Modeling Earth Systems*, 14, e2021MS002889. https://doi.org/10.1029/2021MS002889

Stevens, B., and Bony, S. (2013) What Are Climate Models Missing? *Science,* 340,1053-1054, doi:10.1126/science.1237554

Pietikäinen, J.-P., O'Donnell, D., Teichmann, C., Karstens, U., Pfeifer, S., Kazil, J., Podzun, R., Fiedler, S., Kokkola, H., Birmili, W., O'Dowd, C., Baltensperger, U., Weingartner, E., Gehrig, R., Spindler, G., Kulmala, M., Feichter, J., Jacob, D., and Laaksonen, A. (2012) The regional aerosol-climate model REMO-HAM, Geosci. Model Dev., 5, 1323–1339, https://doi.org/10.5194/gmd-5-1323-2012.

Machine learning: Again just mentioning a topic, without going deeper. There is so much to discuss. What about ML is useful for CMIP, what parts are not useful. Replacing models physics/chemistry with ML has very big problematic sides, creating not understandable black boxes. At the same time using ML might be unavoidable in the future, if higher resolution is necessary for others processes. How does this topic fit into the CMIP framework? What are possible ways forward? Again, no answers or ideas are discussed here, just giving a buzz word.

Yes, there are diverse opportunities arising from machine learning approaches spanning from data mining to parameterisation development. Again we do not want to make a general recommendation about what CMIP should do since it requires soliciting feedback from the much broader community involved in CMIP. We gave examples to illustrate how machine learning can be used for our research interest, specifically for "improving or speeding up process representations in ESMs, as well as designing smart tools for post-processing and evaluating ESM output". There is explainable machine learning and algorithm code is available, such that these methods are not necessarily black boxes. Certainly care should be taken when implementing or interpreting any novel additions to modeling. In our community, we see several routes for using machine learning to advance our research. We add these routes with the following paragraph in Section 4.1.1:

"We see primarily four areas where machine learning could help in advancing the research in our community. These are (1) to include faster and more precise representations of processes in models, e.g., for replacing or modifying physical parameterizations that are thought to not work sufficiently well in all conditions in which they are needed, (2) to develop novel ways to gain a better understanding of physical and chemical interactions, e.g., through data mining employing machine learning techniques, (3) to fill observational gaps, e.g., in satellite products to allow the creation of spatially complete data to more easily validate model results against observational information, and (4) to mimic climate responses to radiative forcing, e.g., to prioritize experiments for the design of new MIP protocols. Proofs of the concept of applying machine learning in our research field exist. (...)"

Further comments:

Figures: Figure 1 contains no information that is not conveyed already in the text. Figure 2, this could be a question of preference, but again here I don't know why a figure is needed, as the arguments between, resolution vs. complexity vs simulation length (maybe add here also ensemble size) is obvious.

We are going with the adage - a picture is worth a thousand words. We agree that the text covers what is conveyed in Figures 1 and 2 but these illustrative figures make it easier to absorb the information presented in the text. Text and figures complement each other in the manuscript. We note "ensemble size" in the revised caption of Figure 4: " The latter axis includes the ensemble size, referring to the number of members in an experiment ensemble."

A discussion could be useful to think if the CMIP6 TriMIPs asked for too many experiments. Was it useful to have so many tiers and experiments asked for? Would a simpler set of runs be more useful? Or was this the correct amount

to ask for? A critical assessment would fit into this paper and would allow this paper to go purely summarizing the previous experiments.

Good idea, we add a discussion at the beginning of Section 3.1 which includes a review of number of experiments and publications from the CMIP6 MIPs AerChemMIP and RFMIP:

"MIPs in CMIP6 as a whole asked for many experiments that jointly placed a big computational demand on climate modeling centers. The requested experiments were designed to address the MIP specific scientific questions. The three MIPs discussed here contributed to that demand. However, the diversity of research interests across the modeling centers meant that some experiments received more attention than others. Setting priorities with tiers was useful to the extent that it highlighted the priority of experiments from the MIP's perspective. In so doing, the tiers guided the participating modelers to set a focus on some experiments to have a larger model ensemble where the MIPs wanted contributions the most. However, in retrospect, some of the Tier 2 experiments may have been more useful than Tier 1. An example here is piClim-histaer (Tier 2) from RFMIP, which quantified the spread in magnitude and timing of historical aerosol forcing in CMIP6 models, was informationally rich, and a contributing factor in deriving the aerosol ERF time series for AR6 WG1. Simpler and multi-purpose experiments would be useful and less burdensome (both in terms of human and computational resources) as long as they facilitate answering the science questions laid out by the MIPs. We propose enhanced coordination across the MIPs during the experimental protocol design phase potentially aid in reducing the number of experiments. ."

The performed experiments have been used in peer-reviewed publications and the results informed the sixth assessment report of the IPCC, particularly chapters 6 and 7 of Working Group 1, and the calibration of emulators which allowed climate projections from emissions scenarios in Working Group 3. We therefore perceive the experiments of the MIPs as useful. We intend to include the new Table 1 for a new analysis of the number of experiments in the MIPs and their usage in peer-reviewed publications. Associated new text is added in Section 3.1:

"One could say more performed experiments are better for obtaining more data for the statistical analysis and for addressing more research questions, and that is certainly true but not feasible in light of restricted resources. In preparation for the next phase of AerChemMIP and RFMIP, we, therefore, revisit the question of the type and number of experiments in our requests based on refined research questions that we jointly want to address as a community. We intend to keep the computational burden for modeling centers as small as possible. In this process, we coordinate our intended activities with other initiatives close to our interests, e.g., via a series of workshops organized by us and others. It could be useful to enhance further the coordination across MIPs in designing multi-purpose experiments that could be mined for several different questions. It potentially allows to free some resources and to simplify workflows, e.g., to generate larger ensembles of identical multi-purpose experiments to account for internal variability like done for CMIP historical experiments. One such experiment type from our community would be transient coupled single-forcing experiments to quantify the contributions from different anthropogenic perturbations to climate change.

In preparation for the second phase of AerChemMIP and RFMIP, we review the current status of the number of experiments and their usage in peer-reviewed publications, summarized in Table 1. A total of 67 models performed CMIP6 *historical* experiments that were used in as many as 15100 publications. Model output to assess differences in forcing and response was, however, more restrictive, e.g., AOD output is available for 45 out of the 67 models. Most of the *historical* experiments (40) are performed with emission-driven models. The ESMs with prescribed aerosols (19) in the *historical* experiments used mostly (13) MACv2-SP (Stevens et al., 2017). MACv2-SP was developed in the framework of RFMIP and is due to the relatively broad implementation in ESMs now included in the works of the CMIP climate forcing task team, although the targeted exploitation of MACv2-SP in RFMIP was with one publication small compared to the usage of other experiments of RFMIP and AerChemMIP so far.

RFMIP and AerChemMIP received output from 103 experiments leading to 204 publications to date. We separate the RFMIP and AerChemMIP experiments here into three classes, namely experiments with full coupling between the atmosphere and ocean (*hist-X*), with prescribed sea-surface temperatures and sea-ice at pre-industrial level (*piClim-X*), and with prescribed transient changes in sea-surface temperatures and sea-ice from a *historical* experiment (*histSST-X*). Inter-comparing these classes, *piClim-X* experiments were performed the most with a total of 50 contributing models followed by *hist-X* with 36 experiments. However, hist-X is used three times more often in scientific publications (146) compared to *piClim-X (52)*. The higher computational demand of *hist-X,* therefore, seems

justified by the much larger scientific output compared to the experiments without a coupled ocean (*histSST-X* and *piClim-X*), measured by the number of published articles."

| Experiment name | MIP | Number of models (ESGF as of June 2023) | Number of publications (google scholar as of June 2023) |
|---|---|---|---|
| *historical* | CMIP6 | 67 | 15100 |
| *hist-aer* | DAMIP | 15 | 111 |
| *hist-piAer* | AerChemMIP | 10
8 also did hist-piNTCF | 21 |
| *hist-piNTCF* | AerChemMIP | 11
3 did only hist-piNTCF | 14 |
| **Total number coupled experiments (Total excl. *historical*)** | | **103**
**(36)** | **15246**
**(146)** |
| *histSST* | CMIP6 | 12 | 32 |
| *histSST-piAer* | AerChemMIP | 7 | 9 |
| *histSST-piNTCF* | AerChemMIP | 10 | 7 |
| **Total number histSST experiments (Total excl. *histSST*)** | | **29**
**(17)** | **48**
**(16)** |
| *piClim-control* | CMIP6 | 23 | 69 |
| *piClim-histaer* | RFMIP | 10 (+4 not on ESGF) | 16 |
| *piClim-spAer-histaer* | RFMIP | 1 (+3 not on ESGF) | 1 |
| *piClim-aer* | AerChemMIP, RFMIP | 19 | 27 |
| *piClim-NTCF* | AerChemMIP | 10 | 7 |
| *piClim-spAer-aer* | RFMIP | 3 | 1 |
| **Total piClim experiments (Total excl. *piClim-control*)** | | **73**
**(50)** | **121**
**(52)** |
| **Total (Total RFMIP and AerChemMIP)** | | **205**
**(103)** | **15415**
**(214)** |

**Table 1**: Overview on existing experiments from RFMIP and AerChemMIP and their use in scientific publications.

On page 5, L 125 challenges of MIP research does point out aerosol diversity, and only cites one paper using an extremely overly simplified methods, and ignoring all the work that has been done over decades by the Aerocom community, where a deep understanding has been collected on model processes and diversity.

Thanks for pointing this out. The line explicitly refers to examples of dedicated simplification for an improved understanding, namely: "Methods to separate out some of these model differences include experiments using, for instance, prescribed aerosols (e.g. Fiedler et al., 2019) or reactive trace gases (e.g. Checa-Garcia et al., 2018), (...)".

Members of the AeroCom community are important partners in the three MIPs, act as co-authors on this manuscript, and have been influential in designing the experiment protocols of MIPs discussed here. We add here: "Such experiments have also been used for a better understanding of reasons for model differences in aerosol forcing in the AeroCom community (Stier et al., 2013) and of circulation responses to idealized aerosol forcing (Voigt et al., 2017).".

Stier, P., Schutgens, N. A. J., Bellouin, N., Bian, H., Boucher, O., Chin, M., Ghan, S., Huneeus, N., Kinne, S., Lin, G., Ma, X., Myhre, G., Penner, J. E., Randles, C. A., Samset, B., Schulz, M., Takemura, T., Yu, F., Yu, H., and Zhou, C.: Host model uncertainties in aerosol radiative forcing estimates: results from the AeroCom Prescribed intercomparison study, Atmos. Chem. Phys., 13, 3245–3270, https://doi.org/10.5194/acp-13-3245-2013, 2013.

Voigt, A., Pincus, R., Stevens, B., Bony, S., Boucher, O., Bellouin, N., Lewinschal, A., Medeiros, B., Wang, Z., and Zhang, H. (2017),  Fast and slow shifts of the zonal-mean intertropical convergence zone in response to an idealized anthropogenic aerosol, *J. Adv. Model. Earth Syst.*,  9,  870–892, doi:10.1002/2016MS000902.

The abstract reads like an introduction, but maybe that is caused by the fact that no solid conclusions are drawn in the paper.

We revise the abstract to include the take-away messages stated in the conclusions that have also been adjusted for clarity.

Revised abstract:

"The climate science community aims to improve our understanding of climate change due to anthropogenic influences on atmospheric composition and the Earth's surface. Yet not all climate interactions are fully understood and uncertainty in climate model results persists as assessed in the latest Intergovernmental Panel on Climate Change (IPCC) assessment report. We synthesize current challenges and emphasize opportunities for advancing our understanding of the interactions between atmospheric composition, air quality, and climate change, as well as for quantifying model diversity. Our perspective is based on expert views from three multi-model intercomparison projects (MIPs) - the Precipitation Driver Response MIP (PDRMIP), the Aerosol and Chemistry MIP (AerChemMIP), and the Radiative Forcing MIP (RFMIP). While there are many shared interests and specialisms across the MIPs, they have their own scientific foci and specific approaches. The partial overlap between the MIPs proved useful for advancing the understanding of the perturbation-response paradigm through multi-model ensembles of Earth System Models of varying complexity. We discuss the challenges of gaining insights from Earth System Models that face computational and process representation limits and provide guidance from our lessons learned. Promising ideas to overcome some long-standing challenges in the near future are kilometer-scale experiments to better simulate circulation-dependent processes where it is possible, and machine learning approaches where they are needed, e.g., for faster and better sub-grid scale parameterizations and pattern recognition in big model data. New model constraints can arise from augmented observational products that leverage multiple datasets with machine learning approaches. Future MIPs can develop smart experiment protocols that strive towards an optimal tradeoff between resolution, complexity, and simulation number and length, and thereby, help to advance the understanding of climate change and its impacts. "

Revised conclusions:

"In part, the difficulty of simulating precipitation responses is related to the grand challenge of representing clouds and circulation, which can be addressed with newly evolving capabilities. Moreover, model-state dependencies affecting radiative forcing and climate responses can potentially be reduced or even resolved in the near future. Promising ideas are the use of:

- ESM experiments with prescribed land temperatures in addition to prescribed sea ice and sea-surface temperatures in more Earth System Models to quantify the effective radiative forcing free of artifacts arising from temperature adjustments over land,
- kilometer-scale model experiments with resolutions of a few kilometers to improve the understanding of interactions of atmospheric composition with circulation, clouds, and precipitation which are long-standing

challenges in climate modeling with coarse-resolution ESMs affecting for instance the representation of atmospheric composition and associated air quality assessments and aerosol-cloud interactions,

- novel machine learning approaches to speed up and improve parameterizations of sub-gridscale processes for experiments with ESMs and kilometer-scale models, to data mining for pattern recognition in big model output and to develop augmented observational products for new constraints on model output,

- emulation techniques to mimic climate responses to different forcings within the solution space of existing experiments to reduce the computation burden on modeling centers, and

- sufficiently long experiments or many ensemble members for an experiment to better distinguish climate and air quality responses to atmospheric composition changes from internal variability and therefore substantially reduce the risk of ambiguity in attributing responses to anthropogenic perturbations."

Page 6 Line 162 etc, This is referring to the comment: Process complexity is not reducing uncertainty. It should not be the goal to have all models to agree with each other. Model diversity should be the goal. Increased 'uncertainty' comes with the territory of increased complexity. Understanding 'uncertainty' model diversity should be the goal. The community will never get to a place where all models agree, neither should they.

The comment refers to the sentence: "High process complexity, although desirable and needed, also poses a challenge to reducing uncertainty in the assessment of the climate response to various forcings." Maybe our words can be interpreted differently than was intended. We did not mean to imply that models should all agree. We clarify the meaning in the revised manuscript by adding: "Model diversity in terms of for instance the combination of parameterizations, number and fidelity of represented processes, choice of coupling of model components, choice of the dynamical core, and the resolution is wanted. Differences in model results cannot be fully resolved to reach a perfect model agreement, but the models ideally converge to similar solutions for a given question, e.g., how the Earth's temperature responds to anthropogenic perturbations. The diversity in model results should therefore reduce over time to gain confidence in our conclusions drawn from simulated responses to imposed perturbations. AerChemMIP emphasized two challenges for reducing uncertainty. (...)"

Page 8, the reason behind differences in CMIP modelled dust trends and observational evidence, lies in the fact that most/all CMIP models are not coupled to dynamic terrestrial/dynamic vegetation models, and as such missing many feedbacks that lead to changes in dust emissions. As surface process modeling on CMIP time scales is extremely difficult, hybrid approaches could be investigated to get more interactions between emissions and ESM processes included in future CMIP experiments. This goes beyond dust, and is true for many other emission sources.

We use the simulation of dust across models as an example to describe the process understanding abyss. In the revised manuscript, we include a new overview table that visually highlights the gaps (see Table 4 in the reply to reviewer 2). We further add in the revised text: "Modeling surface conditions is a challenge and a potential source of the diversity in simulated dust trends. Not all models participating in CMIP6 have the capability to simulate interactive vegetation dynamics but some do, e.g., UKESM and GFDL-ESM4. A lack of coupled vegetation dynamics is not the only potential reason for differences in dust and other aerosols. Winds emit and transport dust aerosols and the soil erodibility is influenced by moisture from rain events, but both regional changes in circulation and precipitation differ across models such that their changes with warming are not fully understood." It would be great to look into these dependencies in depth in future model studies. Potential ways forward are presented in the revised Section 4.1.2.

The Radiative forcing paragraph, what is new here that has not already been summarized in the papers cited in this section? Conclusion Nr 1: using prescribed SST's, this is AMIP, which is part of CMIP.

We give an overview of available methods. In the conclusion, we stated: "prescribed land and ocean surface temperatures". This is different from prescribing sea-surface temperature and sea ice only, which is done in AMIP included in the CMIP6 experiment protocol. In the revised conclusion, we make this point clearer by writing: "prescribed sea ice, land- and sea-surface temperatures". Only a few models have carried out experiments in which both the sea AND land surface temperatures have been prescribed. This is described in the forcing section of the manuscript ("If the capability of fixed land-surface temperatures (Andrews et al., 2021) was facilitated in more ESMs, biases in ERF arising from surface temperature adjustments would be virtually eliminated in the future. By adopting the fixed sea- and land-surface temperature method (Figure 6), the change in the radiation budget would then be

equal to the change in the energy budget of the system, which overcomes the limitations of other methods for estimating ERF.") and we will elaborate more on it for clarity as follows:

"Prescribing sea- and land-surface temperature is different from the experiments carried out for CMIP6 and RFMIP. The requested experiments used prescribed sea-surface temperatures and sea ice following the experimental design of the Atmosphere Model Intercomparison Project (AMIP, Gates, 1992), but the land-surface temperatures were freely evolving. Prescribing the sea ice, sea- and land-surface temperatures has not been done in a MIP to date."

We further rephrase the conclusion to make the distinction clearer at that point too: "(...) performing experiments with prescribed land temperatures in addition to prescribed sea ice and sea-surface temperatures in more Earth System Models to quantify the effective radiative forcing free of artifacts arising from temperature adjustments over land (...) "

Gates, W. L., 1992: AN AMS CONTINUING SERIES: GLOBAL CHANGE--AMIP: The Atmospheric Model Intercomparison Project. *Bull. Amer. Meteor. Soc.*, **73**, 1962–1970, https://doi.org/10.1175/1520-0477(1992)073<1962:ATAMIP>2.0.CO;2.

In summary, I feel this paper can't be fixed by adding some more aspects to the chapters, I feel it needs much more holistic ideas how to push forward and eventually needs a complete rewrite.

Based on the comments from both the reviewers, we substantially revise the manuscript including highlighting new ideas and more details for clarifying content. An example is in conclusion number 1 which requires new experimental setups that not all models are yet able to do and was not done in any existing MIP.

**Reply to Reviewer 2**

[General]

I think this work serves several purposes. It reports on the different MIPs, on their interaction, on their challenges and future opportunities, and is a vision paper for the CACTI activity, which might be a future link between the different MIPs. Learning from the past and stressing what are the large/important remaining knowledge gaps, is possibly important to guide future research directions.
I think the work of these MIPs in the past 5 to 10 years has been very valuable, and their importance can hardly be overstated. These MIPs contributed largely to trying to understand conceptually the chain from initial perturbations to climate responses.
In this manuscript, the authors have put the different MIPs in a similar framework, to allow them to characterize what they differ in and what they have in common. It is nice that this manuscript tries to connect the various MIPs - such an approach can never be expected to be fully comprehensive or satisfying, but I very much appreciate the work of the authors.
It is also important to (critically) look backward to earlier existing MIPs, and synthesize ideas on ways to go forward. One can expect that MIPs will be popular for some time, but might loose interest over time. One can also expect that a new generation of scientists will come up with new ideas and areas to focus on. Within this environment, it is nice that these three MIPs have worked (closely) together (e.g., TriMIP), try to reflect on and synthesize their achievements (this manuscript), and make efforts to find synergies for the future (CACTI).
The manuscript comes up with suggestions for future research directions, and I am fully aware that it is not always easy to be precise in general prospective studies. Coming up with a list of possible future directions is brave, and one cannot expect such lists to be complete.
I think this work is valuable and is appropriate for the journal. E.g., the original description papers for RFMIP and AerChemMIP were both published in GMD. Having this prospective paper in the suggested journal is therefore appropriate.
I think however that the manuscript needs modifications before being fit for publication. Below you can first find a list of my main concerns. It is followed by a more detailed list, referring to specific lines in the manuscript. Both the main concerns and the detailed comments should be addressed by the authors.

Thank you for the positive appraisal of our manuscript and the well-reflected comments on the content. Below are our replies to the suggestions including how we intend to revise the manuscript in response to the comments.

[Main comments]

(1)
 In the backward looking view of the manuscript, one reports mainly on the collaboration, interaction and links between the different MIPs.  However, based on the title of the manuscript, one would expect that it would also report on the general "scientific progress" (or lack of it) made within the field due to these MIPs.  In that sense, I think that the title and content of Section 2 (Advancement through MIP's cross-linkages) is rather limited, and should maybe be widened up to also go more deeply in the scientific progress made.  I think Figures 4 and 5 cover part of the scientific progress, but I think more achievements can/should be mentioned to motivate the existence of these MIPs.  This should not be very long, but possibly a table with main scientific lessons learned from the MIP experiments might be an idea.

Agreed, there can be more detail on the scientific progress through the three MIPs. To that end, we split Section 1 with the revised title "Scientific Advancement" into a new Section 2.1 on "MIP's Key Results" and Section 2.2 "MIP's Cross-linkages". We include a newly designed table like suggested and add accompanying text in the new Section 2.1 like follows: „Table 2 summarizes 
[revised manuscript text omitted]

(2)
 The study is generally well written and structured, although specific sections should be improved for better understanding.  I also think that there should be an effort in making the style more homogeneous over the whole manuscript.  Also, one sometimes gets the impression that some paragraphs (or sentences) do not really belong in a section : they should be brought more in harmony with the text around them.  This is indicated in the detailed list below.

Thank you, we have revised the text for coherence and clarity. There will be changes in the structure for clarity, namely  adjusted titles of (sub)sections, combining two subsections into one, and adding a new subsection. Please refer to the manuscript with color-highlighted changes in the text..

(3)
Some synergies between the MIPs are mentioned, but one should maybe make an effort to elaborate more on this. Some examples have been given (use the same ERF definition, estimate length of simulations, diagnostics, complementary simulations, ...), but maybe the authors could try to analyze and refine it more.

Thanks, we will elaborate more in detail on synergies. We add the following in Section 2.2 :

„RFMIP asked for experiments to diagnose radiative forcing for greenhouse gases and aerosols as bulk quantities with setups parallel to DECK experiments. As such, RFMIP was able to characterise forcing in CMIP for the first time. Due to the parallel setup of the RFMIP experiments to those requested in DECK and additional overlap of experiment requests with other MIPs (DAMIP), RFMIP experiments also allowed model analyses of climate responses and climate feedbacks for well-estimated radiative forcing. AerChemMIP further separated contributions to radiative forcing into individual gases and short-lived climate forcers including different aerosol species. As such, the AerChemMIP experiment request was tailored to gain insights into why model differences in the forcing-response paradigm arise based on individual perturbations in atmospheric composition. The RFMIP tier 1 experiments were carried out by many modeling centers. Some of these contributions, e.g., from UKESM1 and CNRM, arose because the experimental setup was identical to the request in AerChemMIP. It meant that the technical workflow for performing and postprocessing the experiments was already in place such that contributing another variant of such experiments required only little effort.

Experiment requests that were differently designed in RFMIP and AerChemMIP for a similar purpose were the transient historical experiments to identify the response to individual perturbations. Specifically, RFMIP varied the quantity to be assessed over the historical period while keeping all other boundary conditions at the pre-industrial level (piClim-histX, where X is the forcing of interest), whereas AerChemMIP held the quantity to be assessed at the pre-industrial level and varied the boundary conditions for all other climate forcers over the historical period (histSST_piX). These differences in the setup hold the potential to understand where interactions and potential feedbacks arising from chemical composition changes play a role for the climate response, which has not yet been fully explored with the existing model output from the MIPs.“

We comment on the time period for forcing calculations in Section 4.2.1: "RFMIP requested 30-year long experiments for ERF calculations (Pincus et al., 2016) following earlier recommendations based on CMIP5 output (Forster et al., 2016). That experiment length proved to be sufficient for CMIP6 models contributing experiments to RFMIP, e.g., for ERF of anthropogenic aerosols although more simulated decades further improve the precision of the ERF calculation (Fiedler et al., 2017, 2019). Differently from RFMIP, AerChemMIP found that a spin-up time associated with the long-lived trace gases, e.g. halocarbons, is necessary before calculating the ERF. This meant that the approach of 30 year long time slice experiments was not entirely appropriate for the AerChemMIP experiments for all individual climate forcers. The longer spin-up period should be accounted for in future requests for new experiments for ERF calculations of such climate forcers. "

Forster, P. M., Richardson, T., Maycock, A. C., Smith, C. J., Samset, B. H., Myhre, G., Andrews, T., Pincus, R., and Schulz, M. (2016), Recommendations for diagnosing effective radiative forcing from climate models for CMIP6, *J. Geophys. Res. Atmos.*, 121, 12,460– 12,475, doi:10.1002/2016JD025320.

(4)
I am wondering whether the topics which are treated under Section 4 (Methodological opportunities) are representative for the future research environment and directions. I also do not know why it should be limited to "methodological" opportunities only - that is certainly not what the title of the manuscript suggests. I have the impression that the mentioned opportunities are all relevant, but some of them are maybe rather specific, and their total does not seem to cover the broad opportunity space. E.g., although kilometer-scale experiments might be important, they seem rather far in the future for most of the ESMs which have resolutions in the order of 50-200 km. As MIPs are designed for (many) models to participate, one should, e.g., take into consideration that part of the models are maybe not at the forefront of the scientific progress. Although the 5 topics mentioned in Section 4 are relevant, I could imagine a longer, more comprehensive and equilibrated list of relevant research questions and directions. My general impression is that these 5 opportunities do not cover well enough the directions in which these MIPs probably will go. Possibly an extended list of these opportunities could be put in a table.

We kept the exact direction of the future MIPs open as the process of defining the research goals and experimental requests for new future MIPs in CMIP7 out of the CACTI community are only now beginning. This process should be bottom-up driven by the scientists as much as possible, hence we wanted to provide an overview in this perspective piece to support a broader awareness of opportunities for others that we think are useful for the development of the next MIP protocols from CACTI. For the point on kilometer-scale experiments, we agree that a MIP is not possible with the most comprehensive ESMs on the CMIP7 timeline.

There are ways to accelerate such a development in the future, if the scientific community would strive towards it. One such way is to use machine learning to make models faster and better at representing sub-grid scale processes which includes aerosols and chemistry. This aspect is now more clearly discussed in Section 4.1.1: "(...) contribute to improving or speeding up process representations in ESMs, as well as designing smart tools for post-processing and evaluating ESM output. We see primarily four areas where machine learning could help in advancing the research in our community. These are (1) to include faster and more precise representations of processes in models, e.g., for replacing or modifying physical parameterizations that are thought to not work sufficiently well in all conditions in which they are needed, (...)"

Another way is to revisit the use of regional downscaling that in the past has been successfully applied to obtain spatial resolutions on limited-area domains of a few tens of kilometres that today are now also feasible with global models. We would elaborate more on these points with links to CORDEX in the revised manuscript.

A third way could be to include simpler models in MIPs that complement the complex model experiments to diagnose responses of interest. One example is the use of simulated atmospheric and soil conditions from CMIP models as input for aerosol emission models to understand model differences in dust responses, e.g., to pinpoint reasons for model differences in dust-aerosol emissions. This point is also newly added to the manuscript.

We add the text in Section 4.1.2:
"The community of the three MIPs will not be able to entirely rely on global kilometer-scale model experiments in CMIP7, especially in the context of a MIP since only a few models with the necessary capability for coupled aerosols and chemistry might exist in time. In light of this restriction, we see two main routes forward for immediately using spatially refined information in our next MIPs. The first possible and computationally smart way is to use the output from global kilometer-scale experiments that are run for other purposes to drive offline models for aerosols and chemistry or atmospheric radiation transfer calculations. This approach is suitable to answer some but not all research questions in our community. For instance, the response of dust emission fluxes to changes in winds and moisture can be addressed with offline modeling and allows to identify underlying reasons for changes and model differences in the dust response (Fiedler et al., 2016), but the implication of such dust emission changes for air quality and climate responses can not be quantified with offline modeling. For the latter, regional one- or two-way dynamical

downscaling approaches could be used. We perceive dynamical downscaling as the second main avenue for our near-future works to obtain regionally refined spatial information. Regional climate modeling is already well developed and organized via CORDEX with a focus on providing regional climate change information. Regional climate models with the capability to perform experiments with coupled aerosols and chemistry exist for instance in Europe and the US (e.g., Pietikainen et al., 2012, Schwantes et al., 2022), but have not been used in our past MIPs. For CMIP7, UKESM2 and CESM aim also to have regional model configurations. An ensemble of regional composition-climate models therefore exists and could be used in future MIPs. The regional models will nevertheless need output from global climate model experiments with coupled aerosols and chemistry as boundary data for performing the regional experiments. As such our need for experiments with classical global ESMs is retained, at least for CMIP7, although we are not averse to the idea of moving towards global kilometer-scale modeling with a sufficient coupling of physical processes to aerosols and chemistry to address the community's research interests."

We change the title of section 4 to "Opportunities" to better reflect the content and list opportunities in a new Table 3 for a better overview:

| Theme | Opportunities |
|---|---|
| Machine learning | Development of new parameterization schemes that are faster and better than existing schemes or learning from machines |
| | Data mining to better characterize processes in big data |
| | New observational products to constrain model simulations |
| | Improvements of emulators to better inform decisions for future experiments with ESMs |
| Kilometer-scale modeling | More resolved physical processes that potentially better link changes in atmospheric composition to clouds and circulation |
| | Possibly better regional information on climate change and air quality impacts |
| | Global quantification of scale-dependence of forcing and response from synoptic to submesoscale |
| | Better understanding of scale-dependent processes relevant for atmospheric composition, such as natural emissions including mineral dust, marine organics, and others |

**Table 3:** List of opportunities for our community that can arise from novel capabilities.

(5)
In Section 3.2 open research questions are mentioned. However, that section is rather short, and I have the impression that it is possibly under-valued. To maybe stress the ideas in this section, it might be an idea to elaborate them a bit more, and possibly list them in a table in the manuscript. Some of these topics are : non-DMS marine volatile organic compounds, natural primary biological aerosol particles, fire, dust, natural aerosol, aerosol optical properties, and aerosol optical depth.

Good idea, we design an overview table and add in the the text: "Known gaps are listed in Table 4.". We elaborate more on some examples like dust aerosols.

| Topic | Gap in knowledge |
|---|---|
| Non-DMS marine volatile organic compounds | Forcers are not well represented in ESMs, uncertrainties in process understanding |
| Natural primary biological aerosol particles | Not included in ESMs |

| Fire | Interactive fires are not simulated by most ESMs and their role for climate changes can therefore currently not be assessed |
|---|---|
| Mineral dust | Unclear trends of dust aerosol concentrations in a warming world, role of anthropogenic versus natural dust emissions on forcings and feedbacks |
| Natural aerosol | Pre-industrial state of aerosol burden and properties |
| Aerosol optical properties | Aerosol absorption substantially differs across ESMs |
| Aerosol optical depth | Unknown reasons for large model spread in aerosol optical depth |
| Aerosol-cloud interactions | Unclear resolution dependency of the magnitude on global scales |
| Emissions inventories of near-term climate forcers | Emissions not well characterized |
| Biogenic VOCs | Not included in all ESMs and wide variety of emissions and responses in those that do |

**Table 4:** List of gaps in our knowledge for different topics of the three MIPs.

(6)
I miss in the manuscript some perspective on what we have learned from approaches which did not work (if so) in these three MIPs. Although this should not be the focus of the manuscript, these questions might come up for a reader. In addition, such reflections might contain important lessons for future activities, and improve the effectiveness of future research and MIPs. I list here some thoughts which might be covered.

Thank you for the useful questions. We address them in the following and will add details in the manuscript to reflect these points.

 - Were there MIP experiments with a too small signal-to-noise ratio to be useful? Did the suggestions from Forster et al. [2016] appear to be correct? Is it valid for all variables, on all scales : regional and seasonal, or global and annual? Were the simulation lengths and number of simulations appropriate in the MIPs? Were the ensemble sizes (number of identical setups by individual model) large enough?

We add the following in Section 3.1: "The necessary data amount for separating the climate response from internal variability depends on the scientific interest. The response to variability ratio is for instance sufficiently good for the effective radiative forcing in the global multi-annual mean for most climate forcers. The suggestion from Forster et al. (2016) for performing 30 years of model experiments with the same boundary data, therefore, proved useful to diagnose global effective radiative forcing in most time-slice experiments, except for land-use changes (piClim-lu, Smith et al., 2020). We learned that the exact precision of the simulated effective radiative forcing depends on the model due to model differences in the internal variability that induced radiative perturbations from year to year (Fiedler et al., 2019, 2023). Longer simulations of 45 years are needed to diagnose the forcing of some longer-lived trace gases due to the time scale for gas transport through the stratosphere via the Brewer-Dobson circulation (O'Connor et al., 2021). For regional radiative effects, the 30 and 45 year long simulations are not sufficiently long in all regions to obtain a statistically significant for all anthropogenic perturbations. In UKESM, the aerosol radiative effects are for instance statistically significant at the 95% level over about 50% of the globe, but the effects are only statistically significant for 10% of the globe for land use and non-methane ozone precursors (O'Connor et al., 2021). Similarly, regional aerosol forcing is not statistically significant over all world regions in models contributing to RFMIP (Fiedler et al., 2019, 2023).

For model responses, the ensemble sizes and lengths were not sufficient for addressing all research questions of interest in the three MIPs. This is particularly true for regional responses that require a larger number of simulations or longer averaging for sufficiently reducing the impact of model-internal variability on the climate response. The

ensemble sizes were, however, typically large enough for quantifying annual and global responses, e.g., for the global multi-annual mean of precipitation (Myhre et al., 2018, Allen et al., 2020). Quantifying the regional response of climate to forcing requires larger ensembles of simulations, which the Regional Aerosol MIP (RAMIP, Wilcox et al., 2023) is currently addressing through requesting larger ensembles of experiments with regional perturbations of aerosols than available from AerChemMIP. Typically larger data amounts and/or magnitudes of a perturbation with decreasing spatial scale are necessary for separating a response from the internal variability."

O'Connor, F. M., Abraham, N. L., Dalvi, M., Folberth, G. A., Griffiths, P. T., Hardacre, C., Johnson, B. T., Kahana, R., Keeble, J., Kim, B., Morgenstern, O., Mulcahy, J. P., Richardson, M., Robertson, E., Seo, J., Shim, S., Teixeira, J. C., Turnock, S. T., Williams, J., Wiltshire, A. J., Woodward, S., and Zeng, G.: Assessment of pre-industrial to present-day anthropogenic climate forcing in UKESM1, Atmos. Chem. Phys., 21, 1211–1243, https://doi.org/10.5194/acp-21-1211-2021, 2021.

Wilcox, L. J., Allen, R. J., Samset, B. H., Bollasina, M. A., Griffiths, P. T., Keeble, J., Lund, M. T., Makkonen, R., Merikanto, J., O'Donnell, D., Paynter, D. J., Persad, G. G., Rumbold, S. T., Takemura, T., Tsigaridis, K., Undorf, S., and Westervelt, D. M.: The Regional Aerosol Model Intercomparison Project (RAMIP), Geosci. Model Dev., 16, 4451–4479, https://doi.org/10.5194/gmd-16-4451-2023, 2023.

 - RFMIP and AerChemMIP had quite some distinct and complementary experiments.  Possibly, there were also some experiments which better would have been combined into one experiment.  Is that something that should be better organized in the future?

This is a very good point, which we are currently addressing for RFMIP2 and AerChemMIP2. There is potential to even better organize the coordination of different experiments which we are currently discussing for the next phase of the MIPs. We reflect on the differences in experimental designs in the three MIPs in the new Section 2.1 (see reply to reviewer one).

And in Section 2.2: "Experiment requests that were differently designed in RFMIP and AerChemMIP for a similar purpose were the historical experiments to identify the response to individual perturbations. Specifically, RFMIP varied the quantity to be assessed over the historical period while keeping all other boundary conditions at the pre-industrial level, whereas AerChemMIP held the quantity to be assessed at pre-industrial level and varied the boundary conditions for all other climate forcers over the historical period. These differences in the setup hold the potential to understand where interactions and potential feedbacks arising from chemical composition changes play a role for the climate response, which has not yet been fully explored with the existing model output from the MIPs."

 - Was the degree of participation in these MIPs sufficient?  I assume RFMIP and AerChemMIP (and maybe also PDRMIP) were aimed to explain the results/behaviour of the CMIP models (explain better the final model response and its diversity).  However, not all CMIP models contributed to (all) RFMIP and AerChemMIP experiments.  Should having more CMIP models participating in AerChemMIP/RFMIP/PDRMIP be a priority?

It would be fantastic if we can win more modelling centres to do the RFMIP2 and AerChemMIP2 experiments. From our perspective these experiments are crucial to quantify the effective radiative forcing and understand associated climate responses in CMIP7. In RFMIP, the participation for the quantification of effective radiative forcing from Tier 1 experiments was very good, with up to 20 models for some experiments. It would have been very useful to have diagnostic output for aerosol optical properties from more models to explain the differences in forcing. In AerChemMIP less models participated, due to less models being able to do such experiments for individual climate forcers compared to RFMIP and the computational demand on modeling centers. An additional reason was the requirement to perform the DECK experiments as a prerequisite to participate in CMIP6 and hence AerChemMIP which chemical transport models cannot do, although their simulations might have been informative for addressing the research question. This has led to some AerChemMIP articles with only few participating models (e.g., Griffiths et al., Stevenson et al.,). In AerChemMIP2, there is the potential to include more types and numbers of models to increase the ensemble size for specific questions, even if they cannot contribute to all scientific aims of the protocol.

We add in Section 2.2: "RFMIP asked for experiments to diagnose radiative forcing for greenhouse gases and aerosols as bulk quantities with setups parallel to DECK experiments. As such, RFMIP was able to characterise forcing in CMIP for the first time. Due to the parallel setup of the RFMIP experiments to those requested in DECK and additional overlap of experiment requests with other MIPs (DAMIP), RFMIP experiments also allowed model analyses of climate responses and climate feedbacks for well-estimated radiative forcing. AerChemMIP further separated contributions to radiative forcing into individual gases and short-lived climate forcers including different aerosol species. As such, the AerChemMIP experiment request was tailored to gain insights into why model differences in the forcing-response paradigm arise based on individual perturbations in atmospheric composition. The RFMIP tier 1 experiments were carried out by many modeling centers. Some of these contributions, e.g., from UKESM1 and CNRM, arose because the experimental setup was identical to the request in AerChemMIP. It meant that the technical workflow for performing and postprocessing the experiments was already in place such that contributing another variant of such experiments required only little effort."

We further add a new Table 1 (see reply to reviewer 1) and new text in Section 3.1: "One could say more performed experiments are better for obtaining more data for the statistical analysis and for addressing more research questions, and that is certainly true but not feasible in light of restricted resources. In preparation for the next phase of AerChemMIP and RFMIP, we, therefore, revisit the question of the type and number of experiments in our request based on refined research questions that we jointly want to address as a community. In this process, we coordinate our intended activities with other initiatives that are close to our interests, e.g., via a series of workshops organized by us and others. It could be useful to further enhance the coordination across MIPs in designing multi-purpose experiments that could be mined for several different questions. It potentially allows to free some resources and to simplify workflows, e.g., to generate larger ensembles of identical multi-purpose experiments to account for internal variability like done for CMIP historical experiments. One such experiment type from our community would be transient single-forcing experiments to quantify the contributions from different anthropogenic changes.

In preparation for the second phase of AerChemMIP and RFMIP, we review the current status of the number of experiments and their usage in peer-reviewed publications, summarized in Table 1. A total of 67 models performed CMIP6 *historical* experiments that were used in as many as 15100 publications. Model output to assess differences in forcing and response was, however, more restrictive, e.g., AOD output is available for 45 out of the 67 models. Most of the *historical* experiments (40) are performed with emission-driven models. The ESMs with prescribed aerosols (19) in the *historical* experiments used mostly (13) MACv2-SP (Stevens et al., 2017). MACv2-SP was developed in the framework of RFMIP and is due to the relatively broad implementation in ESMs now included in the works of the CMIP climate forcing task team, although the targeted exploitation of MACv2-SP in RFMIP was with one publication small compared to the usage of other experiments of RFMIP and AerChemMIP so far.

RFMIP and AerChemMIP received output from 103 experiments leading to 204 publications to date. We separate the RFMIP and AerChemMIP experiments here into three classes, namely experiments with full coupling between the atmosphere and ocean (*hist-X*), with prescribed sea-surface temperatures and sea-ice at pre-industrial level (*piClim-X*), and with prescribed transient changes in sea-surface temperatures and sea-ice from a *historical* experiment (*histSST-X*). Inter-comparing these classes, *piClim-X* experiments were performed the most with a total of 50 contributing models followed by *hist-X* with 36 experiments. However, hist-*X* is used three times more often in scientific publications (146) compared to *piClim-X* (52). The higher computational demand of *hist-X,* therefore, seems justified by the much larger scientific output compared to the experiments without a coupled ocean (*histSST-X* and *piClim-X*), measured by the number of published articles."

 - Which protocols and experiments were not popular or successful : easy-aerosol?  Double calls for IRF calculations?  Long perturbed historical simulations?  Why?

There were less contributions to ERF-IRF and ERF-SpAer than to ERF-ERF, presumably due to the necessity to implement diagnostics and replacing the host model's anthropogenic aerosol parameterisation with MACv2-SP, respectively. It meant time commitment of personnel at the modelling centres to carry out these works at a time when many experiment requests from diverse MIPs were pending. Double and triple calls were available for a useful number of experiments for inter-comparing effective radiative forcing of anthropogenic aerosols, e.g., following the method by Ghan (2013) to identify direct and cloud-mediated effects or separating the instantaneous radiative forcing

from the net contribution of adjustments. Long perturbed historical simulations were computationally demanding and still done by several models, at least in parts since they were requested by several MIPs (e.g., RFMIP and DAMIP). Easy-aerosol (https://www.wcrp-climate.org/gc-clouds-circulation-activities/gc4-clouds-initiatives/368-gc-clouds-inititative3-easy-aerosol) was not part of the MIPs discussed here.

We add a paragraph in Section 3.1: "Simpler experiments are certainly always easier to perform and have the advantage that no expert at a modeling center is needed to enable the experiment and output, e.g., for implementing requested diagnostic output that is not yet available in the standard variable list of models, e.g., for RFMIP-IRF. Another example is an experiment design that needs to implement a different parameterization, e.g., for RFMIP-SpAer, which requires manpower at the modeling center for carrying out the work including coding, testing, and performing the experiments. In this case, it takes longer to finish the experiments and the associated scientific exploitation, e.g., in the case of RFMIP-SpAer several years after the work began (Fiedler et al., 2023), which is long compared to easy experiments that modelers can quickly set up via a simple change in the run script, e.g., for RFMIP-ERF. A rule of thumb for experimental design in MIPs could be choosing a setup as complex as necessary, but as simple as possible." We further add in Section 2.2: "(...) and to separate direct and cloud-mediated effects following the method by Ghan (2013) in RFMIP experiments (e.g., Fiedler et al., 2023)."

Ghan, S. J.: Technical Note: Estimating aerosol effects on cloud radiative forcing, Atmos. Chem. Phys., 13, 9971–9974, https://doi.org/10.5194/acp-13-9971-2013, 2013.

 - Why did certain protocols appear easier or harder to follow?  E.g., PDRMIP was partially driven in emission-driven and partially in concentration-driven mode, ...

It was probably a question of resources when the decision was made for participating in certain protocols in addition to the model capabilities to perform the requested simulation. We add in Section 3.1: "MIPs already have a specific class of models in mind. For AerChemMIP, emission-driven models were targeted, whereas RFMIP was also including contributions from models with less complex representations of aerosols, e.g., those using prescribed aerosol optical properties. Hence, RFMIP had more participation than for instance AerChemMIP. RFMIP and AerChemMIP were endorsed by CMIP6 and hence had a different structural organization with formal experiment protocols. PDRMIP started earlier and was in comparison more self-organized and dynamic in the MIP life cycle. Hence, PDRMIP comprises an ensemble of models of different complexity. Specifically, some of the models in PDRMIP had the capability to perform experiments with prescribed emissions whereas others needed concentrations resulting in an ensemble of experiments partially driven by emissions and partially driven by concentrations of climate forcers. "

 - Did the use of Tiers help?

We add in Section 3.1: " Setting priorities with tiers was useful to the extent that it highlighted the priority of experiments from the MIP's perspective. In so doing, the tiers guided the participating modelers to set a focus on some experiments to have a larger model ensemble where the MIPs wanted contributions the most. However, in retrospect, some of the Tier 2 experiments may have been more useful than Tier 1. An example here is *piClim-histaer* (Tier 2) from RFMIP, which quantified the spread in magnitude and timing of historical aerosol forcing in CMIP6 models, was informationally rich, and a contributing factor in deriving the aerosol ERF time series for AR6 WG1."

 - Maybe some vision on whether MIPs and their experimental demand should maybe remain limited.  Should these MIPs merge into one MIP?  Are there benefits in keeping (small) separate MIPs?

Very good question that is also on our mind for the next CACTI initiatives, specifically for RFMIP2 and AerChemMIP2. The advantage of keeping the MIPs as two separate endeavours is that we can leverage on their familiarity in the community due to their endorsement during CMIP6. Scientists already have certain research themes, class of

models, and experiment setups in mind when they see the MIP's names. More general reasons against merging (small) MIPs into larger ones are the risk of reduced clarity on the science questions because of more diverse foci, long and potentially less clear motivations for individual experiments in lengthy protocols, the loss of focus on a specific model class, as well as more management and coordination works for large MIPs, which is usually not directly funded. CMIP could for instance be seen as a great large coordinated MIP, which is well structured and financially supported. We think combining smaller MIPs that are more bottom-up driven into larger MIPs would be more difficult to do due to financial and management constraints.

We add in the revised conclusion: "The planning of the second phases of RFMIP and AerChemMIP are currently in preparation as community MIPs for CMIP7 (https://wcrp-cmip.org/model-intercomparison-projects-mips/). The advantage of keeping the MIPs as two separate and comparably smaller endeavours is that we can leverage on their familiarity in the community due to their endorsement during CMIP6, clarity in the science questions because of specific foci of the experiment request from a certain class of models, as well as acceptable workloads for the management and coordination. "

 - Is the multi-model approach put into question?  Should models be selected (even more) on their key performance before they can go into an assessment?  Does the model spread in the results represent our current uncertainty in understanding, or is it partially caused by lacking model selection?

We do not question the fundamental approach of MIPs. We add in Section 3.2:

"There is value in multi-model inter-comparisons to shed light on where the physical understanding is still limited based on the current representation of processes and where we have accomplished a satisfying advancement in our scientific understanding from such model experiments. An open and not restricted inclusion of models by key performance indicators allows broad participation of suitable ESMs in MIPs. Scientists can for their specific assessment decide which model experiments they include since not all experiments are equally suited for all questions, e.g., some models might miss processes and interactions that might be crucial to address the research question. Results of MIPs alone can not fully characterize the uncertainty, if it is at all possible since scientific knowledge might unfold in ways that can not be foreseen at present. This is what we call the process understanding abyss (Figure 2), which limits our ability in advancing the field with our available models. Other evidence should be considered in parallel or ideally in synergy with MIPs to gain new knowledge - may it be observational data from different sources or completely different models that are not suitable for participation in MIPs."

 - In consecutive CMIP rounds, one sees the number of ESM models increase. Is this an efficient way to progress science, or would reducing the number of different models, and trying to build one exceptionally high resolution and very competent model, be a better way?  Would such a uniformity block/hamper scientific creativity?  (CERN of climate science)

There should always be freedom of science. Promising ideas, i.e., those that have proven that there might be potentially great progress if they were pursued further, should find sufficient support in the interest of all without the expense of suppressing ideas of others. If we accomplish to retain this status quo for science, which might require to add more resources rather than redistributing existing ones, we would see no risk for hampering scientific creativity for an improved ability for advancing our understanding of and for tackling climate change.

 - How is the activity rate over the lifespan of a MIP: when do most model results come in?  How long does analysis go on?

We add in Section : "Some key articles based on the experiments were written and submitted close to the IPCC WG1 AR6 deadline. Submission of model output and analyses continued thereafter and are partly still ongoing at the time of writing. We expect this development to continue for several years, although with a decline in new CMIP6 model output, until a quorum of CMIP7 model output come online. Looking at the history of the use of CMIP data, we would expect that also output of RFMIP and AerChemMIP will be re-used later for documenting progress across their phases, e.g., for the effective radiative forcing, which is also often done for tracking progress across CMIP phases."

- Is the use of more observations to constrain the models the way forward?  This is slightly mentioned when discussing "sophisticated methods".  However, this point is maybe worth more focus and ideas.

It is one important aspect. We add: "Constraining ESMs with observations are a key in advancing our understanding. Although many observations and reanalysis data are already well used, more could be done in the future. Specifically, instead of comparing to single observational or reanalysis datasets, using several observational data sources would allow to quantify the observational uncertainty against which model results can be better evaluated, e.g., a good performance might mean that model results fall within the observational uncertainty. Moreover, new combined observational products could help to evaluate model output, which may include translating observables into modelled variables. In the past, some approaches have been taken to translate simulated data into the satellite observable space. In the future, machine learning seems promising to develop new and easier ways for exploiting and combining observational data suitable for comparison to model output, e.g., methods for filling observational gaps in some satellite products due to the presence of clouds have been used. Such ideas could be explored more to unfold the new potential to evaluate and constrain model results in the future in ways we have not done in the past."

 - In the manuscript nothing is said about emerging constraints.  Is that a way forward?

Emergent constraints help and might be most fruitful in combination with novel observational products (see previous comment). We add: " Future work could also expand on the use of emergent constraints for responses including feedback mechanisms. In the past, an emergent constraint approach was for instance used to address the present-day forcing of halocarbons leading to a reduced spread in the forcing estimate (Morgenstern et al., 2020). Another example is adopting the approach to constrain anthropogenic aerosol forcing from ESMs (McCoy et al., 2020)."

McCoy, I. L., McCoy, D. T., Wood, R., Regayre, L., Watson-Parris, D., Grosvenor, D. P., ... & Gordon, H. (2020). The hemispheric contrast in cloud microphysical properties constrains aerosol forcing. *Proceedings of the National Academy of Sciences*, *117*(32), 18998-19006.

Morgenstern, O.,  O'Connor, F. M.,  Johnson, B. T.,  Zeng, G.,  Mulcahy, J. P.,  Williams, J., et al. (2020).  Reappraisal of the climate impacts of ozone-depleting substances. *Geophysical Research Letters*,  47, e2020GL088295. https://doi.org/10.1029/2020GL088295

(7)
Also section 4.2 might benefit from tables containing all the suggested new diagnostics (e.g., IRF, ...) and experiments (e.g., fixed land surface temperature experiments, additional experiments for the impact community, …).

Good idea, we include a new summary Table 5 for the diagnostics and experiments in the revised manuscript:

| Method | Usage |
|---|---|
| Improved diagnostic for PM | Air quality assessments and impact studies for health sector |
| Improved diagnostic for O3 | Air quality assessments and impact studies for health sector |
| Diagnostics for hourly 100 m winds | Wind power studies with associated impact studies for energy sector |
| Diagnostics for hourly direct and diffuse irradiance | Solar power studies with associated impact studies for energy sector |
| Hourly output of surface shear stress and near-surface soil moisture | Dust emission studies with impact studies for health and energy sector |
| Diagnostic from multiple calls to the | Calculation of IRF and better understanding of process contributions to |

| radiation transfer scheme | model diversity in ERF |
| --- | --- |
| Experiments with fixed sea ice, land and sea surface temperatures | Calculation of ERF free of artefacts from land-temperature adjustments for more precise model intercomparisons on radiative forcing |
| Experiments for short-term climate change mitigation | Information for stakeholders on climate penalties and benefits from air pollution emission changes |

**Table 5:** Proposed new and improved diagnostics and experiments for the three MIPs.

Included in the manuscript in Section 4: "Opportunities arising from novel capabilities and diagnostics are listed in Table 5 and elaborated on in the following sections."

(8)
Would this be a good paper to introduce the 3 MIPs to a reader?  Possibly not, there is very little description of the experiments suggested in the 3 different. Maybe explain and characterize the MIPs better, possibly in a table/matrix.

We include a concise text that explains the rationale of the MIPs and highlight their similarities and differences to characterize them in the new Section 2.1. The cited MIP protocols completely describe the details of the MIPs along with the experiment lists.

[Specific comments]

 Below, you can find specific comments on the text of the manuscript.  I have tried to indicate as well as possible the line numbers.

Thank you.

ABSTRACT
- line 3 : "in climate model experiments" : this gives the impression that it refers to the "setup" of experiments, whereas I assume it should refer to the difference in results between models.
Change to: "uncertainty in climate model results"

 - line 4 : "this article" : I would not use the word article in the text (or abstract)
Rephrased throughout.

 - line 9 : "of varying complexity" : is it mentioned because it played an important role in advancing science?  I think the varying complexity was not an item/issue in itself.  However, the MIPs were designed in such a way that even contributions from less complex models could contribute to certain parts of the analysis.  E.g., models containing interactive aerosol but no interactive ozone could still contribute to the analysis of aerosol forcing, but not to the analysis of stratospheric ozone forcing.  In addition, that aspect is not so much related to "the partial overlap between the MIPs".  But it is true, estimating the feedback from natural emissions (which was an AerChemMIP activity), requires models to contain those interactive emissions.

Agreed, varying complexity of models is not an issue in itself, although it can be perceived as a limiting factor for participating in some experiments like stated. We think the varying complexity plays a useful role in advancing science. We intend to include the following at the beginning of Section 3.2 for a more balanced perspective:

„Although varying model complexity can be a difficulty in understanding differences between model results in a MIP, varying complexity helps in advancing our understanding of climate change. Model simulations with different complexity for instance help in quantifying contributions from feedback mechanisms to climate responses. Moreover, additional model components and representations of processes have been incorporated in Earth system models over time in addition to improvements of previously existing physical parameterization schemes and boundary data. Such model developments allowed new insights into the role of processes including feedback mechanisms for climate change, although the overall progress is possibly not as fast as one would hope for all aspects. Clouds and circulation are for instance outstanding challenges that have not been

resolved through the development of CMIP-class models to date."

- line 9-11 : "It specifically ... for estimating effective radiative forcing."  This seems rather limited if this is the main synergy coming from the 3 MIPs.
We remove the example of estimating effective radiative forcing in the abstract.

- line 12 : "... that have specific biases ..." : I don't have the impression that this gets much attention in the text later. Therefore I don't know whether it should be mentioned in the abstract.
Change to: "We discuss the challenges of gaining insights from highly complex models that face computational and process representation limits and provide guidance from our lessons learned."

- line 13 : are global kilometer-scale experiments in view in the next 5 to 10 years in the context of the relation between composition and climate change?  In the last decade, global climate models only doubled their resolution (e.g., from 2x2 degrees horizontal resolution to 1x1 degrees).  Do the authors expect to arrive at a kilometer-scale resolution in 5 to 10 years on a MIP-wide scale?

We now provide details of our thoughts on kilometer-scale experiments in Section 4.1.2 (see reply aloft).

- line 16 : "can" be evaluated -> "should" be evaluated
Change as suggested.

- line 16-18 : although I think this is important, "sophisticated methods" is a bit vague.  It is also used in a few other places in the manuscript.
Change to "observational constraints" in the revised manuscript, e.g., here "Future experiments should be evaluated and improved with observational constraints that leverage multiple datasets, and thereby, help to advance the understanding of climate change and its impacts."

1. INTRODUCTION
- line 25, 26 : why "concentrations" for GHGs but "burdens" for aerosols?
Revise to: "Radiative forcing may be caused by changes in atmospheric composition, including for instance aerosols and their precursors, greenhouse gases such as carbon dioxide and methane, as well as changes in surface albedo or irradiance."

- line 29 : "... direct impact ... instantaneous radiative forcing ..." : to a novice in the field, this should possibly be slightly better explained.  Now it seems more like defining one expression by another expression.
We add: "For example, changes in atmospheric composition and land-use perturb the Earth's radiation balance, as quantified by the radiative forcing. (...) Radiative forcing is measured in units of power density (W m-2). (...) Instantaneous radiative forcing (IRF) is the change in radiation fluxes that arise from a climate forcer, e.g., a perturbation in the atmospheric composition."

- line 33, 34 : "... can take several hundreds years depending on the magnitude" : as long as one stays within a linear regime, perturbations (whether small or large) disappear with the same timescale.  If there is interannual variability, however, the smaller perturbation will sooner disappear behind the detection limit.  Maybe one can be more precise.
Change to: "(...) can take several hundred years because of the slow response of ocean temperature. Smaller forcing and responses are more quickly masked by the internal-variability than larger perturbations."

- line 38 : "due to changes in emissions of reactive trace gases" : I assume one refers, e.g., to DMS and biogenic VOC emissions which are precursors of (radiatively active) O3.  It seems to exclude "emissions of species which have a direct impact on radiation", e.g., dust - however, I think they fall into the same category.
We add: "Moreover, changes in wind-dependent emissions of aerosols that occur due to circulation adjustments can be interpreted as chemical adjustments, although changes in aerosol emissions can occur with surface-temperature responses and would fall into the category of chemical feedback in that case."

- line 40-41 : The second part of the sentence contains twice "response" and twice "changes".  Cannot this be said in an easier way?
Thanks, change to: "Effective Radiative Forcing (ERF), measured at the atmosphere's boundaries, encompasses both the IRF and the contributions from rapid adjustments, that refer to flux-modulating changes in the system driven by IRF in the absence of surface temperature changes."

- line 41 : "steps" : is "steps" the appropriate word in the context of this paradigm?
Steps might suggest a sequential behaviour which is not necessarily the case, e.g., for responses and feedbacks. We change "steps" to "segments" throughout the manuscript.

- line 43-44 : "Understanding and quantification ... derived" : "derived" does not go well together with "quantification".
True, change to "assessed".

- line 43-44 : "typically derived" : some parts in the paradigm can possibly be derived by other models than ESMs.  I think line 51 is on the contrary correct : for climate response and feedbacks one needs the ESMs.
Add: "(...),  although other methods for some of the segments exist, e.g., radiation transfer models to compute IRF."

- line 44 : Heavens et al. (2013) : I have the impression that the text of Heavens et al. (2013) is more about running and verifying the ESMs, and not so much about disentangling the perturbation-response paradigm.
Yes, it may be misleading to include the reference in the long sentence here and we remove it.

- line 46-47 : "... simulate [aerosol and their precursor] emissions, [?] transport, and deposition [of aerosols]" : when one reads this sentence, one gets the feeling that something is missing on the location of the [?].  One could maybe change it into : "... simulate aerosol and their precursor emissions, and transport and deposition of aerosols".
Changed as suggested.

- line 44, 50 : What is meant by "design"?  I would think that "process complexity" (line 49) is part of it.
Design was meant to cover different parameterization schemes, dynamical cores, spatial girds, numerical integration, tuning, boundary data etc.. Change to: "Modern ESMs vary in their design, e.g., concerning different parameterization schemes, dynamical cores, spatial grids, numerical integration, tuning, and boundary data. They also vary in their level of complexity for representing physical, chemical, and biological processes, and how represented processes interact. (...) The simulated aerosols may interact with the radiation transfer, formation of cloud droplets and ice, or just a part of it." And change in line 49: "(...)  for instance be due to differences in process complexity and interactions within the respective ESMs."

- line 47 : "collaborates regularly" : "collaborates" gives the impression of a continuous process, whereas "regularly" gives the impression of a process with several breaks.
Change to: " regularly produces"

- line 47 : "multi-model ensembles of a common set of experiments" : this is clear and referts to multi-model. However, on line 50 in "ensembles of ESM experiments", "ensembles" probably refers to a group of different experimental setups.  Maybe this should be formulated more clearly to avoid confusion.
Change to: "Ensembles of different ESM experimental setups following the same protocol (...)"

- line 49 : "This diversity in response may be due to differences in process complexity within the respective ESMs, and/or may be due to the design and coupling of different model components" : aren't there more reasons for model diversity?  E.g., different parameterisations (without difference in complexity), different parameter values, different but equally complex dynamical cores, ...
We clarify that these things are covered by "design" (see above) and add here "for instance" since also other things may explain model diversity.

- line 57 : "The principle idea of MIPs" : maybe change into "The principle idea of a MIP"

Changed as suggested.

 - line 57 : "The principle idea" : what is the principle idea?  Do MIPs exist in other sciences?
Different MIPs exist in climate science, e.g., here RFMIP and AerChemMIP. They share the same basic idea. We change "principle" to "basic".

 - line 64 : "both the MIPs" -> "both MIPs"
Changed as suggested.

 - line 67 : "of the three MIPs" -> "of three MIPs"
Removed.

 - line 67 : the summing up after ":" is later followed by a continuation of the sentence with ",".  This is a bit strange.
Change to: "(...) connecting the scientific communities of AerChemMIP, RFMIP, and PDRMIP under one umbrella named TriMIP"

 - line 67 : what do the authors mean by "diagnostic tools"?  I think one could be more specific.
We mean for instance double calls to the radiation transfer calculation. Change to "diagnostic request" for clarity at the start of the paragraph and in line 67 removed for brevity.

 - line 66-68 : three aims are mentioned in this sentence as if they constitute a complete set.  However, a few sentences later some extra aims appear.
We move it up to align it better with the stated aims: "In so doing, we discuss the challenges of understanding multi-model climate responses and identify potential opportunities to make further advances in the research areas of these MIPs."

 - line 73 : "... in this area" : this is a bit vague.  Does it refer to "understanding multi-model climate responses"?
Changed to: "(...) in the research areas of these MIPs."

2. ADVANCEMENT THROUGH MIP'S CROSS-LINKAGES
- line 75-76 : "considering structural differences between ESMs" : how should "structural" be interpreted?  Isn't it more often on the level of parameterizations that differences arise?  Having a process in or not, should that be seen as a "structural" difference?  For me, "structural" refers to the broad technical design choices of an ESM (order of processes when numerically solving, how do components technically interact, is a coupler like OASIS used (or something else), is the ESM one model or an assembly of models, ...), whereas I do not think that those differences contribute most in the end.
Change to: "(...) considering structural differences concerning the design and the level of complexity between ESMs."

 - line 77 : "components" in the paradigm.  I would rather say that ESMs have components.  Possibly use a different word for what constitutes the paradigm.
Change to "segment" throughout.

 - line 77-78 : "RFMIP focuses on an improved understanding of the (role of) radiative forcing diversity (for the climate response)" : leaving out a few words, I thought it better described the aims of RFMIP.
Changed as suggested.

 - line 79 : "... on precipitation response to idealized atmospheric composition ..." : I would maybe skip "idealized".  I think the focus was on "precipitation response to atmospheric composition change", and the tool was indeed "using idealized atmospheric composition changes".
Remove "idealized".

 - line 80 : "earlier" : this gives the impression that there is a time dimension in the paradigm approach.
Yes, change to: "addresses all segments"

- line 80-83 : "... making these models more complex ..." : this gives the impression that AerChemMIP uses a specific class of models, which is confusing.  E.g., I think that in RFMIP a large portion of the models and in PDRMIP at least half of the models also start from emissions.
We add: "(...) than is necessary for participation in the other two MIPs (...)"

- line 83 : "inspired by each other" : PDRMIP was set up before AerChemMIP and RFMIP, so it was maybe unidirectional.
Yes, change to: "PDRMIP began earlier and to some degree inspired the experimental protocols of AerChemMIP and RFMIP."

- line 84 : "with a certain class of model in mind" : were they aiming for very different types of models?  It is true that the natural feedback quantification of AerChemMIP needs interactive natural emissions, but in general the models were possibly reasonably similar?
Added: "(...), i.e., CMIP-class models in all three MIPs and specifically AerChemMIP required more interactive processes than the other two MIPs."

- line 84-85 : "ensembles of ESM experiments of different complexity ...." : this is a bit confusing.  Did the  protocols differ in their demand for complexity, resolution, ..., or did the results just finally appear to be like that?
AerChemMIP required more interactive processes, thus more complexity, but a specific resolution of the model experiments was not requested in any of the MIP protocols. We elaborate more on these free choices at the modeling centers at the end of Section 3.1. Change to: " Taken together, there are ensembles of ESM experiments of different complexity, model resolution, number, and length in the three MIPs."

- line 86 : "A major advancement from the synergy between the three MIPs was the widespread adoption ..." : Is meant adoption outside the three MIPs mentioned here, or only within the framework of these 3 MIPs?
Add: "within and outside of the three MIPs "

- line 88 : "consistent calculation" -> "consistent diagnosis"
Changed as suggested.

- line 97-98 : "experiments where the atmospheric composition represents the values in 1850." : I think this formulation is not very nice.
Change to: "experiments with an atmospheric composition as of 1850"

- line 99 : "diagnostic calls" -> "additional diagnostic calls".
Changed as suggested.

- line 100 : "in CMIP6 models" : this sounds a bit sloppy, so maybe it is better to write : in the ESMs used in CMIP6.
Changed as suggested.

- line 107 : concerning the practice of estimating ERF : add "already mentioned"
Added.

- line 113-114 : "more relevant" : If one uses "more", I would think one needs to mention what it is compared with.
Change to: "tailored to the needs"

3. CHALLENGES IN THE MIPs RESEARCH
- line 118 : "components along the perturbation-response paradigm" : what is meant by "components"?  On line 41 and 43, the word "steps" was used in the perturbation-response paradigm ... ("component" is also used on line 122). What is further meant by "model differences"?
We replace "components" and "steps" with "segments" throughout the text. Change to: "A major challenge to further advancing the understanding of climate change with ESMs is that differences in their results for individual segments of the perturbation-response paradigm are not independent from other segments."

- line 119-121 : it is not clear what difference one wants to stress here : (1) difference in forcing for the same composition change, (2) different climate response for same forcing, or (3) different feedbacks?  Possibly write it clearer.
Change to: "Examples are an inter-model spread in forcing for the same change in atmospheric composition, and model-dependent climate responses to the same forcing involving different types and magnitudes of feedbacks."

- line 123 : What is meant by "joint strength" : common approach? (which then facilitates comparison)
Replace with "common approach".

- line 127 : Is here carefully the word "provided" used (leaving it open whether the data were actually used), as 50% of the models still used their own aerosol emission in PDRMIP?
Rephrase to: "PDRMIP asked for prescribing the same aerosol information in models (...)"

- line 131 : This is a bit confusing as the other MIPs also did atmosphere-only experiments.  Possibly "atmosphere-only" is not the correct terminology, as probably the landmodel is also active in these simulations.
Change to: "(...) and removed feedbacks by performing experiments with prescribed sea-surface conditions. (...) To that end, RFMIP requested experiments with prescribed sea-surface conditions similar to AerChemMIP to obtain precise model estimates of ERF. "

- line 131 and 133 : "atmosphere-only" versus "prescribed sea-surface conditions" : I would use one terminology.
We use "prescribed sea-surface conditions" throughout the revised text.

- line 135 : "in a complementary manner" : I am certainly aware that there has been a lot of synergy between the MIPs.  However, this gives the impression that together, the MIPs covered almost (all) the topical research questions.
Add: "(...) for addressing their specific research questions."

3.1 COMPUTATIONAL CAPACITY ABYSS
- In general I think that Section 3.1 is not so well written, and should be improved.
We substantially revise the section. Please see below for our specific replies.

- The three axis approach is nice and illustrative, but I have some concerns: (i) I would think that the triangle area is not a correct representation of the computational needs. I would rather think that, given a specific configuration, the product of the distances along the 3 axis is representative for the computational need (geometrically that would correspond to the volume of a block).  Possibly, the volume behind the triangle (tetrahedron formed by the triangle and part of the 3 axes) could also be seen as proportional with the computational needs (and could be a better quantification of the computational needs than the area of the triangle).  (ii) If one wants the volume to be representative for the computational needs, I would choose the axis linear.  E.g., one places a 1x1 resolution simulation 4 times further from the origin than a 2x2 simulation.  (iii) Setups with the same computational cost, will not lie on flat surfaces but rather on hyperbolic type of surfaces (I would think).
Thanks for the suggestion. Using a tetrahedron (three-sided pyramid) is a good idea for making the figure quantitatively more meaningfu, although the triangle is indicative of the growing computational need as we move out along the axesl.  We change in the text: "The volume of the tetrahedron between the origin and the marked triangle indicates the computational need for the experiments. The computational need scales non-linearly. (...) To account for the non-linearity in the computational need, the volume of the tetrahedron would be calculated on scaled values, i.e., an experiment with twice as fine resolution would be marked four times further away from the origin on the resolution axis. " And change in the caption of the figure: "The triangles mark potential choices along these axes with the volume of the tetrahedron filling the space between the origin and the colored triangle indicating the computational need."

- line 156-157 : "For some research questions, the complexity of ESMs can be reduced to a large degree" : please give an example.
Add: "For instance, concentrations of well-mixed greenhouse gases can be prescribed instead of being simulated from emissions, if one is interested in computing the forcing and response to a given change in the atmospheric composition. "

- line 137 : "Available computational capacity ..." -> "Limited (available) computational capacity ..."
Change to: "Limited (available) computational capacity (...)"

- line 137 : "modeling center" : maybe specify what this is for people not in the field.
Add: "Modeling centers perform the requested experiments with the ESM which they support. They contribute to the decision for which community-driven MIPs experiments of the ESM will be conducted."

- line 138 : "on short timescales" -> "in a short period of time"
Changed as suggested.

- line 138 : "choice" and "defined" is a strange combination.  It is also used on line 139.
Change to: "Not all experimental settings are explicitly defined by the MIP's experiment protocols, giving modelers room to make their own choices."

- line 138 : where is freedom left in the experimental setup?  Isn't it that rather some models do not have interactions?  E.g., in some models in an abrupt-4xCO2 experiment, vegetated area can reduce/increase which can have an impact on dust emissions (in addition to the impact of dryer/wetter/windier conditions).  In other models, the vegetated area is not allowed to change.
It is the process complexity like you say, the resolution of experiments, and the number of simulations in an ensemble, e.g., some modeling centers produce more simulations to better sample internal variability than others. We add: "Such choices may be regarding the capability of the model to specify a certain resolution "

- line 139 : "exact experimental design" : what is meant by exact?
Change "exact" to "final".

- line 139 : "Taken together, there were inevitable tradeoffs in the exact experimental design." -> "... there are ..."
Changed as suggested.

- line 142 : "in an ensemble of experiments" : unclear whether "experiments" refers to identical setup or not.
Change to: "(...) in an ensemble of different experimental setups per ESM."

- line 143 : "area of triangle" -> "volume of tetrahedron"
Changed as suggested.

- line 145 : does not scale linearly : I would still put the real cost on a linear scale; e.g., 2x2 degree horizontal resolution corresponds with 1, 1x1 degree horizontal resolution corresponds with 4, such that the volume calculation is still correct.
Yes, we explain it in the text with: "To account for the non-linearity in the computational need, the volume of the tetrahedron would be calculated on scaled values, i.e., an experiment with twice as fine resolution would be marked four times further away from the origin on the resolution axis."

- line 146 : "doubling the simulation length or number" -> doubling the simulation length or number of simulations
Changed as suggested.

- line 147 : "but this is not true for the model resolution" : I would think that it can be made true (see above)
Yes, added as above.

- line 148 : "for instances" -> "for instance", or "e.g."
Change to "for instance".

- line 149 : "has become available" -> "continues to grow"
Change to: "(...) computing power continues to grow, (...)".

- line 150 : I agree with "interactive chemistry" but I find "competition for priority of experiments" strange.  I would say that "chemistry" competes with "resolution" or with the "number of simulations", but not with the "priority or experiments".

Change to: "This is for instance the case in light of the computational cost of interactive chemistry against the resolution and the number of simulations. Additionally, all model experiments, irrespective of whether the models have interactive chemistry, compete for the priority at modeling centers due to limited computing resources."

- line 151, line 154 : I would not say "the most complex ESMs" -> "complex ESMs"

Changed as suggested.

- line 157-159 : I assume it is true, but it is so general that it would be nice to have an example.

We add: "For instance, an ESM could simulate changes in vegetation cover due to increased greenhouse gases that in turn have an impact on dust-aerosol emissions in addition to potential changes in soil moisture and winds. In less complex models, the vegetation cover is for instance prescribed such that the number of interactive physical processes is smaller. "

- line 159-161 : such that "model-internal variability" can be separated from the "mean radiative forcing", "climate response" and "impacts on air-quality" : although I think I understand what is meant, I think it lacks some better description.  I think one wants to split, e.g., the TOA imbalance in a "mean radiative forcing" and a contribution from "internal variability", and the same for the "climate response" and "impacts on air-quality".

Change to: "It makes creating large ensembles of ESM experiments possible that are needed to split for instance the imbalance in the radiation budget at the top of the atmosphere into a mean radiative forcing and contributions from internal variability. Similarly, a separation of the response in temperature or air quality into a forced signal and a contribution from internal variability is neccessary. The required ensemble size for sufficiently reducing the influence of model-internal variability on the global mean radiative forcing (e.g., Forster et al., 2016, Fiedler et al., 2017), climate responses (e.g., Maher et al., 2019, Deser et al., 2020), and impacts on air quality (e.g., Garcia-Menendez et al., 2017, Fiore et al., 2022) depends on the magnitude of the forced signal against the magnitude of the internal variability."

- line 158 versus line 162 : what is written here seems to contradict itself : "high complexity can be reduced" (line 158) <-> "high process complexity, ... and needed" (line 162)

Add: "needed for specific research questions"

- line 162 : "but also poses" -> "also poses"

Changed as suggested.

- line 162 : what is meant by "needed"?

It depends on the research question what is needed. Now: "needed for specific research questions".

- line 162-163 : in the first sentence, one mentions apparently one "challenge".  In the next sentence, two challenges appear.

Plural now in first sentence: "challenges".

- line 164 : "the number of interacting processes" : I don't know whether one can express this in a number for an ESM.

Change to: "intricacy and fidelity of represented processes".

- line 165 : "specification of the model resolution" : does this refer to the model center's choice of resolution, or the MIP-imposed resolution?

Add: "(...) by the modeling centers."

- line 167 : "prescribeid aerosols such as the spatial distribution" : this should be better formulated

Change to: "(...) while other models prescribe spatial distributions of aerosol optical properties"

- line 168 and 169 : this sentence refers to both types of differences (model capabilities and experimental setup).

However, my impression is that the former sentences only refer to model capabilities (not about experimental setup).
Remove here: "and experimental setup"

- line 170 : what is "model diversity" in the design of a MIP?
Revise to: "The second challenge comes from the consideration of model diversity in the level of complexity already in the process of designing a MIP protocol, since for instance a few models can simulate processes that most others cannot."

- line 170-171 : some models can simulate processes that others cannot : isn't that the same as "diversity in the level of complexity"?
Yes, now explicitly stated, see previous comment.

3.2 PROCESS UNDERSTANDING ABYSS
- line 176 : I would not use "most complex", but just "complex".
Changed as suggested.

- line 177 : I don't think that advancing climate science is limited because not "all" processes are represented.  One can never know or represent them all, but one can make progress by adding the ones which we think are relevant.  What the authors possibly want to say : we cannot reproduce observed climate change and simulate future climate change because we miss or did not represent some processes.  (This corresponds probably more with what is said in the second sentence of this section, line 178.)
Change to: "There are some limits to advancing climate science with today's complex ESMs since we miss or did not represent some processes that are thought to be relevant to reproduce observed and project future climate change. "

- line 179 : "are not represented represented differently" -> "are not represented or represented differently"
Changed as suggested.

- line 183 : "primary organic aerosols" : does this refer to (interactive) marine primary organic aerosols?
Yes, we add: "marine".

- line 183 : "can be represented" -> "are represented"
Changed as suggested.

- line 186 : "... with potential health impacts." : this makes one think that no other impacts will be mentioned, but in the next sentence also the impact on clouds is mentioned.
We add: "Moreover, (...)" for a better coherence of the text.

- line 199 : ".. model consensus and smaller in magnitude might suggest that they are irrelevant ..." : maybe formulate in a different way.
Change to: "And of those feedbacks that are simulated, model consensus or small magnitudes for a feedback might lead to a misleading conclusion that these feedbacks are not important. "

- line 200 : "dust trends" : possibly add "over the historical period"
Added as suggested.

- line 201 : "so much that they are of opposite sign" : a change in sign from a small negative to a small positive value is not automatically a dramatic change.
Change to: "(...) but the dust trends differ in sign across ESMs."

- line 201 : "so much so that" : maybe formulate differently
We remove "so much so that" (see previous comment).

- line 201 : I would think that a small dust feedback does not imply automatically that the dust emission changes are small.

True, especially on regional scales. Change to: "The CMIP6 models show trends of different signs and magnitudes for desert-dust aerosols over the historical time period (Bauer et al., 2020, Thornhill et al., 2021), and there is no ESM that reproduces the magnitude of the reconstructed dust increase from the pre-industrial to the present-day (...)" and add "(...), which has implications for the understanding and quantification of the radiation imbalance".

 - line 203-223 : This section seems to focus on natural processes. It is however not sure, as it is mixed with information which is maybe not only related to natural aerosol. E.g., mentioning that trends in aerosol and ozone do not fit the observations can also be related to errors in anthropogenic emission estimates; the effect that optical properties or size distribution of aerosol is biased can possibly also apply to anthropogenic aerosol. In general, I find this paragraph a bit difficult to follow, and it should be improved.
We split the paragraph into two of which one is addressing natural processes and challenges in constraining historical trends for different aerosol species and their precursors:

"Of those processes that are simulated, a large driver in model diversity for atmospheric composition is thought to stem from the representation of natural processes (e.g., Seferian et al. 2020, Zhao et al., 2022). In particular, a better understanding of natural aerosols in the rapidly warming Arctic may be a key factor in resolving the puzzle of Arctic amplification (Schmale et al., 2021), where diversity across ESMs for short-lived climate forcers is large (Whaley et al., 2022). Another example of the crucial role of representing natural processes is the ability of ESMs to simulate aerosol properties and weather on regional scales. Circulation is a grand challenge for ESMs (Bony et al., 2015), affecting the spatiotemporal distribution of aerosols. Desert-dust aerosols are, for instance, emitted and transported by winds, with a persistently large diversity across ESMs (e.g., Evan et al., 2014, Checa-Garcia et al., 2021, Zhao et al., 2022, Kok et al., 2023). The ability to accurately simulate atmospheric circulation is also relevant to the challenge of realistically simulating clouds and rainfall, including their regional trends due to atmospheric composition changes (e.g., Sperber et al., 2013, Stevens et al., 2013, Fiedler et al., 2020, Wilcox et al., 2020). The simulated clouds influence how aerosols can affect them and rainfall determines when and where aerosols are removed from the atmosphere.

There are a number of challenges in better understanding historical trends in aerosol species and their precursors from different natural and anthropogenic sources. Such knowledge would help to unravel model diversity in the evolution of aerosol forcing over time, and how it is related to time-dependent temperature biases in CMIP6 models (Flynn et al., 2020, Smith et al., 2021a, 2021b, Zhang et al., 2021). ESMs simulate, for instance, different historical trends for O3 and aerosols (Mortier et al., 2020, Griffiths et al., 2021). Even for present-day conditions, outstanding challenges for simulating aerosols persist. AerChemMIP points to model differences in the concentrations of secondary organic aerosols (Turnock et al., 2020). Moreover, aerosol optical properties are partially biased (e.g., Brown et al., 2021), the size distributions of different aerosol species are not sufficiently understood (Mahowald et al., 2014, Croft et al., 2021), and inter-model differences in aerosol optical depth persist across different phases of CMIP and AeroCom (Wilcox et al., 2013, Vogel et al., 2022)."

 - line 206 : secondary organic aerosols : mainly natural?
We add: "(...), which have natural and anthropogenic origins (Fan et al., 2022)"

Fan, W., Chen, T., Zhu, Z., Zhang, H., Qiu, Y., & Yin, D. (2022). A review of secondary organic aerosols formation focusing on organosulfates and organic nitrates. *Journal of Hazardous Materials*, *430*, 128406.

 - line 214 : this sentence discusses dust again, whereas some aspects of dust had already been mentioned in line 198-202. Maybe this could be combined.
Yes, we move: "The CMIP6 models show trends of different signs and magnitudes for desert-dust aerosols over the historical time period (Bauer et al., 2020, Thornhill et al., 2021), (...) " to the previous paragraph.

 - line 219 : Although it is true, I don't know whether the tuning is relevant in this context and should be mentioned here.
Removed for better text coherence.

4. METHODOLOGICAL OPPORTUNITIES

- line 230 : "and and finally" -> "and finally"
Changed as suggested.

- line 231 : "to understand the causes of model diversity" : I don't think one needs observations to understand the differences.  However, observations might constrain the models.
Change to: "to constrain models".

- line 233 : "the further development of the method for radiative forcing calculations" : maybe reformulate
Change to: " further improve radiative forcing calculations".

- line 232-234 : I suggest to improve the sentence
Change to: "Moreover, there is the opportunity to further improve radiative forcing calculations, and diagnostic requests for ESM experiments to allow more synergies with impact assessments."

4.1 AUGMENTED ESMs
4.1.1 EMULATORS WHERE INFORMATIVE
- line 238 : "are informed" : this is a strange expression, and not the same as on line 243; is the first one similar to "trained" (as "training" on line 244)?
Change to: "are trained on output"

- line 237-242 : "reduces computational demand", "fast calculation", "massively reduced computational cost" : the same concept is repeated several times.  Maybe avoid repetition.
We remove the repetitions.

- line 245-248 : (informed by CMIP, idealized experiments, or PPEs) is broader than line 238-239 (informed by CMIP) : maybe it should be consistent.
Added e.g., in line 238: "(...) e.g., from the MIPs and several CMIP phases."

- line 253-254 : "and explore climate responses to different forcing agents" : seems rather similar to what is mentioned on line 243
Agreed, Change in 253-254: "Also, the difficulty of accounting for non-parametric biases of CMIP models in emulators remains (Jackson et al., 2022). Nevertheless, emulators have already been proven useful to sample parametric differences and to study climate change (e.g., Tabaldi and Knutti, 2018)."

- line 254 : what is meant by "different" forcing agents?  Is it the same as on line 243 ("different forcings")?
Remove in line 254 (previous comment) and change to "radiative forcing" in line 243.

4.1.2 KILOMETER-SCALE EXPERIMENTS WHERE POSSIBLE
- line 258 : "which can be enabled" : this gives the impression that this process/evolution is not difficult.  Isn't that an underestimation?
Perhaps it needs to be tried more to say how difficult or easy the process is. We change: "can be enabled" to "would require" which is maybe neutral in tone.

- line 264 : "Simpkins (2018)": why not referring to the real paper, i.e., McCoy et al. (2018)?
We add McCoy et al. (2018).

- line 267 : "that involve" -> "that involves"
Changed as suggested.

- line 272 : "periods of a weeks to years" : "a week" or "weeks"
Change to: "time periods of a few weeks to years"

- line 272 : "they are promising to better simulate clouds, precipitation and circulation" : already mentioned on line 259-261.

We remove it here.

4.1.3 MACHINE LEARNING WHERE NEEDED
4.2 IMPROVED DIAGNOSTICS AND ANALYSIS
4.2.1 RADIATIVE FORCING CALCULATIONS
- line 299 : "e.g. Sherwood et al., 2015" -> "e.g.,"
Comma added.

 - line 299-307 : Some of this (IRF) has been mentioned earlier in the text (Section 1, line 20-42).  Maybe some connection should be made to those earlier mentions.
We improve the definitions of IRF and ERF in the introduction, and remove the definitions here.

 - line 315 : "double calls" : possibly explain this
We explain it in Section 2.2 : " (...) additional diagnostic calls to the radiation schemes, also known as double and triple radiation calls, that enabled calculations of the IRF" and add here "(Section 2.2)"

 - line 322-324 : these two methods have been mentioned earlier in the text (Section 2, line 90-95).  At least mention that in the text.
We add: "As mentioned earlier (...)"

 - line 326 : "atmosphere-only" : in such simulations the land model is also active, together with possibly parts of the sea-ice model.  So maybe one should use another term to describe this setup.
Yes, we remove the phrase "atmosphere-only" throughout, although it is often used in the community, and now use: "model experiments using prescribed sea-surface temperatures and sea ice".

 - line 334-336 : is it realistic to expect the fixed land-surface temperature method to be implemented in many ESMs soon (and thus on a MIP-wide scale)?
It is difficult to say because we have a limited view based on a few modelling centers which are involved here. We change to: "If adopting the fixed sea- and land-surface temperature method (Figure 3) in a MIP becomes feasible, (...)".

 - line 337-344 : this last paragraph is a bit different from the rest of the text in this section. It is not a radiative forcing calculation, but appears in a section with the title "Radiative forcing calculation".  It probably has its place in this section, but it should be better integrated/introduced.

True, we start with: "The radiative forcing of anthropogenic aerosols depends on the optical properties and the effects on clouds.", and add at the end of the paragraph: "RFMIP experiments point to overestimated aerosol absorption from anthropogenic black carbon and a relatively small share of natural aerosol absorption which leads to direct radiative effects of anthropogenic aerosols in some CMIP6 models which is implausible in light of other lines of evidence (Fiedler et al., 2023). That multi-model assessment was not as broad as it could have been due to the limited availability of requested output for aerosol properties and diagnostic calls to the radiation transfer scheme for aerosol effects in the CMIP6 models. If more such output is available from the next phases of RFMIP and AerChemMIP, we would learn more about the reasons for model differences in the radiative forcing of anthropogenic aerosols.  "

 - line 337 : "yet another area" : this is not a nice expression
Change to: "Improved diagnostics and observational constraints in the output analysis for aerosol burden and optical properties would be useful for better understanding the model diversity in the associated radiative forcing and the climate response."

 - line 338, 342  : "sophisticated ways (of analysis)" : this should be more specific.
Yes, change to: "observational constraints", and "Analysis of relevant and correlated model diagnostics together with observational constraints (...)", respectively.

- line 339-340 : "diversity ... limit" -> limits
Changed as suggested.

4.2.2 SYNERGIES WITH IMPACT ASSESSMENTS
- line 346-363 : although I certainly see the value of this, implementation of these extra diagnostics should not only happen in the RFMIP/PDRMIP/AerChemMIP simulations, but also in other CMIP simulations. There is maybe a task for this community, to promote the importance of these diagnostics to the wider climate change modelling community.
It would certainly be useful to more broadly think about output that better supports impact assessments. There are communities that are better suited to have a complete overview of needs for impact assessments, e.g., VIACS and ISIMIP. We are glad to support these communities with our expertise where we can. To that end, we add ideas for output that would facilitate easier assessments of impacts that are part of solving the problem of future climate change, and are connected to our interests. Specifically, output for assessments of potentials for renewable power generations from solar and wind energy in future climate change experiments could help to accelerate the knowledge transfer to the economic sector for the politically fostered energy and mobility transition to mitigate climate change. We add these along with the previously existing ideas on model output from the text in the new Table 5 (see aloft).

We revise the last paragraph: "Another opportunity to connect more with impact-oriented research can arise from ESM experiments for additional future socio-economic and mitigation-based pathways such that uncertainty in emission developments, including mitigation and associated impacts of atmospheric composition changes, can be systematically explored. In addition to new phases of AerChemMIP and RFMIP, examples are a MIP on future methane removal (Jackson et al., 2021) in support of potential climate solutions or on fire emission developments possibly accounting for the new capability to represent fire feedbacks (Teixeira et al., 2021). If these MIP communities would pursue stronger interactions with communities concerned with climate-change impacts, e.g., with the Vulnerability, Impacts, Adaptation and Climate Services (VIACS, Ruane et al., 2016) and the Inter-Sectoral Impact Model Intercomparison Project (ISIMIP, Frieler et al., 2017) community, they could enhance the usage of output from MIPs for societally relevant problems. Such engagement could lead to a better-integrated understanding of links between climate change, extremes, air quality, and the impacts in different sectors, e.g., health, energy, and economics, for climate change preparedness. "

- line 353 : "combination of species" -> "combinations of species"
Changed as suggested.

- line 363-364 : "considerations" : What is meant by "considerations"? What is meant by this sentence?
Replaced (see paragraph above).

- line 366 : "the usage of MIPs" : What is meant by this : the use of data from several MIPs? Or use the concept to start extra MIPs?
Change to: "usage of output from MIPs"

5. CONCLUSIONS
- line 373 : some ideas appear which have not been mentioned earlier, e.g., that the paradigm does not work for precipitation.
It works, but it seems less satisfying for precipitation responses due to model disagreement on regional precipitation trends due to composition changes. We add in the conclusion: "(...) , e.g., due to reduced model consensus on regional changes in precipitation compared to temperature for a given forcing." Difficulties to represent rainfall was listed in Section 3.2. We revise the part for clarity: "The ability to accurately simulate atmospheric circulation is also relevant to the challenge of realistically simulating clouds and rainfall, including their regional trends due to atmospheric composition changes (e.g., Sperber et al., 2013, Stevens et al., 2013, Fiedler et al., 2020, Wilcox et al., 2020). The simulated clouds influence how aerosols can affect them and rainfall determines when and where aerosols are removed from the atmosphere."

- line 371 : the understanding "with" Earth System Models : maybe the understanding of climate change with Earth System models

Changed as suggested.

 - line 374-375 : "In part, this is related to the grand challenge of representing clouds and circulation, which can be addressed with newly evolving capabilities." : it is not clear what "this" refers to.
Change to: "the difficulty of simulating precipitation responses"

 - line 394-396 : this last sentence (about GIANT) appears to be quite different from the rest of the paragraph (which was mainly about experimental design).  I would suggest trying to integrate this better.
True, removed the sentence on GIANT.

 - line 400-405 : I don't know whether an experiment involving 2 models should be mentioned in the conclusions.  The conclusions should have a broad general view.
True, also removed.

SHORT SENTENCES :

 I found a few short sentences, which broke the nice reading flow.  I would try to modify them, and better integrate them in the text.

 - line 80 : "AerChemMIP also focuses on quantifying radiative forcing and responses."
Change to: "AerChemMIP also focuses on quantifying radiative forcing and responses, but addresses all segments in the paradigm since all participating models simulate atmospheric composition based on emissions, transport, chemical transformations, and deposition, making these models more complex in their process representation and interactions than is necessary for participation in the other two MIPs (e.g., Thornhill et al., 2021)."

 - line 163 : "AerChemMIP emphasized two such challenges."
Change to: "There are two challenges for reducing uncertainty that can be emphasized."

 - line 199 : "Dust is one such example."
Change to: "And of those feedbacks that are simulated, model consensus or small magnitudes for a feedback might lead to a misleading conclusion that these feedbacks are not important. Dust trends over the historical period is one such example."

 - line 281-282 : "Proofs of concept from single ESMs exist."
Change to: "Proofs of the concept of applying machine learning in our research field exist."

AERCHEMMIP :

 I have the impression that AerChemMIP stresses a bit more on its achievements than the other two MIPs.  So I would suggest trying to reformulate a few sentences.  They are :

 - line 163 : "AerChemMIP emphasized two such challenges ..."
Change to: "There are two challenges for reducing uncertainty that can be emphasized."

 - line 181-183 : "AerChemMIP showed that including previously missing interactive sources of chemical species in an ESM ..."
Change to: "Including previously missing interactive sources of chemical species in an ESM has the potential for surprising results in estimates of forcing."

 - line 189 : "Of the three MIPs, AerChemMIP played a unique role ..."
Change to: "AerChemMIP played a role in the quantification of non-CO2 biogeochemical feedbacks (...)".

 - line 205 : "AerChemMIP further points to model difference ..."

Change to: "Even for present-day conditions, outstanding challenges for simulating aerosols persist, e.g., for the concentrations of secondary organic aerosol (Turnock et al., 2020), which have natural and anthropogenic origins (Fan et al., 2022)."

REFERENCES :

 - line 482 : "JOURNAL OF CLIMATE" -> "Journal of Climate"
Corrected.

 - line 633 : "GEOSCIENTIFIC MODEL DEVELOPMENT" -> "Geoscientific Model Development"
Corrected.

FIGURES :

 - Figure 1 : does "Earth System Model" in the red box refer to the AOGCM (component of an ESM)?  Maybe this figure can be improved.

This figure has been redrawn and is included below. The revised caption of Figure 1 is: "Schematic depiction of the perturbation-response paradigm in understanding and quantifying climate changes to perturbations using an Earth System model. Blue circles indicate options for simpler ESMs that prescribe perturbations in concentrations and land-use. Climate responses are simulated in the model configuration coupled to an ocean model.".

Revised Figure 1:

[Figure]

 - Figure 2 : "The main goals of AerChemMIP are ..." : "The main goal of AerChemMIP is ..."
Changed as suggested.

- Figure 2 : "... where the emissions or concentrations of the species of interest is perturbed" -> "are" perturbed
Changed as suggested.

- Figure 3 : some arrows have points in colours, some in black
Thanks, all points in black now.

- Figure 3 : in the upper right figure, the temperature in the troposphere should not change (I assume). However, the lines do not completely overlap in the troposphere, which is confusing.
Yes, the lines are overlapping in the toposphere now.

Revised Figure 3:

---

## Referee Report (RR1)

**Referee report of gmd-2023-29 manuscript :**
**Interactions between atmospheric composition and climate change – Progress in understanding and future opportunities from AerChemMIP, PDRMIP, and RFMIP**

October 6, 2023

To the authors,

I appreciate the modifications and improvements to the manuscript, and that review comments have been taken into account.

However, I repeat point (2) of my earlier review : "The study is generally well written and structured, although specific sections should be improved for better understanding. I also think that there should be an effort in making the style more homogeneous over the whole manuscript. Also, one sometimes gets the impression that some paragraphs (or sentences) do not really belong in a section : they should be brought more in harmony with the text around them. This is indicated in the detailed list below."

After reading the revised manuscript, I think there are important issues remaining concerning the text of the manuscript :
- some paragraphs contain parts that remain unclear or vague, and some sections are too long;
- ideas should be more precise, without multiple sentences containing very similar messages or information; repetition should be avoided;
- the reasoning within a paragraph should be clear, logical and well-structured; and
- the style should be more homogeneous throughout the text.

I hope the authors will make an effort to improve the manuscript. I am aware that this is a time-consuming task, but I think it is necessary in order for the manuscript to be useful and attractive. I think the manuscript does not need a structural change, but mainly an improvement in how the thoughts, ideas and reasonings are formulated. Is it clearly and concisely written? Is there no repetition? Are all the reasonings clear?

Below is a list of comments mainly related to the clarity of the text. Please take these comments into account when trying to create an updated version of the manuscript.

ABSTRACT :
- line 16 : "simulation number and length" : maybe "number of simulations and their length".

1. INTRODUCTION
- line 19-20 : "aspects" and "components" : two rather vague terms in one sentence.
- line 19-20 and line 25-27 : these sentences in the same paragraph are very similar in their message (although one gives more examples). This should be more structured, without repetition.
- line 21-22 and line 23-25 : it should be avoided that a definition of "radiative forcing" via "is called a radiative forcing" on line 23-25, comes after the first use of "radiative forcing" on line 21-22.
- line 25 : "power density" : is this the correct term?
- line 30 : "... that arise from a climate forcer, e.g., a perturbation in the atmospheric composition." This is still vague. It does not mention that IRF, in principle, is just a diagnostic, not affecting the temporal evolution of the atmosphere.
- line 34-35 : "Smaller forcings ... than larger perturbations." I don't know whether this addition is useful.

- line 35-39 : I assume that "change in atmospheric composition" is the initial perturbation, and "changes in emissions of reactive trace gases" is a consequence. For the last part, is one thinking of natural responses such as modification in emission changes from "soil NOx", "lightning NOx" and "isoprene/monoterpenes from vegetation"? Is that what is meant?
- line 42-43 : "Relevant examples are ... desert-dust and sea-spray aerosol" : maybe add "emission changes".

GENERAL : line 48-62 : This paragraph is presenting quite some ideas : ESMs, their differences, MIPs, ensembles, ... I think most of the ideas mentioned here have their place, but it should be written in a more logical and coherent way.
- line 48-49 : Understanding and quantification ... are "assessed" : that is a strange expression. I would rather use "obtained".
- line 49 : "typically" : sounds not very nice
- line 50 : "Modern" ESMs → "Current" ESMs
- line 50 : in their "design" : I would see "design" choices as high level decisions, and implementation as low-level realizations of those. For me "design" remains a strange term to use in this context.
- line 50-51 : "Modern ESMs vary in their design ... and boundary data." : the examples listed seem to refer mainly to how things are implemented at low-level, except for "concerning different parameterization schemes", which seems rather vague and could maybe fall under "complexity".
- line 53-54 : "can simulate aerosol and their precursor emissions, and transport and deposition of aerosols." What about aerosol growth and coagulation? Maybe use a general term describing that aerosol undergoes various processes in the atmosphere.
- line 55 : formation of cloud droplets and "ice" → and "ice crystals"
- line 55 : "or just part of it" : not so nice
- line 55 : "radiation" transfer → "radiative" transfer (both are used in the text; maybe choose one)
- line 55 : regularly produces : not so nice
- line 56 : "However" : I would skip it. Maybe I would build up the reasoning differently in this sentence.
- line 58 : "Ensembles of different ESM experimental setups following the same protocol ..." : this is vague

- line 62-63 : The most prominent example is the experimental "protocols" : there is an issue with singular/plural I would think. I also think that a MIP is not the same as a protocol. Finally, I would not say that the "protocol" has "informed" the "assessment reports".

**2 SCIENTIFIC ADVANCEMENT**
**2.1 MIP'S KEY RESULTS**
GENERAL : line 83-96 : I am not so happy with how this paragraph is written (the rest of this section is ok)
- line 84 : "considering structural differences" : is this the main purpose of the MIPs?
- line 88-91 : this sentence does not read very well
- line 92 : CMIP-class models : strange term
- line 92-93 : "with a certain class of models in mind" (here the reader thinks that there are 3 different ideas); "i.e., CMIP-class models in all three MIPs" (now they seem equal) and specifically AerChemMIP required .... (now one seems to be different) : this sentence is not very nice
- line 94 : "Taken together" : not nice
- line 95 : number : sounds strange

- line 98 : "irradiance, sulfate, and black carbon aerosols." → "irradiance, and sulfate and black carbon aerosols."
- line 105 : "." should be added at the end of the line
- line 108-109 : ... to estimate the real-world evolution and timing of emission changes ..." : if this refers to anthropogenic emissions, then it is a bit strange (timing of emissions is an externally imposed factor). Should one add "natural"?
- line 112 : "( hist)" : should be "(hist)"
- line 115 : where the emissions or concentrations → where "only" the emissions ...
- line 119 : "That is ..." → "It implied ..."

**2.2 MIP's CROSS LINKAGES**
- line 147 : the magnitude of a perturbation "in" the radiation budget : is "in" or "on" best representing the meaning?
- line 155 : "the parallel use of preindustrial control experiments" : this is a bit confusing. Does it refer to the fact that both MIPs use the same reference simulation? Or that for most analyses two simulations are used together to

derive results? Maybe use "common use".

- line 158-161 : should one make clear that the "double calls" are used for the IRF calculation and the "triple calls" to separate aerosol direct and cloud-mediated effects?

- line 166-169 : these two sentences on AerChemMIP fall within a paragraph mainly on RFMIP. Even if these two sentences have their place here, they refer to the first sentence in the paragraph, not to the sentence they immediately follow on. So the structure might be improved.

- line 167-169 : this is already said on line 128-130. It is allowed to mention things twice, but one should be careful with it.

- line 178 : "(histSST_piX)" → "(histSST-piX)"

- line 183-184 : "... and the specialized communities for aerosols and atmospheric chemistry that do not participate in CMIP." : my impression is that part of the AeroCOM and CCMI community participates/contributes to CMIP via actually RFMIP and AerChemMIP. So I would maybe not formulate it so strongly.

- line 195-196 : "Some key articles based ... deadline." It is unclear what the authors want to stress with this sentence. Now it feels that it stresses that there was little time and some analyses were ready only just in time, whereas it is also possible that the sentence wants to express that there was good collaboration and timing.

- line 197-198 : until a quorum of CMIP7 model output "come" online : "comes".

- line 198 : "online" → "available"

3. CHALLENGES IN THE MIP'S RESEARCH

This first paragraph is ok, but there is a lot of repetition of the same words such as "model diversity" and "model differences". I think this paragraph can be improved.

- line 202-205 : I assume that I understand what the authors want to express - however it is not very clear. Is the line of thought : When a difference in climate response to emissions is observed between models, one cannot say whether it is due to : (i) same emissions give difference in radiative forcing; (ii) same radiative forcings give different response; or (iii) same climate perturbation gives different feedback. Or does one rather want to express that spreads in the radiative forcing (for same emissions), and spreads in the response (for same forcing), cannot just be combined independently?

- line 211 : "elsewhere" (general) "... in the AeroCom community" (specific) : there seems to be a contradiction between "elsewhere" (which sounds general), and the rather specific mentioning of "AeroCom" at the end. I would write : "(in the Aerocom community) (Stier et al., 2013)"

- line 213 : "model-to-model diversity" : whereas on line 207, 208, and 210 "model diversity"

3.1 COMPUTATIONAL CAPACITY ABYSS

GENERAL : This section is not very nice to read. It is also reasonably long (4 pages is a considerable part of the manuscript). I list here detailed comments, but the section as a whole should be improved.

- line 231-233 : "Simpler and multi-purpose experiments would be useful and less ..." : This sentence falls a little bit out of context. So maybe start with "Also ..." or "In addition to making Tiers, ...".

- line 232-233 : "as long as they facilitate answering the science questions laid out by the MIPs." : I don't know whether this has to be mentioned so explicitly. I would think that the reader assumes this.

- line 233-234 : "We propose ..." : I don't know whether this conclusion belongs already here.

. line 235-237 : "Simpler experiments ..." : This sentence is not very nice.

- line 236 : the example seems to suggest that implementing a diagnostic is never a difficult task, although I think it sometimes can be.

- line 237 : "Another example is an experiment design that needs to implement ..." : grammatically not ok. So maybe : "needs the implementation"

- line 240 : "and the associated scientific exploitation "→ and "do" the scientific exploitation

- line 241 : "run script" : not so nice language

- line 242 : A rule of thumb ... : the message in this sentence is rather general, and I don't know whether it is useful.

- line 244-245 : "One could say ..." : this is not a nice sentence

- line 244-252 : This part of the text contains a lot of "we"/"our". I would try to limit that.

- line 253 : we "review" → we "reviewed"

- line 253 : of "the" number of experiments → of "a" number of experiments

- line 256 : "Model output to assess differences in forcing and response was, however, more restrictive" : is this expressed correctly?

- line 258 : emission-driven → "SLCF" emission-driven models (to avoid confusion with CO2 emission driven models from C4MIP)

- line 260-261 : in the "works" of the CMIP climate forcing task team → "work"

- line 267-268 : with 36 "experiments" → with 36 "models"

GENERAL line 271-275 : This should be improved.
- line 272-273 : Modelling centers perform the requested experiments "with the ESM which they support." : should this be mentioned explicitly? Isn't this assumed automatically?
- line 273 : They "contribute to the decision" ... : sounds strange. Isn't it that they actually decide?
- line 273 : "of" the ESM → "with" the ESM
- line 274-275 : "Not all ... giving modelers room to make their own choices." This seems to be a slightly different topic than the sentences around.
- line 275-276 : "Taken together, there are inevitable tradeoffs in the final experiment design." Not nice, very general.

GENERAL line 287-309 : This should be improved
- line 287 : "tradeoff" (is already mentioned on line 275)
- line 290 : "priority" : this is a new aspect, which has not been mentioned before. As it falls a little bit out of the suggested framework, it should be mentioned in the beginning or at the end (not in the main part of the reasoning).
- line 297 : "restricts the scope for increasing computing resources" → limits the attribution of computing resources
- line 300 : "by" a certain degree → "to" a certain degree
- line 300-301 : "while retaining sufficient process detail for the scientific problem that is to be studied" : can maybe be skipped.
- line 305-306 : "Similarly, ... is necessary." Why such a strong statement? Why not as in the former sentence : "... possible that are needed to split ..."
- line 306 : "the required ensemble size" : possibly also length of the simulation falls under this category.

- line 310 : "data amount" is a derived product. I would think that the amount of simulated years is the determining factor.
- line 310 : "scientific interest" → "research question"
- line 311 : maybe add at the end of this sentence : "in the current setup/experiment"
- line 314 : "exact precision" → "accuracy"
- line 314-315 : "... depends on the model due to model differences in the internal variability ..." → "... depends on the model's internal variability that induces ..."
- line 318 : in "all" regions → in "some" regions; or move it to the end of the sentence (then it can remain "in all regions")
- line 318-319 : the aerosol radiative effects : BC, OC, sulfate separately? Or together?
- line 322 : "For model responses" : vague
- line 322 : the ensemble sizes and lengths → the ensemble sizes and "simulation" lengths
- line 323-324 : content is very similar to line 326-328
- line 324-326 : very similar to line 310-314
- line 328-330 : "Typically larger data amounts ... from the internal variability." : not a nice sentence

GENERAL line 331-337 : vague, not well written. It should be improved.

GENERAL line 338–358 : vague, not well-written.
- line 341-342 : "emissions, transport, and deposition" : what about "growth" and "coagulation"? Maybe use an expression which covers all the aerosol processing.
- line 349 : "had more participation" : not nice
- line 349-251 : three times "hence" in three lines.
- line 350 : "had a different structural organization with formal experiment protocols." This sounds strange. Might the following be better : "had different structural organizations of their experimental protocols."?
- line 351 : "dynamic" in the MIP life cycle : "flexible"?
- line 352-354 : gives the impression that the emission-driven setup was the prefered one (mentioned first and using the expression "had the capability"), whereas it was the prescribed concentration setup which actually was the preferred one.
- line 354-357 : should be improved
- line 357-358 : very general sentence

3.2 PROCESS UNDERSTANDING ABYSS
- line 363 : "Moreover" : Why adding "moreover"?

- line 366 : ... for all aspects. : this is vague.
- line 366 : "Clouds and circulation are for instance outstanding challenges" : maybe add "Correctly representing ..." or "Understanding ...".
- line 368-372 : "There is value ..." : not a nice sentence. Maybe just start with "Multi-model inter-comparisons shed light on ...".
- line 371 : which model "output" they → which model they
- line 372 : since not all "experiments" → since not all "models"
- line 372 : "e.g., some models might miss processes and interactions that might be crucial to address the research question." Isn't this already mentioned earlier?
- line 373 : "Results of MIPs alone can not fully characterize the uncertainty ..." : Possibly start a new paragraph as this is a new thought.
- line 373-374 : "if it is at all possible" : I don't know if this general thought contributes something to the text.
- line 376 : or "ideally" in synergy : I would skip "ideally"
- line 380 : would allow us to quantify ... → would allow us to "first" quantify
- line 392-394 : Aren't both sentences expressing the same?

GENERAL line 413-447 : not so nice to read. Please improve.
- line 413-414 : "satisfyingly" simulated : not nice
- line 416 : model concensus → "erroneous" model consensus?
- line 419-420 : no ESM in CMIP5 "and" CMIP6 : no ESM in CMIP5 "or" CMIP6
- line 424-425 : "Winds emit" : not nice
- line 427-434 : vague, not nice

4. OPPORTUNITIES
4.1 AUGMENTED ESMS
4.1.1 MACHINE LEARNING WHERE USEFUL
- line 497 : Training emulators require → require(s)
- line 506 : ... remanins. → ... "however" remains.

4.1.2 KILOMETER-SCALE EXPERIMENTS
GENERAL : this section is not very well written. The different thoughts should be ordered better and presented more clearly, and repetition should be avoided.
- line 513 : Representing clouds ... → Representing clouds "correctly" ...
- line 518 : "These meteorological variables" : a little bit unclear what it refers to, but the reader will probably assume "clouds and precipitation".
- line 519 : "their" : also not so clear what it refers to.
- line 522 : "hold the potential for surprises in understanding climate responses" : the reader is left in doubt what to expect from this. So maybe be more specific.
- line 526 : "of a few weeks to years" : "years" (in plural) gives the impression that one can already do reasonably long simulations. Is that correct? Or should one write "weeks to months"?
- line 526-528 : "... in maintaining concentrated emissions, non-linearities in chemistry, and atmospheric transport of pollutants" : my impression is that the first two aspects implicitly assume and express a benefit from high resolution (maintaining strong gradients in concentration, and non-linearities can be resolved), whereas the 3rd aspect misses something, to what the transport improves : e.g., "and fine-scale resolved transport of pollutants"?
- line 531-532 : strange sentence
- line 531-534 : "... can be done." [in first sentence] "Such model experiments have been carried out ..." [second sentence]. That sounds contradictory between the first and second sentence.
- line 534 : "...to answer open questions concerning climate change to provide information for societal needs." Why is this rather general information mentioned here?
- line 528-529 : "... to advance the understanding of climate and air quality interactions." This is very similar to line 535-536 : "... for the understanding of interactions of atmospheric composition and climate change."
- line 536-542 : "For some questions ... . One question that can better be answered ..." : maybe use "Another question ..."
- line 543 : "will not be able to entirely rely on" : this is an understatement I think. With more nuance : "will not be able to mainly rely on"
- line 544 : at a resolution 1 km → at a resolution "of" 1 km
- line 547 : "The first possible and computationally smart way" → "The first possible way"

- line 553 : our near-future "works" → our near-future "work"
- line 557-558 : "... UKESM2 and CESM aim also .... An ensemble of regional composition-climate models exists ..." : I would not call a group of two models already an ensemble.
- line 557-562 : ... in "our" past MIPs. / As such "our" need for experiments ... / ... although "we" are not averse to the idea ... : I would avoid using too often "we" or "our".

**4.2 IMPROVED DIAGNOSTICS AND ANALYSIS**
**4.2.1 RADIATIVE FORCING CALCULATIONS**
- line 592-594 : "Differently from RFMIP, AerChemMIP found ... before calculating the ERF". This is already mentioned earlier, please indicate that.

**4.2.2 SYNERGY WITH IMPACT ASSESSMENTS**
- line 629 : "the opportunity" → "an opportunity"
- line 634 ; "... a smaller model spread in O3". (too general) Shouldn't it be "... in tropospheric O3 burden."?

**5. CONCLUSIONS**
- line 657 : "(ESM)" → "(ESMs)" (as on line 49)
- line 669-671 : this sentence is grammatically a bit unbalanced : "(i) to speed up and improve ..., (ii) to data mining ..., and (iii) to develop ... ." Maybe add a verb to "(ii) to data mining" : "to do data mining".
- line 674 : or many ensemble members → or "sufficiently" many ensemble members
- line 681 : closer collaborations with computer "science" → with computer "scientists"
- line 686 : "a" international vibrant community → "an"
- line 689-690 : The planning of ... and ... "are" currently → "is"
- line 690-693 : sentence sounds a bit strange

**FIGURES**
Figure 1 : The drawing of the arrows should be improved : no intermediate arrows. Possibly the authors can modify the figure to even better represent the perturbation-response paradigm.
Figure 3 : Frederiksen et al. [2021] : not in reference list. Possibly homogenize references with the main text (adding "," before year).

**TABLES**
Table 1, caption : explain "X" (as it was done in the main text); maybe explain piClim-2xEms (maybe refer to them also as piClim-2xX, but explain what it means).
Table 2, 3rd column : "little" change in aerosol forcing between 1970s and 2000s → "Little" change ...
Table 2, 4th column : O'Connor et al. (2021)";" O'Connor et al. (2022) → ","
Table 3, 4th column : The numbers on the 3 lines containing "Total number coupled experiments", "Total number histSST experiments", and "Total piClim experiments" possibly do not make a lot of sense. I assume that they are obtained by summing the individual numbers, whereas the number of studies using at least one of the experiments is probably slightly lower.
Table 4 : and their role for climate "changes" → "change"
Table 4 : Aerosol absorption substantially "differ" across ESMs → "differs"
Table 6 : Improved diagnostic for "O3" → "3" in O3 should be a subscript.

---

## Editor Decision (ED1)

Dear Author,

Thank you for the revised version of your manuscript. I have to say that I had to spend a significant amount of time to analyze your responses to Referee #2's comments. And I take the opportunity to thank Referee #2 for his detailed comments that certainly took him an even more significant amount of time to write. As Referee #2, I don't criticize the content of the paper but its form. With this new version, the manuscript has improved but what Referee #2 wrote is still true:

- some paragraphs contain parts that remain unclear or vague, and some sections are too long;
- ideas should be more precise, without multiple sentences containing very similar messages or information; repetition should be avoided;
- the reasoning within a paragraph should be clear, logical and well-structured; and
- the style should be more homogeneous throughout the text.

In many places, the sentences are over convoluted and too many (useless) words are included. Also, in many places, useless sentences or parts of sentences could be removed as they contain obvious ideas and do not bring any addition to the text. Next time, I strongly suggest to the author to have the manuscript reviewed by a colleague to simplify the text and make it more concise and right to the point.

With this in mind, here are additional simplifications that I propose, in addition to the ones that you already included in response to Referee #2's comments (the line numbers refer to version 3 of the manuscript). I also make additional remarks here below.

Additional propositions of simplification

L. 57-61: "The climate modeling community creates multi-model ensembles of a common set of ESM experiments with the same perturbations applied. The simulated climate responses can differ across a multi-model ensemble. This diversity in responses may for instance be due to differences in process complexity and interactions within the respective ESMs. Experimental protocols are used to create multi-model ensemble simulations for specific ESM experimental setups. These aim to better understand the reasons for the diversity in climate responses and feedback and to create future climate projections." : First and before last sentences are redundant; I suggest: "The climate modeling community creates experimental protocols to set up multi-model ensembles of a common set of ESM experiments. The simulated climate responses can differ across the multi-model ensemble members in response, for instance, to differences in process complexity and interactions within the respective ESMs. The aim is to better understand the reasons for the diversity in climate responses and feedback."

L.87: "… consider structural differences concerning the design and the level of complexity between ESMs." => "… consider structural differences in the design and complexity of the different ESMs."

L. 90: "… radiative forcing diversity to anthropogenic perturbations …" => "radiative forcing linked to anthropogenic perturbations …"

L. 173: "The RFMIP protocol included experiments to diagnose radiative forcing for greenhouse gases and aerosols as bulk quantities with setups parallel to the CMIP6 experiments for the "Diagnostic, Evaluation, and Characterization of Klima" (DECK)" => "The RFMIP protocol included experiments to diagnose radiative forcing from greenhouse gases and aerosols as bulk quantities with setups common to CMIP6 "Diagnostic, Evaluation, and Characterization of Klima" (DECK) experiments."

L. 216-220: "A major challenge to further advancing the understanding of climate change with ESMs is that differences in their results for individual segments of the perturbation-response paradigm are not independent of other segments. Specifically, a model-to-model difference in a climate response might be caused by various segments in the paradigm. For instance, the same emissions can lead to different ERFs, the same ERF can induce different climate responses and the same response can trigger different feedbacks across multi-model ensembles. In multi-model studies, one therefore sees inter-model spreads in forcing for the same change in atmospheric composition and model-dependent climate responses to the same forcing involving different types and magnitudes of feedbacks."

The text you added makes the paragraph even more fuzzy from my point of view. In the first sentence, you discuss dependence of what you call the "segments of the perturbation-response paradigm". In the second sentence, you discuss the fact that differences may be linked to more than one segments; I am not sure then why the second sentence starts with "Specifically". Then, if I understand well, in the 3rd sentence, you give an example of different responses in different segments. Please simplify. Consider removing the second sentence. In all cases, remove "across multi-model ensembles" in "different feedbacks across multi-model ensembles" as I think you mean that the difference in the feedbacks are differences across the ensemble members (but this is confusing as one may understand that the feedbacks themselves occur between the ensemble members).

L.226: "for a better understanding of reasons for model differences" => "for a better understanding of model differences"

L 249: "newly requested diagnostic output that is not yet available in the standard variable list of ESMs, e.g., for RFMIP-IRF" => remove "that is not yet available in the standard variable list of ESMs"

L.252-255: "In this case, it takes longer to finish the experiments and to do the associated scientific exploitation, e.g., in the case of RFMIP-SpAer several years after the work began (Fiedler et al., 2023), which is long compared to easy experiments that modelers can quickly set up via a simple change in a setting for performing an experiment, e.g., for RFMIP-ERF, thanks to prior work on the development and testing of models." This is a way too long and over convoluted sentence and does not bring much to the previous sentence. I suggest removing it and just modifying the end of the previous sentence as "…to carry out the work including coding, testing, performing the experiments and associated scientific exploitation."

L.265: "we reviewed the current status of the experiments" => "we reviewed the status of the experiments"

L.287: "They contribute to the decision for which community-driven MIP experiments with the ESM will be conducted, e.g., through granting computational resources and prioritizing experiments to be completed." => "They contribute to the decision as to which MIPs will be conducted with which priority, e.g., through granting computational resources."

L.288-290: "Additional decisions for the experiments are made by the scientists interested in the MIP. There is some room to make their own choices since not all experimental settings are explicitly defined by the MIP's experiment protocols, e.g., they may use a coarser spatial resolution and to some degree less model complexity to reduce the computational burden."

Another example of the same idea expressed twice. Simply remove this paragraph as the same ideas are expressed with slightly different word in the paragraph just after.

L.291: "There are inevitable tradeoffs in the final experimental designs for individual MIPs. Such choices can be categorized along the three axes of (1) *model complexity* addressing how many process interactions ESMs allow or how much fidelity processes have, (2) *model resolution* referring to the grid spacing of the model, and (3) *simulation length* covering the length and number of simulations in an ensemble of different experimental setups per ESM." => "There are inevitable tradeoffs in the final experimental choices that can be categorized along the three axes of (1) *model complexity* addressing how many process interactions ESMs allow or how much fidelity processes have, (2) *model resolution* referring to the grid spacing, and (3) *simulation length* covering the length and number of members in ensemble simulations. "

L.339: remove "of the experiments" in "the ensemble sizes and simulation lengths of the experiments"

L.445-455 (L. 427-434 in version 2) : Referee #2 wrote that these sentence are "vague, not nice" but you did not change anything. Please rephrase, clarify, simplify.

L.528-530 : "Much finer spatial resolutions with horizontal grid spacings of a few kilometers hold the potential to overcome some of the long-standing challenges concerning the representation of clouds, precipitation, and circulation in global climate simulations, which would require a step change in collaboration between climate science and high-performance computing (Slingo et al., 2022)."

is another example of a too long sentence; please cut in two parts, end first sentence after "global climate simulations"; in the second part, it is not clear what "which" refer to, please rephrase.

L.531 : Change "in coarse resolution models of several tens to hundreds of kilometers of grid spacings" for "in models with resolution of several tens to hundreds of kilometers".

L. 560-561: "especially in the context of a MIP since fully coupled ESMs with interactive aerosols and chemistry at a resolution of 1 km fast enough to perform multi-decadal simulations are unlikely to be ready in the time of CMIP7": please rephrase for something like "since fully coupled kilometer-scale ESMs with interactive aerosols and chemistry fast enough to perform multi-decadal simulations are unlikely to be ready in the time of CMIP7"

L. 564-566: "This approach is suitable to answer some but not all research questions in our community. For instance, the response of dust emission fluxes to changes in winds and moisture can be addressed with offline modeling and allows to identify underlying reasons for changes and model differences in the dust response (Fiedler et al., 2016), but the implication of such dust emission changes for air quality and climate responses can not be quantified with such an approach." ; please simplify for something like "This approach is suitable to answer some but not all research questions in our community, for instance, the response of dust emission fluxes to changes in winds and moisture (Fiedler et al., 2016), but not the implication of such dust emission changes for air quality and climate responses."

L. 691: "sufficiently long experiments or sufficiently many ensemble members" is not grammatically correct; I suggest ""sufficiently long experiments or enough ensemble members"

L 692: "to better distinguish climate and air quality responses to atmospheric composition changes from" is another example of useless words in a sentence; it should simply be ""to better distinguish climate and air quality responses to atmospheric composition from"

L.711: "multi-purpose experiments can be useful and less burden some in terms of human and computational resources" => "multi-purpose experiments can be useful and associated to less burden in terms of human and computational resources"

L.712: "as long as they facilitate answering the science questions laid out by the MIPs" is one example of an obvious sentence that does not bring much to the text

Figure 1 captions:  "are simulated with a model configuration coupled to an ocean model." => "can be simulated with a model configuration including coupling to an ocean model."

Additional comments and propositions of modifications

L.62-64: "Results from multi-model intercomparison projects (MIPs) are widely used to advance scientific understanding and inform stakeholders on climate change. The most prominent example is the Coupled Model Intercomparison Project (CMIP, Meehl et al., 2000) that has contributed …": I don't agree that CMIP is an example of à MIP; instead, CMIP is composed of several MIPS, please rephrase.

L.85-100: Some lines use the past tense and other the present tense, please standardise .

L.88: "for the answer to the MIP's question" => "in the answer to the MIP's question "

L.115: "and refers to the same term as short-lived climate forcers (SLCFs) used by Collins et al." => "and refers to the same concept than short-lived climate forcers (SLCFs) used by Collins et al. "

L 155: I don't understand what "quantify radiative forcing within and outside of the three MIPs" means, please clarify.

L.268: Change "out of the" for "of the" in "Available model output to assess differences in forcing and response was, however, limited, e.g., output for the mid-visible aerosol optical depth is available only for 45 out of the 67 models providing *historical* experiments."

L.345: "and also pose challenges for reducing model-based uncertainty" => "but pose challenges in reducing model-based uncertainty"

L.365: remove "in" in "do not prescribe the level of process complexity in a …"

L.545: "globally for restricted time periods of a few weeks to years" : be more specific on the number of years so to be coherent with the first part of the sentence.

L.569: make a new paragraph before "We perceive dynamical downscaling …"

L.645: As suggested by referee #2: "There is the opportunity" => "There is an opportunity" (you did not make the change).

L.710: "is smaller for the separate MIPs" => "than what it would be for one converged MIP"

Table 1: The additions made in the legend are not self-coherent. For example "hist-X" does not appear as is in the column (does it refer to "hist" and/or "hist-piAer" ?) while "histSST-X", "piClim-X and "piClim-2xX" do appear as is. Also "piControl" and all "ssp370…" are not described; please change.

Finally, I see that Table 4 has thoroughly changed without justification; can you make some comments on why you made all those changes?

---

## Author Response (AR2)

**Point-by-point reply to the review of gmd-2023-29 manuscript :
„Interactions between atmospheric composition and climate change – Progress in
understanding and future opportunities from AerChemMIP, PDRMIP, and RFMIP"**

We would like to thank the two anonymous reviewers and the editor for the appraisal of our manuscript. Please find below our replies in blue to the comments in black. We include details on how we modified the text in response to the comments and enclose a version of the revised manuscript where we marked all changes in color.

**Reviewer #1**
I don't think I need to see this manuscript again. My comments can be seen as opinion or suggestion, but not necessarily need to be included in the paper.

Thank you for your comments that helped to improve the manuscript.

**Reviewer #2**
To the authors,
I appreciate the modifications and improvements to the manuscript, and that review comments have been taken into account. However, I repeat point (2) of my earlier review : "The study is generally well written and structured, although specific sections should be improved for better understanding. I also think that there should be an effort in making the style more homogeneous over the whole manuscript. Also, one sometimes gets the impression that some paragraphs (or sentences) do not really belong in a section : they should be brought more in harmony with the text around them. This is indicated in the detailed list below."
After reading the revised manuscript, I think there are important issues remaining concerning the text of the manuscript :
- some paragraphs contain parts that remain unclear or vague, and some sections are too long;
- ideas should be more precise, without multiple sentences containing very similar messages or information; repetition should be avoided;
-   the reasoning within a paragraph should be clear, logical and well-structured; and
-   the style should be more homogeneous throughout the text.

I hope the authors will make an effort to improve the manuscript. I am aware that this is a time-consuming task, but I think it is necessary in order for the manuscript to be useful and attractive. I think the manuscript does not need a structural change, but mainly an improvement in how the thoughts, ideas and reasonings are formulated. Is it clearly and concisely written? Is there no repetition? Are all the reasonings clear?
Below is a list of comments mainly related to the clarity of the text. Please take these comments into account when trying to create an updated version of the manuscript.

Thank you for your thorough review. We appreciate your commitment and your time to help making the content more attractive for readers.

ABSTRACT :
- line 16 : "simulation number and length" : maybe "number of simulations and their length".
Changed as suggested.

1. INTRODUCTION
-   line 19-20 : "aspects" and "components" : two rather vague terms in one sentence.
Changed to: „This endeavor involves the assessment of numerous spatiotemporally changing variables in the Earth system, which can be determined by multiple, interacting physical, chemical, and biological processes."

- line 19-20 and line 25-27 : these sentences in the same paragraph are very similar in their message (although one gives more examples). This should be more structured, without repetition.

We removed the sentence in line 25-27 and rephrased the sentence aloft: „For example, changes in irradiance, land use, and atmospheric composition, including for instance aerosols and their precursors, greenhouse gases such as carbon dioxide and methane, perturb the radiation fluxes in and at the top of the atmosphere and hence the Earth's radiation balance. "

- line 21-22 and line 23-25 : it should be avoided that a definition of "radiative forcing" via "is called a radiative forcing" on line 23-25, comes after the first use of "radiative forcing" on line 21-22.

Resolved, the term radiative forcing is now first stated at the end of the paragraph.

- line 25 : "power density" : is this the correct term?

Thanks, power density with the unit for radiative forcing in brackets might be misleading without providing further context. Changed to: „and is measured as energy flux in W m$^{-2}$".

- line 30 : "... that arise from a climate forcer, e.g., a perturbation in the atmospheric composition." This is still vague. It does not mention that IRF, in principle, is just a diagnostic, not affecting the temporal evolution of the atmosphere.

Changed to: „ (…) initial change in radiation fluxes that arise from a perturbation in a climate forcer, which could be, for instance, associated with increased greenhouse gas concentrations in the atmosphere due to anthropogenic emissions, in the absence of other changes. IRF excludes any changes in the system other than an imbalance in the Earth's top-of-the-atmosphere (TOA) radiation budget and is a diagnostic output from Earth System Models (ESMs)."

- line 34-35 : "Smaller forcings ... than larger perturbations." I don't know whether this addition is useful.

Removed.

- line 35-39 : I assume that "change in atmospheric composition" is the initial perturbation, and " changes in emissions of reactive trace gases" is a consequence. For the last part, is one thinking of natural responses such as modification in emission changes from "soil NOx", " lightning NOx" and "isoprene/monoterpenes from vegetation"? Is that what is meant?

Yes, natural emissions are meant here. Examples are given further below in the revised paragraph: „Relevant examples are adjustments and feedbacks that modify desert-dust and sea-spray aerosol emission changes." The initial perturbation is the change in atmospheric composition, which is technically induced by perturbing concentrations of the atmospheric constituents directly or by perturbing the emission fluxes that lead to a change in atmospheric composition. It depends on the model and the experiment protocol which of the two are done.

- line 42-43 : "Relevant examples are ... desert-dust and sea-spray aerosol" : maybe add " emission changes".

Done, now: „Relevant examples are adjustments and feedbacks that modify desert-dust and sea-spray aerosol emission changes."

GENERAL : line 48-62 : This paragraph is presenting quite some ideas : ESMs, their differences, MIPs, ensembles, ... I think most of the ideas mentioned here have their place, but it should be written in a more logical and coherent way.

Thanks, we have changed the paragraph as follows in the reply to your specific comments.

- line 48-49 : Understanding and quantification ... are "assessed" : that is a strange expression. I would rather use "obtained".
- line 49 : "typically" : sounds not very nice

Changes as suggested: „Understanding and quantification of the different segments in the perturbation-response paradigm of climate science are obtained through experiments with Earth System Models (ESMs), (…)"

- line 50 : "Modern" ESMs ! "Current" ESMs

Changed as suggested: „Current ESMs (…)"

- line 50 : in their "design" : I would see "design" choices as high level decisions, and implementation as low-level realizations of those. For me "design" remains a strange term to use in this context.

Using design was motivated by the context of „program design" which addresses the process of high-level decisions for the structure of a program before one starts coding, consistent with your understanding. It is true that also the implementation differs across ESMs. We therefore add: „Current ESMs vary in their design and implementations, e.g., (…)"

- line 50-51 : "Modern ESMs vary in their design ... and boundary data." : the examples listed seem to refer mainly to how things are implemented at low-level, except for "concerning different parameterization schemes", which seems rather vague and could maybe fall under "complexity".

The design, implementation and complexity are not independent of each other. We therefore now change the sentence to better reflect the link in writing: „These imply diversity in the level of complexity for representing physical, chemical, and biological processes, and how represented processes interact."

- line 53-54 : "can simulate aerosol and their precursor emissions, and transport and deposition of aerosols." What about aerosol growth and coagulation? Maybe use a general term describing that aerosol undergoes various processes in the atmosphere.

We picked these as example and now expand this sentence to: „For example, some ESMs prescribe aerosol properties while models with additional process complexity simulate the complex evolution and interactions of aerosols and their precursors in the atmosphere (…)"

- line 55 : formation of cloud droplets and "ice" ! and "ice crystals"

Changed as suggested: „ice crystals"

- line 55 : "or just part of it" : not so nice

Now more specific: „The simulated aerosols may interact with the radiative transfer and formation of cloud droplets and ice crystals, but not all ESMs simulate all interactions with the cloud microphysics."

- line 55 : "radiation" transfer ! "radiative" transfer (both are used in the text; maybe choose one)

Thanks for spotting this. Now it is radiative transfer throughout the manuscript.

- line 55 : regularly produces : not so nice

Removed and now: „The climate modeling community creates multi-model ensembles (…)"

- line 56 : "However" : I would skip it. Maybe I would build up the reasoning differently in this sentence.

Now: „The climate modeling community creates multi-model ensembles of a common set of ESM experiments with the same perturbations applied. The simulated climate responses can differ across a multi-model ensemble. This diversity in responses may for instance be due (…)"

- line 58 : "Ensembles of different ESM experimental setups following the same protocol ..." : this is vague

Agreed, now changed to: „Experimental protocols are used to create multi-model ensemble simulations for specific ESM experimental setups. These aim to better understand the reasons for the diversity in climate responses and feedback and to create future climate projections. "

- line 62-63 : The most prominent example is the experimental "protocols" : there is an issue with singular/plural I would think. I also think that a MIP is not the same as a protocol.

Corrected: „The most prominent example is the Coupled Model Intercomparison Project (…)

- Finally, I would not say that the "protocol" has "informed" the "assessment reports".

Changes to: „that has contributed through multiple phases to the assessment reports of the Intergovernmental Panel on Climate Change (IPCC, Meehl et al., 2023), e.g., the sixth phase of CMIP (CMIP6, Eyring et al., 2016) created experiments that were also used in the sixth IPCC assessment report (IPCC-AR6).“

2 SCIENTIFIC ADVANCEMENT
2.1 MIP'S KEY RESULTS
GENERAL : line 83-96 : I am not so happy with how this paragraph is written (the rest of this section is ok)
We revise the paragraph as follows.
- line 84 : "considering structural differences" : is this the main purpose of the MIPs?
The purpose is the first part of the sentence: „The three MIPs sought to advance the understanding of modern climate change due to anthropogenic influences.“ We split the second part and explain it more as follows: „MIPs address specific research questions and, in comparison to studies with a single ESM, consider structural differences concerning the design and the level of complexity between ESMs. The multi-model spread in the response allows the quantification of a model-based uncertainty for the answer to the MIP's question.“

- line 88-91 : this sentence does not read very well
We now split the content of the sentence as follows: „AerChemMIP also focused on quantifying radiative forcing and responses but addressed more segments in the paradigm. Specifically, all participating models in AerChemMIP simulated atmospheric composition based on emissions, transport, chemical transformations, and deposition, making these models more complex in their process representation and interactions than was necessary for participation in the other two MIPs (…).“

- line 92 : CMIP-class models : strange term
Removed „CMIP-class“.

- line 92-93 : "with a certain class of models in mind" (here the reader thinks that there are 3 different ideas); "i.e., CMIP-class models in all three MIPs" (now they seem equal) and specifically AerChemMIP required .... (now one seems to be different) : this sentence is not very nice
Agreed, changed to: „ (…) e.g., AerChemMIP required more interactive processes than the other two MIPs.“

- line 94 : "Taken together" : not nice
Removed: „There are (…)“

- line 95 : number : sounds strange
Now: „number of realizations“

- line 98 : "irradiance, sulfate, and black carbon aerosols." ! "irradiance, and sulfate and black carbon aerosols."
Thanks, corrected.

- line 105 : "." should be added at the end of the line
Added.

- line 108-109 : ... to estimate the real-world evolution and timing of emission changes ..." : if this refers to anthropogenic emissions, then it is a bit strange (timing of emissions is an externally imposed factor). Should one add "natural"?
Added: „(…) anthropogenic and natural emission changes (…)“

- line 112 : "( hist)" : should be "(hist)"
Corrected.

- line 115 : where the emissions or concentrations ! where "only" the emissions …
Changed as suggested.
- line 119 : "That is ..." ! "It implied …"
Changed as suggested.

**2.2 MIP's CROSS LINKAGES**
- line 147 : the magnitude of a perturbation "in" the radiation budget : is "in" or "on" best representing the meaning?
Revised to: „(…) characterizing the influence on the radiation budget due to a perturbation."

- line 155 : "the parallel use of preindustrial control experiments" : this is a bit confusing. Does it refer to the fact that both MIPs use the same reference simulation? Or that for most analyses two simulations are used together to derive results? Maybe use "common use".
Changed as suggested: „the common use of pre-industrial control experiments "

- line 158-161 : should one make clear that the "double calls" are used for the IRF calculation and the "triple calls" to separate aerosol direct and cloud-mediated effects?
Yes, that can be useful. Now: „(…) and a better understanding of contributions from different processes to ERF. Double calls typically refer to IRF calculations, whereas the term triple calls is used for separating cloud-mediated effects from direct effects of aerosols."

- line 166-169 : these two sentences on AerChemMIP fall within a paragraph mainly on RFMIP. Even if these two sentences have their place here, they refer to the first sentence in the paragraph, not to the sentence they immediately follow on. So the structure might be improved.
We moved the content up in the paragraph: „The RFMIP protocol included experiments to diagnose radiative forcing for greenhouse gases and aerosols as bulk quantities with setups parallel to the CMIP6 experiments for the "Diagnostic, Evaluation, and Characterization of Klima" (DECK). The RFMIP tier 1 experiments were carried out by many modeling centers. Some of these contributions, e.g., from UKESM1 and CNRM, arose because the experimental setup was identical to the request in AerChemMIP. It meant that the technical workflow for performing and postprocessing the experiments was already in place such that contributing another variant of such experiments required only little effort. ."

- line 167-169 : this is already said on line 128-130. It is allowed to mention things twice, but one should be careful with it.
Removed here: „As such, RFMIP was able to characterize forcing in CMIP for the first time."

- line 178 : "(histSST piX)" ! "(histSST-piX)"
Thanks, corrected.

- line 183-184 : "... and the specialized communities for aerosols and atmospheric chemistry that do not participate in CMIP." : my impression is that part of the AeroCOM and CCMI community participates/contributes to CMIP via actually RFMIP and AerChemMIP. So I would maybe not formulate it so strongly.
True, individuals are involved in both communities, although not the entire communities are overlapping. Now removed: „that do not participate in CMIP"

- line 195-196 : "Some key articles based ... deadline." It is unclear what the authors want to stress with this sentence. Now it feels that it stresses that there was little time and some analyses were ready only just in time, whereas it is also possible that the sentence wants to express that there was good collaboration and timing.
Now: „In fact, some key articles based on the experiments were written and submitted very close to the IPCC-AR6 WG1 deadline, which might not have been completed in time if that exchange had not happened."

- line 197-198 : until a quorum of CMIP7 model output "come" online : "comes".

Thanks, corrected and adjusted to „becomes".

- line 198 : "online" ! "available"
Now: „becomes available"

3. CHALLENGES IN THE MIP'S RESEARCH
This first paragraph is ok, but there is a lot of repetition of the same words such as "model diversity" and "model differences". I think this paragraph can be improved.
Ok, we improve it as follows.
- line 202-205 : I assume that I understand what the authors want to express - however it is not very clear. Is the line of thought : When a difference in climate response to emissions is observed between models, one cannot say whether it is due to : (i) same emissions give difference in radiative forcing; (ii) same radiative forcings give different response; or (iii) same climate perturbation gives different feedback. Or does one rather want to express that spreads
in the radiative forcing (for same emissions), and spreads in the response (for same forcing), cannot just be combined independently?
We are thinking of the former line of thought, although the second interpretation is also true. We now expand this topic in the text for clarity: „Specifically, a model-to-model difference in a climate response might be caused by various segments in the paradigm. For instance, the same emissions can lead to different ERFs, the same ERF can induce different climate responses and the same response can trigger different feedbacks across multi-model ensembles. In multi-model studies, one therefore sees inter-model spreads in forcing for the same change in atmospheric composition and model-dependent climate responses to the same forcing involving different types and magnitudes of feedbacks."

- line 211 : "elsewhere" (general) "... in the AeroCom community" (specific) : there seems to be a contradiction between "elsewhere" (which sounds general), and the rather specific mentioning of "AeroCom" at the end. I would write : "(in the Aerocom community) (Stier et al., 2013)"
Now: „Such experiments have also been used in the AeroCom community for a better understanding of reasons for model differences in aerosol forcing (…)"

- line 213 : "model-to-model diversity" : whereas on line 207, 208, and 210 "model diversity"
Thanks, now also here: „model diversity" for brevity.

3.1 COMPUTATIONAL CAPACITY ABYSS
GENERAL : This section is not very nice to read. It is also reasonably long (4 pages is a considerable part of the manuscript). I list here detailed comments, but the section as a whole should be improved.
Thank you, we have now split this section into two subsections. The first one addresses tradeoffs across MIPs and the second one tradeoffs within MIPs.

- line 231-233 : "Simpler and multi-purpose experiments would be useful and less ..." : This sentence falls a little bit out of context. So maybe start with "Also ..." or "In addition to making Tiers, …".
Agreed, we move the content of these sentences to the conclusion section where it fits better as recommendation for future MIPs.

- line 232-233 : "as long as they facilitate answering the science questions laid out by the MIPs." : I don't know whether this has to be mentioned so explicitly. I would think that the reader assumes this.
Yes, it might be clear for many, but we keep it just in case. The sentence is now included in the conclusions which possibly also less specialised scientists read.

- line 233-234 : "We propose ..." : I don't know whether this conclusion belongs already here.
Moved to conclusions.

. line 235-237 : "Simpler experiments ..." : This sentence is not very nice.
Now: „Nevertheless, multi-purpose experiments can be useful and less burdensome in terms of human and computational resources, as long as they facilitate answering the science questions laid out by the MIPs." In the conclusions.

- line 236 : the example seems to suggest that implementing a diagnostic is never a difficult task, although I think it sometimes can be.
A changed setting in the run script within physically sensible ranges does not require an implementation. Now: „in a setting for performing an experiment" for clarity.

- line 237 : "Another example is an experiment design that needs to implement ..." : grammatically not ok. So maybe: "needs the implementation"
Thanks, changed as suggested: „needs the implementation of"

- line 240 : "and the associated scientific exploitation "! and "do" the scientific exploitation
Changed as suggested.

- line 241 : "run script" : not so nice language
It is a common term in some technical environments, but now: „in a setting for performing an experiment"

- line 242 : A rule of thumb ... : the message in this sentence is rather general, and I don't know whether it is useful.
Removed.

- line 244-245 : "One could say ..." : this is not a nice sentence
Changed to: „A greater number of experiments performed creates more data for statistical analyses and for addressing a variety of research questions, but it is taxing in light of restricted resources. "

- line 244-252 : This part of the text contains a lot of "we"/"our". I would try to limit that.
Agreed, we revised the paragraph accordingly.

- line 253 : we "review" ! we "reviewed"
Changed as suggested.

- line 253 : of "the" number of experiments ! of "a" number of experiments
Now: „current status of the experiments and their usage"

- line 256 : "Model output to assess differences in forcing and response was, however, more restrictive" : is this expressed correctly?
Revised to: „ Available model output to assess differences in forcing and response was, however, limited (…)"

- line 258 : emission-driven ! "SLCF" emission-driven models (to avoid confusion with CO2 emission driven models from C4MIP)
Added as suggested, but we now use the term NTCF throughout the manuscript for consistency with IPCC-AR6. We explain both terms in Section 2.1: „The term NTCF is used in IPCC-AR6 and refers to the same term as short-lived climate forcers (SLCFs) used by (Collins et al., 2017). Both NTCFs and SLCFs refer to radiatively active atmospheric constituents whose climate effects occur primarily within two decades of their emission or formation."

- line 260-261 : in the "works" of the CMIP climate forcing task team ! "work"
Changed as suggested.

- line 267-268 : with 36 "experiments" ! with 36 "models"
Changed as suggested.

GENERAL line 271-275 : This should be improved.

- line 272-273 : Modelling centers perform the requested experiments "with the ESM which they support." : should this be mentioned explicitly? Isn't this assumed automatically?

The modelling centres do not necessarily own and develop the model on their own or perform the experiments for the scientists in all cases. Now revised to: „Modeling centers provide the resources for the requested experiments with the ESM which they support."

- line 273 : They "contribute to the decision" ... : sounds strange. Isn't it that they actually decide? Scientists at or external to the modelling centre make a request for experiments to be performed. The modelling centres contribute to that decision since they provide the computing resources, decide who gets how many computing resources and/or what experiments are carried out in which order. We add this now: „They contribute to the decision for which community-driven MIP experiments with the ESM will be conducted, e.g., through granting computational resources and prioritizing experiments to be completed."

- line 273 : "of" the ESM ! "with" the ESM

Changed as suggested.

- line 274-275 : "Not all ... giving modelers room to make their own choices." This seems to be a slightly different topic than the sentences around.

Now better linked to the previous sentences as follows: „Additional decisions for the experiments are made by the scientists interested in the MIP. There is some room to make their own choices since not all experimental settings are explicitly defined by the MIP's experiment protocols, e.g., they may use a coarser spatial resolution and to some degree less model complexity to reduce the computational burden."

- line 275-276 : "Taken together, there are inevitable tradeoffs in the final experiment design." Not nice, very general.

Now: „There are inevitable tradeoffs in the final experimental designs for individual MIPs." To introduce the concept of the three axes.

GENERAL line 287-309 : This should be improved
- line 287 : "tradeoff" (is already mentioned on line 275)

Now here with a future perspective: „Although computing power continues to grow, tradeoffs along the three axes of experimental design and prioritizations will continue to be necessary. "

- line 290 : "priority" : this is a new aspect, which has not been mentioned before. As it falls a little bit out of the suggested framework, it should be mentioned in the beginning or at the end (not in the main part of the reasoning).

Now included in the first sentence of the paragraph already, again: „Although computing power continues to grow, tradeoffs along the three axes of experimental design and prioritizations will continue to be necessary."

- line 297 : "restricts the scope for increasing computing resources" ! limits the attribution of computing resources

Changed as suggested.

- line 300 : "by" a certain degree ! "to" a certain degree

Changed as suggested.

- line 300-301 : "while retaining sufficient process detail for the scientific problem that is to be studied" : can maybe be skipped.

Removed as suggested.

- line 305-306 : "Similarly, ... is necessary." Why such a strong statement? Why not as in the former sentence : "… possible that are needed to split …"

Now as in the previous sentence: „Similarly, a separation of the response in temperature or air quality into a forced signal and a contribution from internal variability is possible. "

- line 306 : "the required ensemble size" : possibly also length of the simulation falls under this category.

Yes, now added: „The required ensemble size and length of experiments (…)"

- line 310 : "data amount" is a derived product. I would think that the amount of simulated years is the determining factor.

Now revised: „necessary number of simulated years".

- line 310 : "scientific interest" ! "research question"

Changed as suggested.

- line 311 : maybe add at the end of this sentence : "in the current setup/experiment"

Added as suggested: „in the current experiments".

- line 314 : "exact precision" ! "accuracy"

Precision is the correct term. Precision means to have a sufficiently large number of measurements or in our case data from model experiments for calculating a representative mean value. That mean, however, might still be biased compared to the truth such that accuracy would be a misleading term.

- line 314-315 : "... depends on the model due to model differences in the internal variability ..." ! "... depends on the model's internal variability that induces …"

Changed as suggested.

- line 318 : in "all" regions ! in "some" regions; or move it to the end of the sentence (then it can remain "in all regions")

Moved to the end as suggested.

- line 318-319 : the aerosol radiative effects : BC, OC, sulfate separately? Or together?

All together, changed to: „the anthropogenic aerosol radiative effects"

- line 322 : "For model responses" : vague

Revised: „For simulated climate responses, (,,,)"

- line 322 : the ensemble sizes and lengths ! the ensemble sizes and "simulation" lengths

Changed as suggested.

- line 323-324 : content is very similar to line 326-328

Now combined and shorter: „For simulated climate responses, the ensemble sizes and simulation lengths of the experiments were not sufficient for addressing all research questions of interest in the three MIPs, especially for regional responses."

- line 324-326 : very similar to line 310-314

Now include in line 310: „The signal-to-variability ratio is for instance sufficiently good for the response of the global mean of precipitation (Myhre et al., 2018, Allen et al., 2020) and the ERF in the global mean for most climate forcers in the current experiments. Specifically, (…)"

- line 328-330 : "Typically larger data amounts ... from the internal variability." : not a nice sentence

Removed.

GENERAL line 331-337 : vague, not well written. It should be improved.

Revised: „Complex models simulating many processes and their interactions are desirable and needed for specific research questions, and also pose challenges for reducing model-based uncertainty in the assessment of the climate response to various forcings."

GENERAL line 338–358 : vague, not well-written.
Revised: „Model simulations ideally converge to similar solutions for a given question, e.g., how the Earth's temperature responds to anthropogenic perturbations."

- line 341-342 : "emissions, transport, and deposition" : what about "growth" and "coagulation"? Maybe use an expression which covers all the aerosol processing.
Revised: „models simulating the evolution of different aerosol species and their interactions"

- line 349 : "had more participation" : not nice
Revised: „received more output from model experiments"

- line 349-251 : three times "hence" in three lines.
Thanks, resolved.

- line 350 : "had a different structural organization with formal experiment protocols." This sounds strange. Might the following be better : "had different structural organizations of their experimental protocols."?
Changed to: „ (…) had different structural organizations while PDRMIP started earlier and was in comparison more self-organized and flexible in the MIP life cycle."

- line 351 : "dynamic" in the MIP life cycle : "flexible"?
Changed to „flexible"

- line 352-354 : gives the impression that the emission-driven setup was the prefered one (mentioned first and using the expression "had the capability"), whereas it was the prescribed concentration setup which actually was the preferred one.
Revised: „Specifically, some of the models in PDRMIP performed experiments with prescribed emissions whereas others used concentrations resulting in an ensemble of experiments partially driven by emissions and partially driven by concentrations of climate forcers."

- line 354-357 : should be improved
Revised: „Yet, MIP experimental protocols do not prescribe the level of process complexity in and the resolution of ESMs. This freedom is well justified since ESMs might otherwise not be able to participate in a MIP if they can not fulfill stricter requirements. A wider participation of ESMs in MIPs ensures a sufficiently large multi-model ensemble needed to robustly quantify forcings and climate responses considering structural model differences."

- line 357-358 : very general sentence
Revised: „A full exploration of the role of climate-composition feedbacks with focus on biogeochemical processes, however, remains an outstanding challenge due to this difficulty."

3.2 PROCESS UNDERSTANDING ABYSS
- line 363 : "Moreover" : Why adding "moreover"?
Removed.

- line 366 : ... for all aspects. : this is vague.
Removed and point now to the example in the following sentence more explicitly: „(…) as one would hope. For example, (…)"

- line 366 : "Clouds and circulation are for instance outstanding challenges" : maybe add " Correctly representing …" or "Understanding …".
Added as suggested: „correctly representing clouds and circulation (…)"

- line 368-372 : "There is value ..." : not a nice sentence. Maybe just start with "Multi-model inter-comparisons shed light on …".

Changed as suggested: „Multi-model inter-comparisons shed light on (…)"

- line 371 : which model "output" they ! which model they

Changed as suggested.

- line 372 : since not all "experiments" ! since not all "models"

Changed as suggested.

- line 372 : "e.g., some models might miss processes and interactions that might be crucial to address the research question." Isn't this already mentioned earlier?

Removed.

- line 373 : "Results of MIPs alone can not fully characterize the uncertainty ..." : Possibly start a new paragraph as this is a new thought.

Done.

- line 373-374 : "if it is at all possible" : I don't know if this general thought contributes something to the text.

Removed.

- line 376 : or "ideally" in synergy : I would skip "ideally"

Removed.

- line 380 : would allow us to quantify ... ! would allow us to "first" quantify

Added as suggested.

- line 392-394 : Aren't both sentences expressing the same?

Removed: „However, due to their potential magnitude, these methane feedbacks are important yet uncertain. "

GENERAL line 413-447 : not so nice to read. Please improve.
- line 413-414 : "satisfyingly" simulated : not nice

Changed to: „well simulated"

- line 416 : model concensus ! "erroneous" model consensus?

Yes, added: „erroneous model consensus"

- line 419-420 : no ESM in CMIP5 "and" CMIP6 : no ESM in CMIP5 "or" CMIP6

Changed as suggested.

- line 424-425 : "Winds emit" : not nice

Changed to: „ Winds control the emission and transport of desert-dust aerosols (…)"

- line 427-434 : vague, not nice

Changed to: „ Winds control the emission and transport of desert-dust aerosols and the soil erodibility is influenced by the available moisture from rain events. There is a large diversity in model-simulated regional changes in winds and precipitation in response to warming which in turn introduces uncertainty in simulated dust trends."

4. OPPORTUNITIES
4.1 AUGMENTED ESMS
4.1.1 MACHINE LEARNING WHERE USEFUL
- line 497 : Training emulators require ! require(s)

Thanks for spotting this. Corrected.

- line 506 : ... remanins. ! ... "however" remains.
Added as suggested.

4.1.2 KILOMETER-SCALE EXPERIMENTS
GENERAL : this section is not very well written. The different thoughts should be ordered better and presented more clearly, and repetition should be avoided.
- line 513 : Representing clouds ... ! Representing clouds "correctly" …
Added as suggested.
- line 518 : "These meteorological variables" : a little bit unclear what it refers to, but the reader will probably assume "clouds and precipitation".
Changed to: „These processes (…)"

- line 519 : "their" : also not so clear what it refers to.
Changed to: „ and associated effects on the atmosphere"

- line 522 : "hold the potential for surprises in understanding climate responses" : the reader is left in doubt what to expect from this. So maybe be more specific.
Added: „ (…) e.g., with respect to future projections of temperatures and rare high-impact events (…)".

- line 526 : "of a few weeks to years" : "years" (in plural) gives the impression that one can already do reasonably long simulations. Is that correct? Or should one write "weeks to months"?
It depends on the exact resolution and model, but global kilometre-scale experiments have already been done for a few years (Hohenegger et al., 2023). We add the citation as link to further detail.

- line 526-528 : "... in maintaining concentrated emissions, non-linearities in chemistry, and atmospheric transport of pollutants" : my impression is that the first two aspects implicitly assume and express a benefit from high resolution (maintaining strong gradients in concentration, and non-linearities can be resolved), whereas the 3rd aspect misses something, to what the transport improves : e.g., "and fine-scale resolved transport of pollutants"?
We added: „fronts in the atmospheric transport of pollutants"

- line 531-532 : strange sentence
Revised: „Global coupled atmosphere-ocean simulations with a few kilometers resolution can be done and progress has been made in incorporating some representation of atmospheric composition for example the carbon cycle (…)"

- line 531-534 : "... can be done." [in first sentence] "Such model experiments have been carried out ..." [secondsentence]. That sounds contradictory between the first and second sentence.
Removed the second sentence.

- line 534 : "...to answer open questions concerning climate change to provide information for societal needs." Why is this rather general information mentioned here?
Removed.

- line 528-529 : "... to advance the understanding of climate and air quality interactions." This is very similar to line 535-536 : "... for the understanding of interactions of atmospheric composition and climate change."
Here, now removed for brevity.

- line 536-542 : "For some questions ... . One question that can better be answered ..." : maybe use "Another question…"
Changed as suggested.

- line 543 : "will not be able to entirely rely on" : this is an understatement I think. With more nuance: "will not be able to mainly rely on"

Changed as suggested.

- line 544 : at a resolution 1 km ! at a resolution "of" 1 km

Added as suggested.

- line 547 : "The first possible and computationally smart way" ! "The first possible way"

Changed as suggested, although this should also be the computationally smarter way, at least for the time being.

- line 553 : our near-future "works" ! our near-future "work"

Changed as suggested.

- line 557-558 : "... UKESM2 and CESM aim also .... An ensemble of regional composition-climate models exists ...": I would not call a group of two models already an ensemble.

Changed to: „ At least two different regional composition-climate models therefore (…)"

- line 557-562 : ... in "our" past MIPs. / As such "our" need for experiments ... / ... although "we" are not averse to the idea ... : I would avoid using too often "we" or "our".

Changed to: „As such a need for experiments with classical global ESMs is retained, at least for CMIP7, although moving towards global kilometer-scale modeling with a sufficient coupling of physical processes to aerosols and chemistry to address the community's research interests will be a goal to aspire to."

**4.2 IMPROVED DIAGNOSTICS AND ANALYSIS**
**4.2.1 RADIATIVE FORCING CALCULATIONS**
- line 592-594 : "Differently from RFMIP, AerChemMIP found ... before calculating the ERF". This is already mentioned earlier, please indicate that.

Added the reference to the section in brackets.

**4.2.2 SYNERGY WITH IMPACT ASSESSMENTS**
- line 629 : "the opportunity" ! "an opportunity"

Changed as suggested.

- line 634 ; "... a smaller model spread in O3". (too general) Shouldn't it be "... in tropospheric O3 burden."?

The tropopause height would also affect the burden calculated for the stratosphere. We nevertheless added „tropospheric" due to the relevance for air quality: „(…) smaller model spread in tropospheric $O_3$, which is relevant for air quality assessment."

**5. CONCLUSIONS**
- line 657 : "(ESM)" ! "(ESMs)" (as on line 49)

Changed as suggested.

- line 669-671 : this sentence is grammatically a bit unbalanced : "(i) to speed up and improve ..., (ii) to data mining..., and (iii) to develop ... ." Maybe add a verb to "(ii) to data mining" : "to do data mining".

Thanks, added „to do" as suggested.

- line 674 : or many ensemble members ! or "sufficiently" many ensemble members

Changed as suggested.

- line 681 : closer collaborations with computer "science" ! with computer "scientists"

Changed as suggested.

- line 686 : ”a” international vibrant community ! ”an”
Changed as suggested.

- line 689-690 : The planning of ... and ... ”are” currently ! ”is”
Changed as suggested.

- line 690-693 : sentence sounds a bit strange
Changed to: „Keeping the two MIPs separate has advantages over connecting both initiatives in a single new MIP. The MIP names and the general ideas of RFMIP and AerChemMIP are already known through CMIP6. Moreover, the science questions and the associated experiment protocols are clearer, and the workload for coordination and management is smaller for the separate MIPs.“

FIGURES
Figure 1 : The drawing of the arrows should be improved : no intermediate arrows. Possibly the authors can modify the figure to even better represent the perturbation-response paradigm.
We have redrawn the figure. All arrows are now solid lines, and segments have been rearranged. The depiction has further been modified to include irradiance at the top of the atmosphere, air quality and physical feedbacks to better represent the paradigm.

Perturbation - response paradigm

Figure 3 : Frederiksen et al. [2021] : not in reference list. Possibly homogenize references with the main text (adding”,” before year).
Added Frederiksen et al. (2021) in the reference list and adjusted the style of the references in Figure 3 as suggested.

TABLES
Table 1, caption : explain ”X” (as it was done in the main text); maybe explain piClim-2xEms (maybe refer to them also as piClim-2xX, but explain what it means).
Added in the caption: „Listed experiments are fully coupled atmosphere-ocean experiments (*hist-X*), experiments with prescribed transient changes in sea-surface temperatures and sea-ice from a historical experiment (*histSST-X*) and experiments with prescribed sea-surface temperatures and

sea-ice at pre-industrial level (*piClim-X*), where *X* refers to single or a combination of several climate forcers and *piClim-2xX* refers to experiments with prescribed doubled emission fluxes. "

Table 2, 3rd column : "little" change in aerosol forcing between 1970s and 2000s ! "Little" change
…
Changed as suggested.

Table 2, 4th column : O'Connor et al. (2021)";" O'Connor et al. (2022) ! ","
Changed as suggested.

Table 3, 4th column : The numbers on the 3 lines containing "Total number coupled experiments", "Total number histSST experiments", and "Total piClim experiments" possibly do not make a lot of sense. I assume that they are obtained by summing the individual numbers, whereas the number of studies using at least one of the experiments is probably slightly lower.
Yes, we added the numbers from the individual entries to list totals, now explicitly stated in the table caption: „Totals are calculated by adding the individual numbers listed aloft and are generous estimates since some publications used more than one experiment type. "

Table 4 : and their role for climate "changes" ! "change"
Changed as suggested.

Table 4 : Aerosol absorption substantially "differ" across ESMs ! "differs"
Changed as suggested.

Table 6 : Improved diagnostic for "O3" ! "3" in O3 should be a subscript.
Changed as suggested.

---

## Author Response (AR3)

**Point-by-point reply for the manuscript „Interactions between atmospheric composition and climate change - Progress in understanding and future opportunities from AerChemMIP, PDRMIP, and RFMIP" by Fiedler et al.**

Dear Sophie Valcke,

Thank you very much for the appraisal of our manuscript. We appreciate the time for the review and the support for further improving the clarity of the text. Below are our replies in blue to your suggestions in black.

L. 57-61: "The climate modeling community creates multi-model ensembles of a common set of ESM experiments with the same perturbations applied. The simulated climate responses can differ across a multi-model ensemble. This diversity in responses may for instance be due to differences in process complexity and interactions within the respective ESMs. Experimental protocols are used to create multi- model ensemble simulations for specific ESM experimental setups. These aim to better understand the reasons for the diversity in climate responses and feedback and to create future climate projections." : First and before last sentences are redundant; I suggest: "The climate modeling community creates experimental protocols to set up multi-model ensembles of a common set of ESM experiments. The simulated climate responses can differ across the multi-model ensemble members in response, for instance, to differences in process complexity and interactions within the respective ESMs. The aim is to better understand the reasons for the diversity in climate responses and feedback."
Changed as suggested.

L.87: "... consider structural differences concerning the design and the level of complexity between ESMs." => "... consider structural differences in the design and complexity of the different ESMs."
Changed to : "considered structural differences in the design and complexity of the different ESMs".

L. 90: "... radiative forcing diversity to anthropogenic perturbations ..." => "radiative forcing linked to anthropogenic perturbations …"
Changed as suggested.

L. 173: "The RFMIP protocol included experiments to diagnose radiative forcing for greenhouse gases and aerosols as bulk quantities with setups parallel to the CMIP6 experiments for the "Diagnostic, Evaluation, and Characterization of Klima" (DECK)" => "The RFMIP protocol included experiments to diagnose radiative forcing from greenhouse gases and aerosols as bulk quantities with setups common to CMIP6 "Diagnostic, Evaluation, and Characterization of Klima" (DECK) experiments."
Changed to: „The RFMIP protocol included experiments to diagnose radiative forcing from greenhouse gases and aerosols as bulk quantities."

L. 216-220: "A major challenge to further advancing the understanding of climate change with ESMs is that differences in their results for individual segments of the perturbation-response paradigm are not independent of other segments. Specifically, a model-to-model difference in a climate response might be caused by various segments in the paradigm. For instance, the same emissions can lead to different ERFs, the same ERF can induce different climate responses and the same response can trigger different feedbacks across multi-model ensembles. In multi-model studies, one therefore sees inter-model spreads in forcing for the same change in atmospheric

composition and model-dependent climate responses to the same forcing involving different types and magnitudes of feedbacks." The text you added makes the paragraph even more fuzzy from my point of view. In the first sentence, you discuss dependence of what you call the "segments of the perturbation-response paradigm". In the second sentence, you discuss the fact that differences may be linked to more than one segments; I am not sure then why the second sentence starts with "Specifically". Then, if I understand well, in the 3rd sentence, you give an example of different responses in different segments. Please simplify. Consider removing the second sentence. In all cases, remove "across multi-model ensembles" in "different feedbacks across multi-model ensembles" as I think you mean that the difference in the feedbacks are differences across the ensemble members (but this is confusing as one may understand that the feedbacks themselves occur between the ensemble members).

We removed the second sentence and "across multi-model ensembles" to simplify this part as suggestion.

L.226: "for a better understanding of reasons for model differences" => "for a better understanding of model differences"
Changed to: „for a better understanding of model diversity".

L 249: "newly requested diagnostic output that is not yet available in the standard variable list of ESMs, e.g., for RFMIP-IRF" => remove "that is not yet available in the standard variable list of ESMs"
Removed.

L.252-255: "In this case, it takes longer to finish the experiments and to do the associated scientific exploitation, e.g., in the case of RFMIP-SpAer several years after the work began (Fiedler et al., 2023), which is long compared to easy experiments that modelers can quickly set up via a simple change in a setting for performing an experiment, e.g., for RFMIP-ERF, thanks to prior work on the development and testing of models." This is a way too long and over convoluted sentence and does not bring much to the previous sentence. I suggest removing it and just modifying the end of the previous sentence as "...to carry out the work including coding, testing, performing the experiments and associated scientific exploitation."
Changed as suggested.

We also shortened text in the following paragraph: „In preparation for the next phase of AerChemMIP and RFMIP, the type and number of experiments in the experimental protocol will therefore be revised based on a refined set of research questions and the desire to reduce the computational burden for modeling centers as much as possible. In this process, AerChemMIP activities will be closely coordinated with other community MIPs with common or similar interests.", to make the text more coherent with the experiment review in the following paragraph.

L.265: "we reviewed the current status of the experiments" => "we reviewed the status of the experiments"
Changed as suggested.

L.287: "They contribute to the decision for which community-driven MIP experiments with the ESM will be conducted, e.g., through granting computational resources and prioritizing experiments to be completed." => "They contribute to the decision as to which MIPs will be conducted with which priority, e.g., through granting computational resources."
Removed this paragraph as suggested in next comment.

L.288-290: "Additional decisions for the experiments are made by the scientists interested in the MIP. There is some room to make their own choices since not all experimental settings are explicitly defined by the MIP's experiment protocols, e.g., they may use a coarser spatial resolution and to some degree less model complexity to reduce the computational burden." Another example of the same idea expressed twice. Simply remove this paragraph as the same ideas are expressed with slightly different word in the paragraph just after.
Removed as suggested.

L.291: "There are inevitable tradeoffs in the final experimental designs for individual MIPs. Such choices can be categorized along the three axes of (1) model complexity addressing how many process interactions ESMs allow or how much fidelity processes have, (2) model resolution referring to the grid spacing of the model, and (3) simulation length covering the length and number of simulations in an ensemble of different experimental setups per ESM." => "There are inevitable tradeoffs in the final experimental choices that can be categorized along the three axes of (1) model complexity addressing how many process interactions ESMs allow or how much fidelity processes have, (2) model resolution referring to the grid spacing, and (3) simulation length covering the length and number of members in ensemble simulations. "
Changed to: „There are inevitable tradeoffs in the final experimental choices that can be categorized along the three axes of (1) *model complexity* addressing the number of process interactions represented in ESMs and the fidelity of the processes simulations, (2) *model resolution* referring to the grid spacing, and (3) *simulation length* covering the length and number of members in ensemble simulations. "

L.339: remove "of the experiments" in "the ensemble sizes and simulation lengths of the experiments"
Removed as suggested.

L.445-455 (L. 427-434 in version 2) : Referee #2 wrote that these sentence are "vague, not nice" but you did not change anything. Please rephrase, clarify, simplify.
The paragraph has now been removed and relevant information integrated in the previous section on natural processes: „ESMs differently simulate desert-dust aerosols (e.g. Evan et al., 2014; Checa-Garcia et al., 2021; Zhao et al., 2022) and do not reproduce the magnitude of the reconstructed dust increase from the pre-industrial to the present-day (Kok et al., 2023). (…) Skillful simulations of winds (e.g., Bony et al., 2015) and rain (e.g., Fiedler et al., 2020) are challenges for ESMs, which in turn introduce uncertainty in simulated dust trends."

L.528-530 : "Much finer spatial resolutions with horizontal grid spacings of a few kilometers hold the potential to overcome some of the long-standing challenges concerning the representation of clouds, precipitation, and circulation in global climate simulations, which would require a step change in collaboration between climate science and high-performance computing (Slingo et al., 2022)." is another example of a too long sentence; please cut in two parts, end first sentence after "global climate simulations"; in the second part, it is not clear what "which" refer to, please rephrase.
We shortened it to: "Much finer spatial resolutions with horizontal grid spacings of a few kilometers hold the potential to overcome some of the long-standing challenges concerning the representation of clouds, precipitation, and circulation in global climate simulations.". We removed the subordinate clause here and include the information in the following paragraph for better readability: „Global kilometer-scale simulations for climate change assessments would require a step change in collaboration between climate science and high-performance computing (Slingo et al., 2022)."

L.531 : Change "in coarse resolution models of several tens to hundreds of kilometers of grid spacings" for "in models with resolution of several tens to hundreds of kilometers".
Changed as suggested.

L. 560-561: "especially in the context of a MIP since fully coupled ESMs with interactive aerosols and chemistry at a resolution of 1 km fast enough to perform multi-decadal simulations are unlikely to be ready in the time of CMIP7": please rephrase for something like "since fully coupled kilometer-scale ESMs with interactive aerosols and chemistry fast enough to perform multi-decadal simulations are unlikely to be ready in the time of CMIP7"
Changed to: "(…) since several fully coupled kilometer-scale ESMs with interactive aerosols and chemistry fast enough to perform multi-decadal simulations are unlikely to be ready in the time of CMIP7."

L. 564-566: "This approach is suitable to answer some but not all research questions in our community. For instance, the response of dust emission fluxes to changes in winds and moisture can be addressed with offline modeling and allows to identify underlying reasons for changes and model differences in the dust response (Fiedler et al., 2016), but the implication of such dust emission changes for air quality and climate responses can not be quantified with such an approach." ; please simplify for something like "This approach is suitable to answer some but not all research questions in our community, for instance, the response of dust emission fluxes to changes in winds and moisture (Fiedler et al., 2016), but not the implication of such dust emission changes for air quality and climate responses."
Changed as suggested.

L. 691: "sufficiently long experiments or sufficiently many ensemble members" is not grammatically correct; I suggest ""sufficiently long experiments or enough ensemble members"
Changed as suggested.

L 692: "to better distinguish climate and air quality responses to atmospheric composition changes from" is another example of useless words in a sentence; it should simply be ""to better distinguish climate and air quality responses to atmospheric composition from"
Changed as suggested.

L.711: "multi-purpose experiments can be useful and less burdensome in terms of human and computational resources" => "multi-purpose experiments can be useful and associated to less burden in terms of human and computational resources"
Changed to: „multi-purpose experiments can be useful and contribute to reducing the use of human and computational resources"

L.712: "as long as they facilitate answering the science questions laid out by the MIPs" is one example of an obvious sentence that does not bring much to the text
Removed.

Figure 1 captions: "are simulated with a model configuration coupled to an ocean model." => "can be simulated with a model configuration including coupling to an ocean model."
Changed as suggested.

Additional comments and propositions of modifications
L.62-64: "Results from multi-model intercomparison projects (MIPs) are widely used to advance scientific understanding and inform stakeholders on climate change. The most prominent

example is the Coupled Model Intercomparison Project (CMIP, Meehl et al., 2000) that has contributed ...": I don't agree that CMIP is an example of à MIP; instead, CMIP is composed of several MIPS, please rephrase.

Now: „The Coupled Model Intercomparison Project (CMIP, Meehl et al., 2000) has contributed through multiple phases to the assessment reports of the Intergovernmental Panel on Climate Change (IPCC, Meehl et al., 2023), e.g., the sixth phase of CMIP (CMIP6, Eyring et al., 2016) created experiments that were also used in the sixth IPCC assessment report (IPCC-AR6)"

L.85-100: Some lines use the past tense and other the present tense, please standardise .
Thanks, it is written in past tense now.

L.88: "for the answer to the MIP's question" => "in the answer to the MIP's question "
Changed as suggested.

L.115: "and refers to the same term as short-lived climate forcers (SLCFs) used by Collins et al."
=> "and refers to the same concept than short-lived climate forcers (SLCFs) used by Collins et al."
Changed to: „NTCF is used by Collins et al. (2017) and is the same as short-lived climate forcers (SLCFs) used in IPCC-AR6."

L 155: I don't understand what "quantify radiative forcing within and outside of the three MIPs" means, please clarify.
We mean that the concept of forcing calculation is also followed by others that are not participating in the MIPs. We removed it, now: „(…) consistent methodology to quantify radiative forcing, which facilitated easier comparisons across CMIP6."

L.268: Change "out of the" for "of the" in "Available model output to assess differences in forcing and response was, however, limited, e.g., output for the mid-visible aerosol optical depth is available only for 45 out of the 67 models providing historical experiments."
Changed as suggested.

L.345: "and also pose challenges for reducing model-based uncertainty" => "but pose challenges in reducing model-based uncertainty"
Changed as suggested.

L.365: remove "in" in "do not prescribe the level of process complexity in a …"
Changed as suggested.

L.545: "globally for restricted time periods of a few weeks to years" : be more specific on the number of years so to be coherent with the first part of the sentence.
Revised to: „Kilometer-scale experiments with horizontal grid spacings finer than 10 km are presently only possible for climate studies on limited area domains or globally for restricted time periods of a few weeks to single years."

L.569: make a new paragraph before "We perceive dynamical downscaling …"
Done.

L.645: As suggested by referee #2: "There is the opportunity" => "There is an opportunity" (you did not make the change).
Thanks for spotting it, it is now changed as suggested.

L.710: "is smaller for the separate MIPs" => "than what it would be for one converged MIP"

Added as suggested.

Table 1: The additions made in the legend are not self-coherent. For example "hist-X" does not appear as is in the column (does it refer to "hist" and/or "hist-piAer" ?) while "histSST-X", "piClim-X and "piClim- 2xX" do appear as is. Also "piControl" and all "ssp370..." are not described; please change.

Thanks, we now simplified the entries in the experiment column for a better coherence and introduce all experiment abbreviations in the caption: „Key results from the three MIPs for their research topics. Listed experiments are fully coupled atmosphere-ocean experiments for the historical time period (*hist-X*), experiments with prescribed transient changes in sea-surface temperatures and sea-ice for the historical time period (*histSST-X*), experiments with prescribed sea-surface temperatures and sea-ice at pre-industrial level (*piClim-X*), fully coupled atmosphere-ocean experiments for future projections using scenario SSP3-7.0 (*ssp370-X*), experiments for future scenarios with prescribed sea-surface temperatures and sea-ice (*ssp370SST-X*), where *X* refers to single or several climate forcers. Experiments with prescribed doubled emission fluxes are listed as *piClim-2xX*, pre-industrial control experiments as *piControl,* and experiments with abruptly quadrupled $CO_2$ concentrations as *Abrupt-4xCO2*."

Finally, I see that Table 4 has thoroughly changed without justification; can you make some comments on why you made all those changes?

These changes were part of improving the readability and clarity that was requested. In Table 4, we alphabetically sorted the rows and adjusted words of the entries.

---

## Author Response (AR4)

**Point-by-point reply to the review of gmd-2023-29 manuscript :**
**„Interactions between atmospheric composition and climate change – Progress in understanding and future opportunities from AerChemMIP, PDRMIP, and RFMIP"**

We thank the two anonymous reviewers and the editor Sophie Valcke for the review of the manuscript. The comments were useful to improve the presentation of ideas and include additional content. We appreciate the collegial support and add our thanks in the acknowledge section. Please find below our replies in blue to the comments in black. We enclose the revised manuscript with marked changes.

Thanks you for your careful revision of the manuscript, which gained in readability.
I think there is still one sentence that needs to be improved, i.e. L.216: "One therefore sees intermodel spreads in forcing for the same change in atmospheric composition and model-dependent climate responses to the same forcing involving different types and magnitudes of feedbacks. " I am not sure what "inter-model spreads in forcing for the same change ..." means; can you please rephrase more clearly?
With best regards,
Sophie Valcke

Thank you, we change the sentence to: „One can therefore see an inter-model spread in forcing, even when the same perturbation in the atmospheric composition is prescribed in the models, and an inter-model spread in climate responses, even if the simulated forcing would be identical across the models. "

In addition to the technical correction identified by Sophie Valke, please also bring your Code and Data availability section into line with GMD policy. Specifically:
1.  The section must be exactly called "Code and Data Availability"
Changed as suggested.

2. Due to the nature of the work, I expect that there is no code associated with the manuscript. You can simply say that.
We add: „Due to the nature of this review, there is no model code associated with this manuscript."

3. "via ESGF" and "via WDCC" are not data citations. Please put data references for the relevant datasets on those services in the bibliography and cite them from this section. Currently there is not enough information for a reader to find the data you are referring to.
The full set of data citations would be

[revised manuscript text omitted]